# Homopolymer switches mediate adaptive mutability in mismatch repair-deficient colorectal cancer

Mismatch repair (MMR)-deficient cancer evolves through the stepwise erosion of coding homopolymers in target genes. Curiously, the MMR genes MutS homolog 6 (*MSH6*) and MutS homolog 3 (*MSH3*) also contain coding homopolymers, and these are frequent mutational targets in MMR-deficient cancers. The impact of incremental MMR mutations on MMR-deficient cancer evolution is unknown. Here we show that microsatellite instability modulates DNA repair by toggling hypermutable mononucleotide homopolymer runs in MSH6 and MSH3 through stochastic frameshift switching. Spontaneous mutation and reversion modulate subclonal mutation rate, mutation bias and HLA and neoantigen diversity. Patient-derived organoids corroborate these observations and show that MMR homopolymer sequences drift back into reading frame in the absence of immune selection, suggesting a fitness cost of elevated mutation rates. Combined experimental and simulation studies demonstrate that subclonal immune selection favors incremental MMR mutations. Overall, our data demonstrate that MMR-deficient colorectal cancers fuel intratumor heterogeneity by adapting subclonal mutation rate and diversity to immune selection.

In human cells, DNA mismatch repair (MMR) is performed by protein complexes consisting of MutL homolog 1 (MLH1) and PMS1 homolog 2 (PMS2), known as MutL$_\alpha$, and MutS homolog 2 (MSH2) and MutS homolog 6 (MSH6), known as MutS$_\alpha$[1]. Alternatively, MSH2 can pair with MutS homolog 3 (MSH3) in a complex called MutS$_\beta$. MutS$_\alpha$ and MutS$_\beta$ each function as DNA mismatch detection modules with partially overlapping specificities, whereas MutL$_\alpha$ (MLH1/PMS2) executes MMR. Although the mutagenic impact of isolated MSH6 or MSH3 loss is relatively mild, combined MSH6/MSH3 inactivation in model systems drives a robust hypermutator phenotype[2]. Importantly, while MMR had previously been treated as a single linear pathway focused on postreplicative mismatch correction, recent studies indicate that MutS$_\alpha$ and MutS$_\beta$ also participate in the repair of endogenous mutational processes during interphase (for example, due to 5-methylcytosine deamination or oxidative damage) independent of MLH1 (refs. 3–5; Fig. 1a,b). Overall, these studies suggest that MutS cooperates with MutL during canonical postreplicative repair of misincorporated bases,

while MutS can liaise with other partners such as MBD4 in the interphase noncanonical repair of endogenous DNA damage.

Loss of MMR proficiency occurs in about 15% of colorectal cancers (CRCs) resulting in the accumulation of single-nucleotide mismatches and frameshift variants due to short insertion and deletion (InDel) mutations in repetitive homopolymer sequences[6]. In most cases, this is due to sporadic *MLH1* hypermethylation. The relentless accumulation of somatic variants renders MMR-deficient (MMRd) tumors immunogenic and provokes extensive immunoediting[7]. While many of the genetic targets associated with immune escape (for example, HLA (human leukocyte antigen) complex and B2M (beta-2 microglobulin) mutations) have been characterized, the evolutionary trajectories MMRd tumors take to navigate their immune selection landscape remain unknown[8].

Here we visualize the clonal architecture of evolving MMRd tumors to allow joint analysis of individual tumor subclones and the immune microenvironment at clonal resolution. We find that subclonal MMRd

✉e-mail: h.j.g.snippert@umcutrecht.nl; m.jansen@ucl.ac.uk

lineages harness hypermutable homopolymer sequences in *MSH6* and *MSH3* to adapt cellular mutation rate and mutation bias to subclonal immune selection. This strategy allows MMRd tumor subclones to engage in an evolutionary arms race with the evolving immune system and efficiently explore immune adaptation solutions while minimizing the deleterious impact of prolonged genomic hypermutation on cellular fitness.

## Results

### Subclonal *MSH6* and *MSH3* homopolymer frameshifts drive increased mutation burden

The loss of DNA MMR in microsatellite (MS)-instable cancer drives tumor progression, but also provokes the unbridled accumulation of neoantigenic and deleterious mutations. We set out to investigate how growing MMRd cancers manage this balancing act between adaptive and deleterious mutations by screening for gene mutations that modulate tumor mutation burden (TMB) in a large whole-genome sequencing (WGS) dataset of 217 MMRd CRCs from the Genomics England (GEL) 100,000 Genomes Project. Given that MMRd cancers predictably progress through the erosion of coding homopolymers in microsatellite instability (MSI) target genes such as *TGFBR2* (transforming growth factor beta receptor 2; involved in TGFβ-mediated growth inhibition) and *BAX* (BCL2-associated X; involved in apoptosis regulation), we hypothesized the occurrence of homopolymer variants that correlate with mutation burden. To explore this, we carried out multiple linear regression analysis for the relationship between homopolymer frameshifts in MSI target genes and total mutation burden, controlling for patient age and tumor purity (Methods, 'GEL 100,000 Genomes CRC dataset').

Homopolymer frameshift mutations in *MSH3* and *MSH6* revealed a strong positive correlation ($P = 9 \times 10^{-4}$ and $P = 3.6 \times 10^{-3}$, respectively) with mutation burden in this large MMRd CRC dataset (Fig. 1c and Supplementary Table 1a). This result was surprising because, while frequent *MSH6* and *MSH3* homopolymer alterations have previously been described in large-scale compendium studies of MMRd cancers[9–11], these subclonal homopolymer mutations had so far been disregarded as neutral passengers in the context of preceding (truncal) MMR deficiency. As a control, we examined the relationship between homopolymer frameshift mutations of coding microsatellites (MSs) in other frequently hit MMR gene targets in our multiple linear regression model, again controlling for patient age and tumor purity. This showed no correlation with mutation burden for any of these frequently hit targets, indicating that the positive correlation with mutation burden was specific to *MSH3* and *MSH6* homopolymer InDels (Fig. 1c and Supplementary Table 1a). Combined *MSH3* and *MSH6* frameshifts had an additive effect on mutation burden compared to either mutation alone (Supplementary Table 1b). Homopolymer frameshift mutation of *MSH3* or *MSH6* increased mutation burden from a baseline estimate of 161,267 mutations by 88,038 and 63,675 mutations, respectively, whereas homopolymer frameshift mutation of both was associated with an increase of 139,338 mutations. In these bulk sequencing data, frameshift mutations of *MSH3*, *MSH6* or both were observed in 79% (172/217) of cases.

*MSH6* and *MSH3* each contain a coding homopolymer tract (C8 and A8, respectively), which acts as a hypermutable site. Indeed, the majority of *MSH6* and *MSH3* somatic mutations in the GEL cohort occurred at these homopolymers (Fig. 1d and Supplementary Table 2). As expected, these mutations are overwhelmingly subclonal as revealed by analysis of *MSH6* and *MSH3* homopolymer frameshift variant allele frequencies (VAFs; inset in Fig. 1e). As a further control, we compared the frequency of homopolymer frameshift mutations in these genes to the proportion of all exonic length 8 C:G or A:T homopolymers mutated across the cohort and found highly significant enrichment of homopolymer frameshift mutations in the *MSH6* C8 homopolymer (37.3% versus 23.4%, $P = 1.9 \times 10^{-6}$) and the *MSH3* A8 homopolymer (67.3% versus 9.9%; $P < 2.2 \times 10^{-16}$).

Ranking our overall cohort by mutation burden illustrates the relationship between *MSH6* or *MSH3* homopolymer frameshift and overall TMB (Fig. 1e). Stratifying by *MSH3* and/or *MSH6* homopolymer frameshift mutation status demonstrated a clear stepwise increase for single-nucleotide variants (SNVs), InDels and overall mutation burden with incremental MMR homopolymer mutations (Fig. 1f–h). As a further control, we restricted our analysis to cases with confirmed truncal loss of MutL$_\alpha$ (MLH1/PMS2), which corroborated the relationship between *MSH6*[F1088fs]/*MSH3*[K383fs] and increased mutation burden (Extended Data Fig. 1a–c and Supplementary Note 1).

As further validation, we analyzed The Cancer Genome Atlas (TCGA) whole-exome sequencing (WES) data[9]. MSI cancers from the colorectal ($n = 48$), uterine ($n = 67$), stomach ($n = 63$) and esophageal ($n = 3$) cohorts were identified and pooled. This analysis confirms the stepwise relationship between increased mutation and neoantigen burden in MSI tumors with *MSH6* and/or *MSH3* homopolymer frameshift mutations (Extended Data Fig. 1d–i). Together, these data from two bulk sequencing cohorts suggest that homopolymer frameshift mutations in *MSH3* and *MSH6* are functional targets of MSI and drive increased mutation burden. We do not exclude the possibility that other, non-homopolymer mutations in *MSH6* and *MSH3* also drive increased subclonal mutation burden; however, these were distinctly less common in the GEL cohort ($n = 27$ *MSH6* and $n = 18$ *MSH3* missense variants, none of which were recurrent), limiting functional interpretation.

*MSH6* and *MSH3* homopolymer frameshift mutations are also frequently found in MMRd cancer cell lines[12]. Indeed, previous in vitro work showed that *MSH6* homopolymer length varies between isogenic MMRd cell line isolates and fluctuates over time to drive spontaneous loss and restoration of MSH6 expression by moving in and out of the reading frame[13]. Moreover, concomitant inactivation of MSH6 and MLH1 in isogenic cell lines increased the cellular mutation rate compared to the inactivation of MLH1 alone. These *MSH6* homopolymer length fluctuations may thus provide a potential substrate for selection during tumor evolution; however, this could not be evaluated in the in vitro context[13].

Accurately determining the allelic status of *MSH6* and *MSH3* from bulk sequencing is complex due to the polymorphic nature of these homopolymers in clonal mixtures. Fortunately, immunohistochemistry (IHC) faithfully tracks MSH6 expression and is routinely used clinically to detect loss of MSH6 protein expression in MMRd tumors. To delineate the frequency of subclonal MSH6 loss in a large clinical series, we prospectively investigated a series of 546 unselected CRCs using whole-slide MMR IHC (Extended Data Fig. 2a). Of these cases, 88 (16%) were MMRd, of which 77 showed combined MLH1 and PMS2 loss, and 6 cases showed isolated PMS2 loss. We found that 32 cases (36%) showed subclonal MSH6 loss within the context of MLH1/PMS2 loss (Extended Data Fig. 2b,c). Many of these tumors showed multiple geographically isolated MSH6-deficient subclones, which varied substantially in size (Extended Data Fig. 2d).

Remarkably, within larger MSH6-deficient tumor clones, we frequently found numerous nested subclones with intact MSH6 expression (Fig. 2a–e and Extended Data Fig. 2d). To further delineate the identity of these patches, we used a multiplex IHC panel (Methods). Colabeling for MSH6 and a pan-cytokeratin (pan-CK) epithelial marker showed scattered epithelial tumor ribbons and single tumor cells with intact MSH6 labeling within MSH6-deficient tumor regions (Fig. 2f, insets ii and iii). To verify that this nested patchwork of MSH6 protein labeling faithfully reflected the *MSH6* genotype, we carried out detailed laser capture microdissection (LCM) followed by Sanger sequencing of the MSH6-deficient lineage, the MSH6-proficient nested subclone and background MSH6-proficient tumor cells from each of three tumors. We indeed found sequential loss and restoration of the C8 homopolymer reading frame, suggesting that the *MSH6* homopolymer had dynamically contracted and expanded during tumor growth (Fig. 2g). Together, these data support the hypothesis that MMR homopolymer frameshifts act as a stochastic ON/OFF switch, dynamically regulating MSH6 expression during MMRd tumor evolution, akin to bacterial

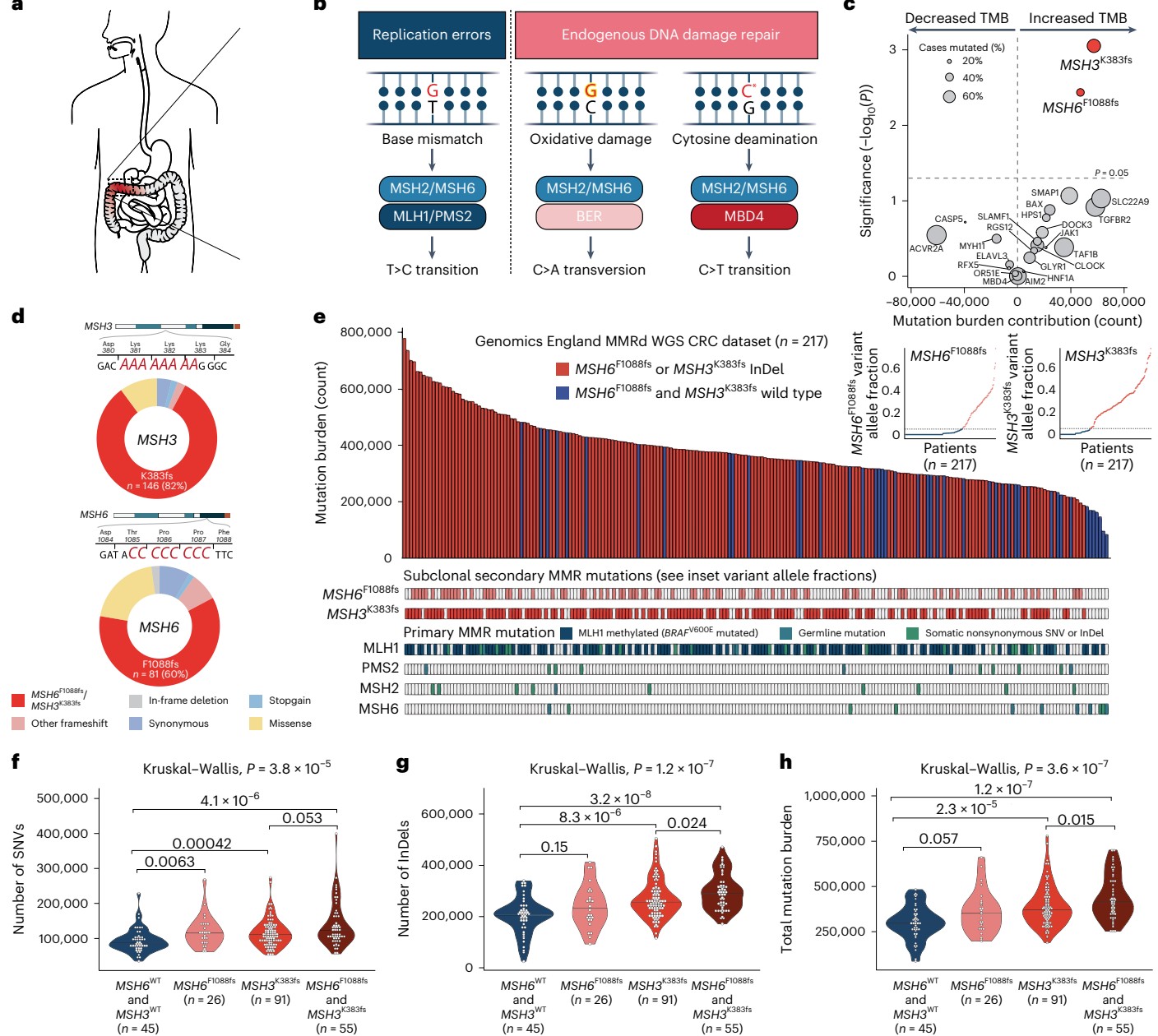

**Fig. 1 | Subclonal *MSH6*[F1088fs] and *MSH3*[K383fs] homopolymer frameshift mutations drive increased mutation burden in the MMRd CRC GEL WGS cohort. a**, MS-instable CRC. **b**, The MMR system safeguards genomic integrity by detecting and repairing replication-associated mismatches (left, blue). Recent studies indicate that MutS also participates in the repair of endogenous mutational damage independent of MLH1 (right, pink). **c**, Volcano plot showing the relationship between MS frameshifts in individual genes and total mutation burden in multiple linear regression analysis. For each independent variable, the *P* value of a two-sided *t*-test is plotted as $-\log_{10}(P)$. Two-sided *F* statistic (accounting for multiple independent variables in the regression model)

$P = 4.2 \times 10^{-7}$. **d**, Pie charts showing mutation categories for *MSH3* (top) and *MSH6* (bottom). **e**, Cases with *MSH6*[F1088fs] and/or *MSH3*[K383fs] homopolymer frameshifts (in red), and cases without such mutations (in blue) ranked by mutation burden (*n* = 217). Clonal alterations in MMR genes *MLH1*, *PMS2*, *MSH2* and *MSH6*, as well as subclonal *MSH6*[F1088fs] and *MSH3*[K383fs] frameshift status, are indicated below. Insets show *MSH6*[F1088fs] and *MSH3*[K383fs] mutation variant allele fraction. Extended Data Fig. 1a–c shows analysis restricted to *BRAF*[V600E] tumors. **f**–**h**, Number of SNV (**f**), number of InDel (**g**) and total mutation burden (**h**) according to *MSH6*[F1088fs] and *MSH3*[K383fs] mutation status. Median values are represented by horizontal black lines.

contingency loci (Fig. 2h). We next set out to examine the functional genomic impact of subclonal MMR homopolymer length fluctuations during MMRd tumor evolution.

### *MSH6* and *MSH3* homopolymer mutations cooperatively shift substitution bias

The spatially variegated pattern of *MSH6* labeling can be leveraged to interrogate the clonal landscape of MMRd tumors at monophyletic

resolution. Using LCM and whole-exome sequencing, we can directly assess mutation burden within individual MSH6-proficient and deficient tumor subclones and reconstruct highly resolved phylogenetic trees (Fig. 3a). We collected multiregion WES data after MSH6 IHC from 49 LCM regions (29 MSH6-proficient and 20 MSH6-deficient regions) from 22 MMRd tumors, including 11 MMRd tumors with heterogeneous MSH6 loss, and a control group of 11 MMRd MSH6-proficient cancers. All cases came from stage II/stage III

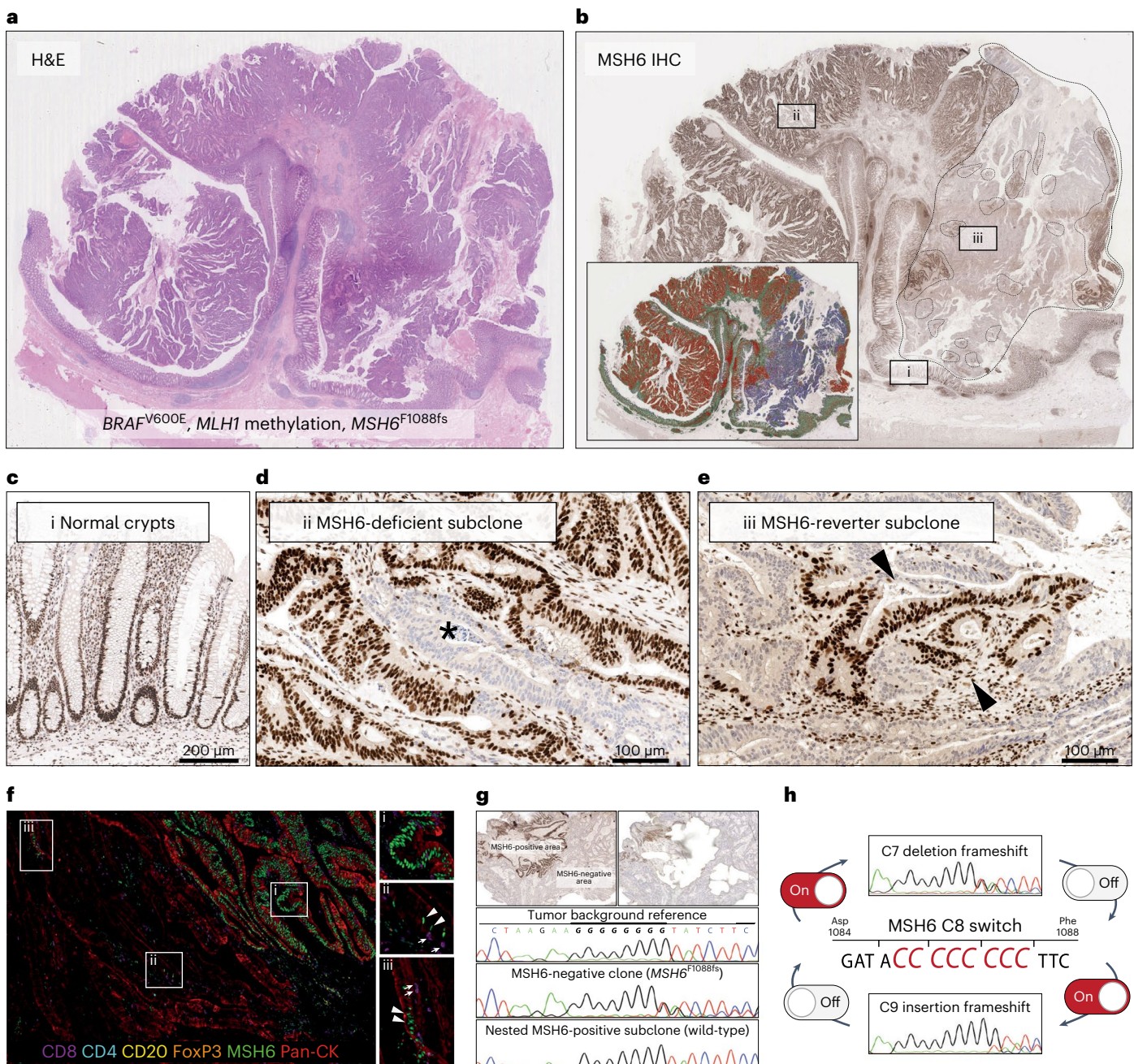

**Fig. 2 | Frameshift switching of the *MSH6* C8 coding homopolymer drives stochastic loss and restoration of MSH6 expression like a molecular ON/OFF switch. a,b,** Example hematoxylin and eosin (H&E) staining (**a**) and MSH6 IHC for polypoid cancer displaying subclonal loss of MSH6 expression (marked by dashed line, **b**). The tumor showed a *BRAF*^V600E mutation and *MLH1* methylation with loss of MLH1 and PMS2 labeling throughout the tumor (not shown). Boxes i, ii and iii are shown in **c–e**. (**c**) Region i, normal crypts show reference MSH6 labeling. (**d**) Region ii, small MSH6-deficient subclone. **e**, Region iii, nested MSH6-reverter subclone shows restoration of MSH6 labeling in tumor cells. **f**, Multiplex IHC (Methods) confirms scattered nested individual tumor cells (region ii) and small strips (iii) marked by pan-CK (red), which have restored MSH6 labeling (nuclear green) within MSH6-deficient tumor regions. MSH6-proficient region (box i) is shown for reference. **g**, LCM followed by Sanger sequencing of DNA from microdissected tumor regions confirms that frameshift switching of the C8 coding homopolymer underpins loss and subsequent restoration of MSH6 expression (Hg38 chr2: 47,803,501). **h**, Schematic representation showing that frameshift reversion mutations in the coding MSs of *MSH6* and *MSH3* allow them to act as a molecular ON/OFF switch for mutation rate. **a–g**, The workflow described was performed in *n* = 3 independent tumors.

surgical resection specimens with no prior exposure to systemic therapy. All cases showed clonal loss of either MLH1 and PMS2 or PMS2 alone (here collectively referred to as MLH1/PMS2 loss) and included patients with sporadic and germline MMR mutations. Clinicopathological patient characteristics are provided in Supplementary Table 3.

WES after LCM confirmed that tumor regions carrying frameshift mutations in either the *MSH6* or *MSH3* homopolymer had significantly increased SNV, InDel and overall mutation burden (Fig. 3b–d). As before, frameshift slippage of both *MSH6* and *MSH3* homopolymers had an additive effect. To account for the nonindependence of multiple sampling per patient and the confounding impact of age and

tumor purity on mutation burden, linear mixed-effect modeling was performed, controlling for these variables. The effect of $MSH6^{F1088fs}$ and $MSH3^{K383fs}$ on total mutation burden remained significant after accounting for the random effect of individual variation between tumors and fixed effects of age and tumor purity (analysis of variance (ANOVA), $P = 0.0296$; Supplementary Table 4). Large clinical cohorts have shown that TMB in patients with $MSH2$ and/or $MSH6$ mutations is generally greater than in patients with $MLH1$ and/or $PMS2$ mutations[14]. Our data reveal that this relationship is recapitulated between individual tumor regions.

Next, we compared mutation bias between regions. Data from a variety of model organisms (*Escherichia coli*, yeast and mice), as well as from patients constitutively lacking MMR, have revealed that mutation of $MutS_\alpha$ or $MutS_\beta$ drives a different mutation bias compared to mutations in $MutL_\alpha$ ($MLH1/PMS2$)[3,13,15,16]. The mutation bias provoked by MLH1/PMS2 loss is dominated by C>T and T>C transitions, while MutS loss drives a proportional decrease in T>C transitions in favor of C>T transitions and C>A transversions, along with a general increase in base substitutions over InDels. Analysis of the mutation spectra in our microdissected samples recapitulated a comparable stepwise decrease of T>C transitions and increase in C>T transitions and C>A transversions in the presence of $MSH6$ and $MSH3$ homopolymer frameshifts (Fig. 3e). In addition, while $MSH6$ or $MHS3$ frameshift drove an absolute increase in both base substitutions and InDels (Fig. 3b–d), the proportional increase in base substitutions was markedly greater (Fig. 3e and Extended Data Fig. 3a,b). Analysis of the trinucleotide context of mutations comparing samples with versus without incremental $MSH6^{F1088fs}$ (Fig. 3f), $MSH3^{K383fs}$ (Fig. 3g) or both (Fig. 3h) showed that the channels showing the largest proportional increase were GCG>GTG and CCT>CAT alongside other NCG>NTG channels and decreases in NTG>NCG channels. These shifts in mutation bias recapitulate previous analyses comparing the functional impact of MutL and $MutS_{\alpha/\beta}$ mutations across model systems and show that incremental MMR mutations fuel intratumor heterogeneity.

We next set out to investigate the functional impact of a shift in substitution bias by examining nonsynonymous mutations in a core set of genes contributing to neoantigen presentation (Methods) in samples with incremental MMR mutations. Subclones carrying $MSH6^{F1088fs}$ and $MSH3^{K383fs}$ homopolymer InDels showed a greater number of nonsynonymous antigen presentation machinery mutations (Fig. 3i,j). Notably, analysis of the trinucleotide context of these mutations recapitulates the mutation bias associated with incremental $MSH6$ and/or $MSH3$ homopolymer frameshift mutations (Fig. 3k). These results suggest that the shift in mutation bias functionally contributed to immune escape in subclones carrying $MSH6^{F1088fs}$ and $MSH3^{K383fs}$ homopolymer InDels.

Finally, we derived clone-specific in vivo mutation rates using a workflow around a recently developed computational method (Supplementary Note 2). We applied this workflow to samples from patient UCL_1014 (Fig. 3l) and found a significantly increased mutation rate $\mu$ in the MSH6-deficient sample ($\mu = 2.11^{-7}$ versus $\mu = 1.51^{-6}$ for the MH6-proficient and MH6-deficient samples, respectively; Fig. 3m–p), further corroborating that MMRd tumors are a mosaic of varying mutation rates and biases.

## Mutation burden and spectrum in patient-derived organoids (PDOs)

Next, we wanted to analyze the impact of incremental MMR mutations in MMRd tumors in an experimental context that allows temporal dissection of mutation accumulation. In normal tissues—where the background mutation rate is low—MMR conforms to a classic 'two-hit' paradigm, and loss of gene function requires biallelic mutation. However, our exome sequencing data on microdissected clonal patches suggested that in the context of $MLH1/PMS2$ (epi)mutation—where mutation supply is high—further increases in cellular mutation rate occur in a stepwise fashion with each additional allelic MMR mutation (Extended Data Fig. 3c)[17,18].

We set out to test these findings in a PDO model that allows the investigation of gene dosage effects in a well-controlled setting. We reasoned that subclonal incremental MMR mutations are common in the context of MMRd CRC (Fig. 1d,e) and thus multiregion-sampled a sporadic $MLH1^{meth}/BRAF^{V600E}$ (hereafter $MLH1^{-/-}$) MMRd CRC to obtain bulk PDO lines carrying incremental MMR mutations (Fig. 4a). Multiregion punch biopsies were briefly expanded and subcloned to establish clonal derivatives at first passage (Fig. 4b). Clonal lineages underwent 15× WGS to establish clonal mutation repertoire and lineage ancestry (Extended Data Fig. 4a). We retrieved five separate MMR genotypes ($MLH1^{-/-}$ from bulk Ca_1, $MLH1^{-/-}/MSH2^{+/-}$ from bulk Ca_2, $MLH1^{-/-}/MSH3^{+/-}$ from bulk Ca_3, $MLH1^{-/-}/MSH3^{-/-}$ from bulk Ca_4 and $MLH1^{-/-}/MSH3^{-/-}MSH6^{+/-}$ from bulk Ca_5; Fig. 4b,c). Homopolymer InDel mutations from WGS were verified by Sanger sequencing (Fig. 4d). We also carried out IHC on sections of the patient's tumor. This revealed, as expected, a large subclone that showed loss of MSH3 immunolabeling spatially corresponding with the $MLH1^{-/-}/MSH3^{-/-}$ PDO lineage, as well as scattered subclones showing loss of MSH6 immunolabeling (Extended Data Fig. 5a). The latter result suggests that further subclones carrying biallelic $MSH6$ mutations were present, but either these had not been sampled or failed to expand in PDO lines. Of note, the $MLH^{-/-}/MSH2^{+/-}$ lineage mentioned above carried a rare $MSH2^{A230fs}$ variant. This variant was encountered only once in the context of our bulk GEL WGS dataset ($n = 217$), also in a patient with truncal $BRAF^{V600E}$. Together, these PDOs thus provided a fortuitous 'full house' opportunity to test our predictions.

We set out to compare mutation accumulation between genotypes during extended PDO culture (Fig. 4e). To this end, each of four MMR genotypes ($MLH1^{-/-}$, $MLH1^{-/-}/MSH2^{+/-}$, $MLH1^{-/-}/MSH3^{+/-}$ and $MLH1^{-/-}/MSH3^{-/-}/MSH6^{+/-}$) was subcloned in six parent lineages (24 total), each of which was allowed to accumulate mutations during an 8-week mutation

**Fig. 3 | Subclonal $MSH6^{F1088fs}$ and $MSH3^{K383fs}$ homopolymer frameshift mutations drive intratumor mutation burden and mutation bias heterogeneity and provoke increased antigen presentation machinery mutations. a**, LCM strategy. Top, MSH6 IHC results with four target regions indicated—two regions show loss of MSH6 immunolabeling and two regions show retained MSH6 immunolabeling. Bottom, consecutive slide after LCM. **b–d**, Number of SNV (**b**), number of InDel (**c**) and total mutation burden (**d**) in LCM samples according to $MSH6^{F1088fs}$ and $MSH3^{K383fs}$ mutation status. **e**, SNV and InDel mutation bias in sample groups according to $MSH6^{F1088fs}$ and $MSH3^{K383fs}$ mutation status. **f–h**, Detailed 96-channel trinucleotide mutation spectra comparing substitution bias between (**f**) $MSH6^{wt}/MSH3^{wt}$ and $MSH6^{F1088fs}$ regions, (**g**) $MSH6^{wt}/MSH3^{wt}$ and $MSH3^{K383fs}$ regions and (**h**) $MSH6^{wt}/MSH3^{wt}$ and $MSH6^{F1088fs}$ plus $MSH3^{K383fs}$ regions, respectively. **i**, Heatmaps showing the number of mutations in antigen presentation machinery genes in samples according to $MSH6$ and $MSH3$ homopolymer InDel mutation status. **j**, Violin plot showing

increased number of HLA or antigen presentation machinery gene mutations in samples according to $MSH6$ and/or $MSH3$ homopolymer frameshift status. Two-sided Wilcoxon test. **k**, Trinucleotide context of antigen presentation machinery gene mutations. **l**, Overview case UCL_1014 with LCM samples as indicated (boxed regions). MSH6-proficient tumors are in blue, and MSH6-deficient areas are in red. Arrowheads in high-power images indicate minute reverter clones. **m**, Phylogenetic tree for tumor UCL_1014 annotated with HLA mutations identified in samples s111 and s112. **n**, MOBSTER subclonal deconvolution from diploid variants detected in LCM samples s111 (left, blue) and s112 (right, pink) from patient UCL_1014 shows a clonal population C1 with a subclonal tail of neutral variants. **o**, Cumulative frequency distribution of subclonal tail variants in s111 and s112. The point estimate of the normalized mutation rate $\mu$ in **p** is estimated from the slope of the cumulative frequency distribution. **p**, Bootstrapped 95% CI for the point estimate of the mutation rates in **o**.

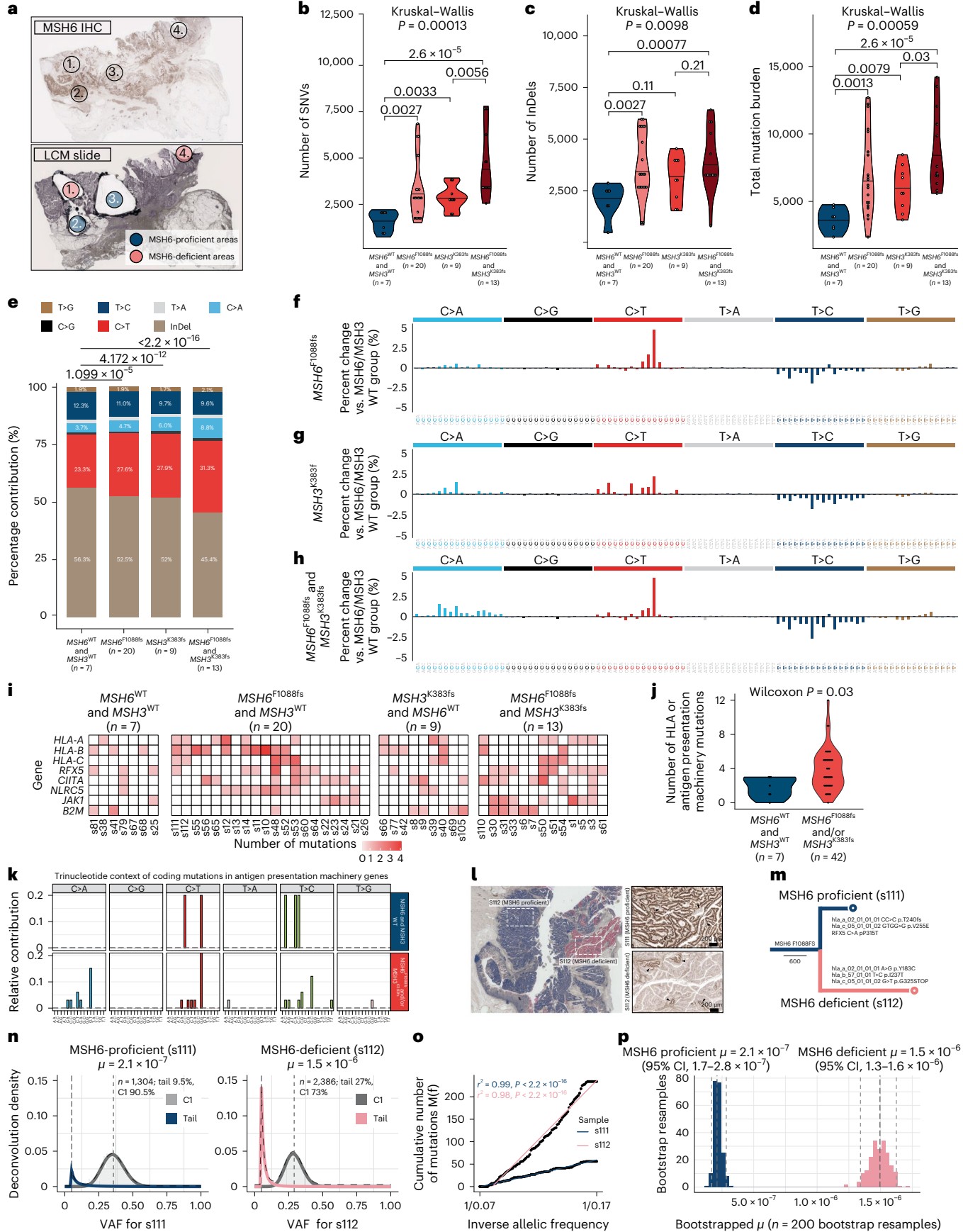

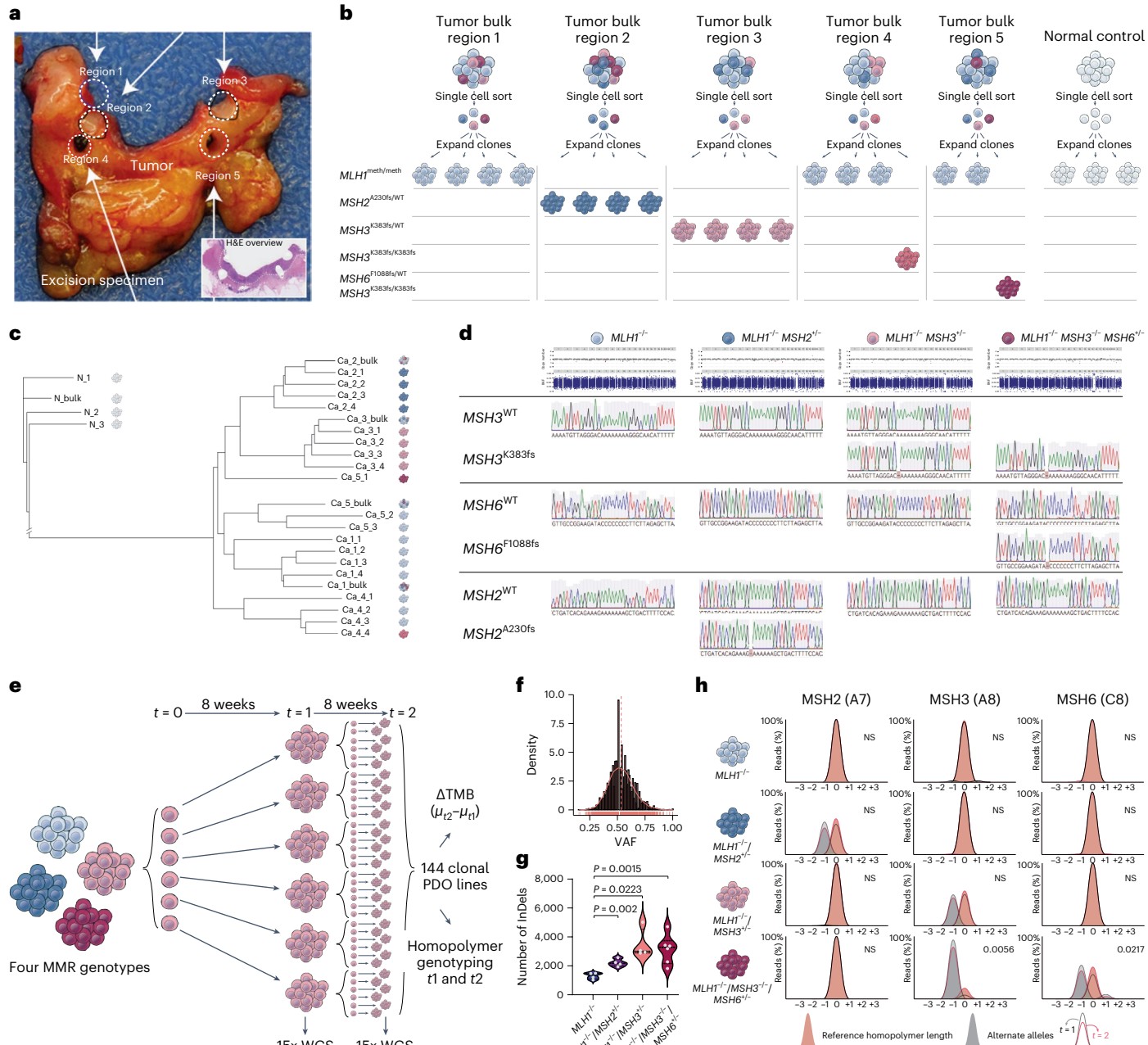

**Fig. 4 | Mutation burden and spectrum associated with incremental MMR mutations in PDOs. a**, Macropicture of sampled excision specimen, tumor regions one to five indicated. The inset shows the corresponding H&E-stained tumor section. **b**, Cartoon showing clonal PDO derivation strategy. Bulk samples were briefly expanded and subcloned at passage 1. Individual clonal organoids were expanded and genotyped at passage 4 (15× WGS). **c**, Neighbor-joining tree showing lineage relationships of bulk and clonal organoids. Tree labels refer to sample names, where N indicates normal tissue and Ca indicates cancer tissue. **d**, Homopolymer genotyping confirms the allelic status of MMR homopolymers as shown. **e**, Cartoon extended culture mutation accumulation experiment

(see 'Mutation burden and spectrum in patient-derived organoids (PDOs)' for details). **f**, Variant allele fraction density distribution shows symmetric binomial distribution around 0.5 confirming single cell origin. **g**, InDel burden across MMR PDO genotypes ($MLH1^{-/-}$ ($n = 4$), $MLH1^{-/-}/MSH2^{+/-}$ ($n = 4$), $MLH1^{-/-}/MSH3^{+/-}$ ($n = 4$) and $MLH1^{-/-}/MSH3^{-/-}/MSH6^{+/-}$ ($n = 6$)) after 8 weeks of extended culture (two-sided Welch's $t$-test). **h**, Homopolymer population diversity is shown as proportional VAF for each homopolymer length across genotypes analyzed at $t = 1$ (black) and $t = 2$ (pink) for the $MSH2$ A7, $MSH3$ A8 and $MSH6$ C8 homopolymers. Beige shows reference length and gray shows alternate alleles (two-tailed Fisher's exact test). NS, not significant.

accumulation period ($t = 1$). These clonal populations were split. Part of the population underwent 15× WGS as a proxy for the mutation burden of the cells-of-origin at $t = 0$. The remaining part was again subcloned in six individual daughter lineages (144 in total), where an additional 8-week culture period was used to amplify the cellular material to carry out 15× WGS at $t = 2$ as a proxy for mutation burden of the cells picked at $t = 1$. Mutation accumulation was assessed by comparing variant burden

at $t = 1$ and $t = 2$ of matched parent and daughter lineages. A minimum of four independent pairs of parent and daughter PDO lineages were assessed for each of the four MMR genotypes.

Individual PDO clones showed variant allele fractions around 0.5, confirming that these indeed derived from a single cell (Fig. 4f). We also confirmed that the loss of a single copy of $MSH3$ reduced PDO protein levels without eliminating antibody labeling (Extended Data Fig. 5b).

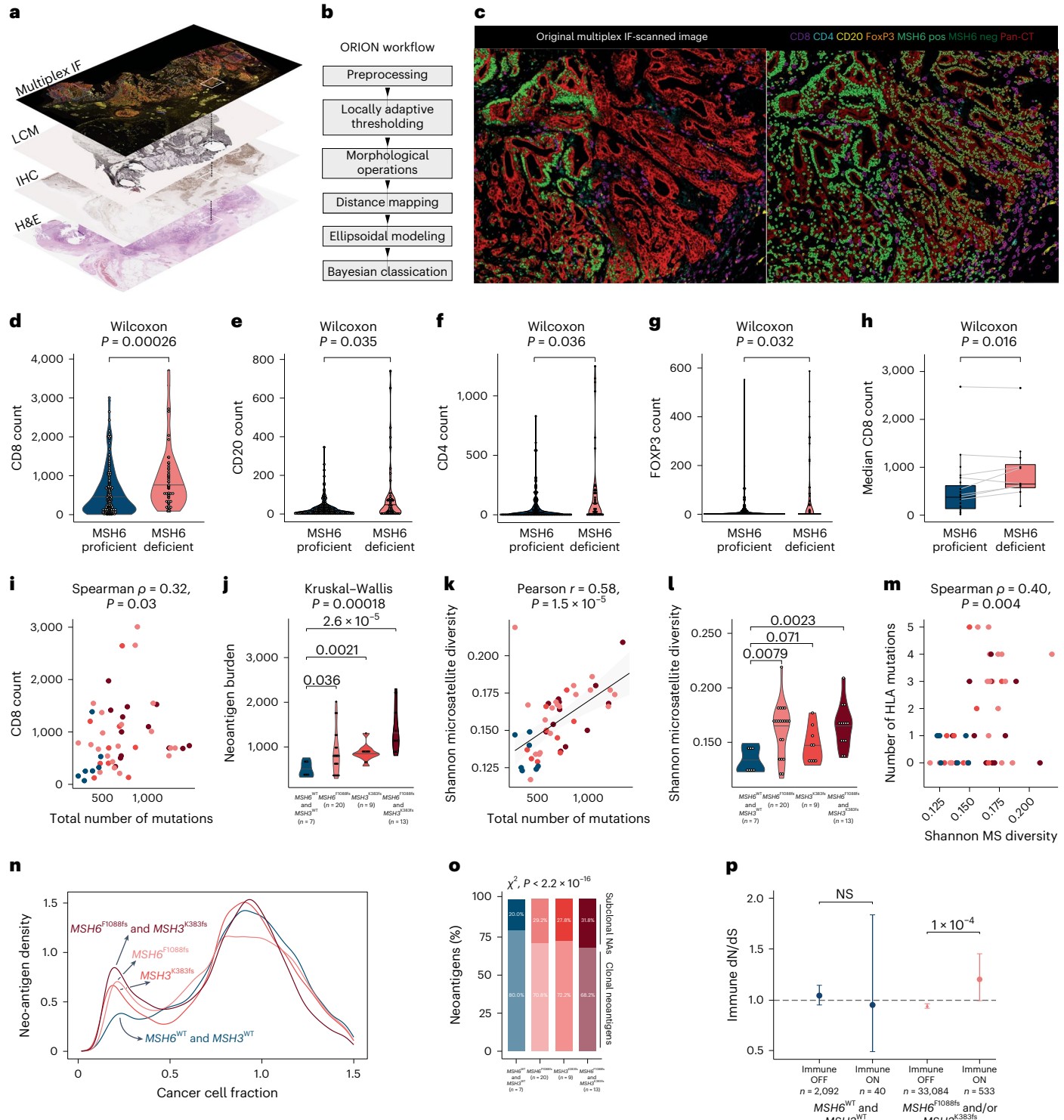

**Fig. 5 | *MSH6*^F1088fs^ and *MSH3*^K383fs^ homopolymer frameshift mutations accelerate clonal HLA diversity at the cost of increased neoantigen burden and immune cell infiltration. a**, Workflow for integrating MMRd clonal architecture, LCM sampling and MIF experiments. **b**, ORION workflow developed to investigate immune cell infiltration in MSH6-proficient and MSH6-deficient tumor subclones. **c**, Example segmented MIF image showing the interface between MSH6-proficient and MSH6-deficient subclones. MIF dataset consisted of *n* = 26 independent tumors. **d**–**g**, Infiltration levels of CD8-pos (**d**), CD20-pos (**e**), CD4-pos (**f**) and FOXP3-pos (**g**) immune cells within 100 μm radius of MSH6-proficient or MSH6-deficient tumor cells. **h**, Median CD8 infiltration levels in MSH6-proficient and MSH6-deficient subclones of individual tumors. **i**, CD8 count against total mutation burden according to *MSH6*^F1088fs^ and *MSH3*^K383fs^

mutation status. Color scheme as before. **j**, Neoantigen burden in samples according to *MSH6*^F1088fs^ and *MSH3*^K383fs^ mutation status. **k**, Shannon population diversity of length 8 homopolymers against total mutation burden according to *MSH6*^F1088fs^ and *MSH3*^K383fs^ mutation status. Color scheme as before. **l**, Shannon population diversity of length 8 homopolymers according to *MSH6*^F1088fs^ and *MSH3*^K383fs^ mutation status. **m**, Frequency of HLA class I mutations per sample according to *MSH6*^F1088fs^ and *MSH3*^K383fs^ mutation status (Polysolver package). **n**, Density plot showing fraction of neoantigens according to CCF in samples grouped according to *MSH6*^F1088fs^ and *MSH3*^K383fs^ status. **o**, Percentage of clonal versus subclonal neoantigens in samples grouped according to *MSH6*^F1088fs^ and *MSH3*^K383fs^ status. **p**, Immune dN/dS scores according to *MSH6*^F1088fs^ and *MSH3*^K383fs^ mutation status.

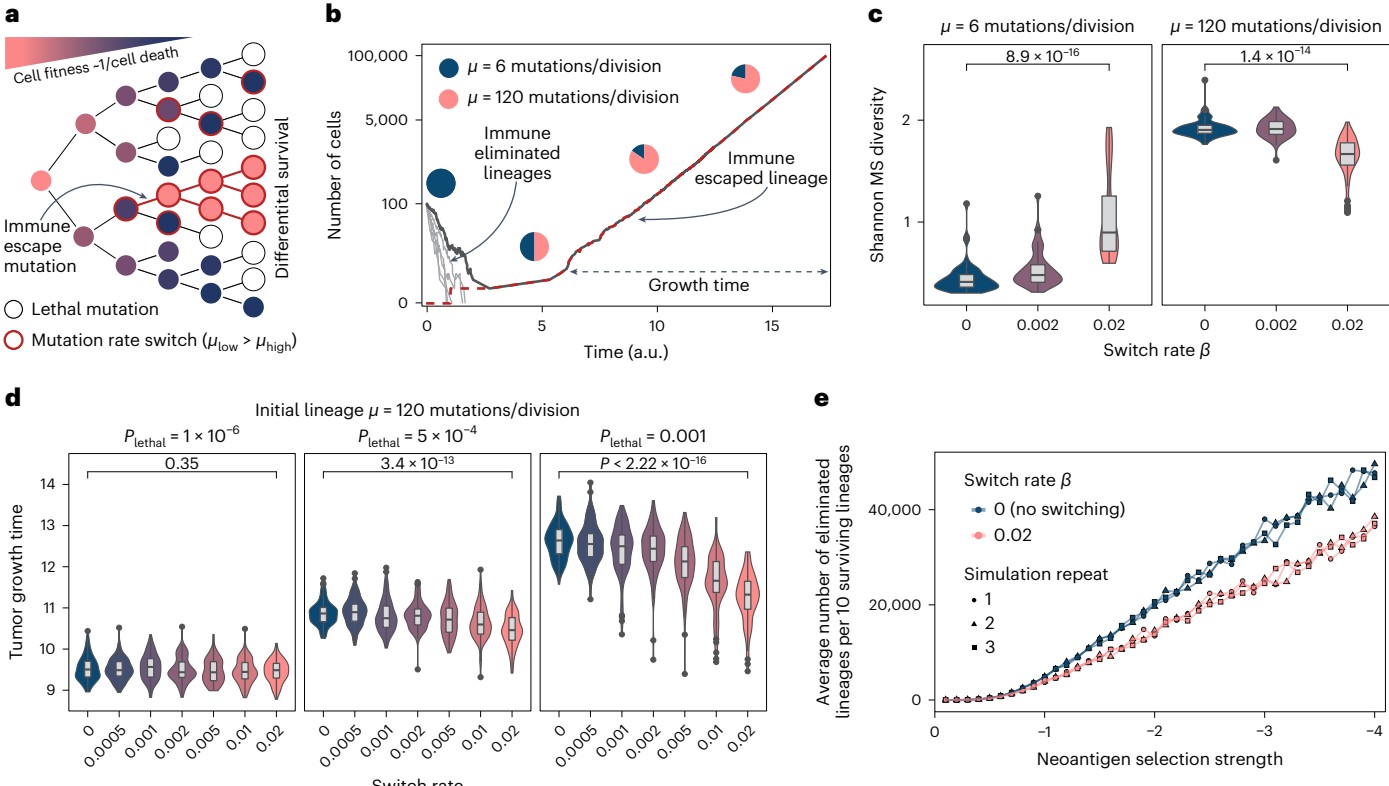

**Fig. 6 | Mathematical model of the effect of stochastic mutation rate switching on tumor growth. a**, The model captures early tumor growth (from 100 to 100,000 cells) using a stochastic birth–death process. Tumor cells can either die or proliferate according to their overall fitness and accumulate new mutations during cell division, which may affect their fitness value. Fitness is codetermined by the lineage-specific burden of stochastically accumulated neoantigens and the prevailing immune selection. Cells can either be in a basal MMRd hypermutated state or in a higher mutation rate regime. The probability of switching between $\mu_{basal}$ and $\mu_{high}$ is given by the switch rate $\beta$, where $\beta = 0$ corresponds to mutation rates that remain constant and $\beta = 0.01$ represents frequent switching to or from the higher mutation rate regime. The shade and outline color of each circle (cell) represent that cell's fitness and mutation rate, respectively (for further details of the model and parameter choice, see Methods). **b**, Six simulated tumor growth trajectories. Five eliminated lineages are indicated in light gray, one surviving lineage in dark gray with the number of immune-escaped cells within the tumor shown in red (overlapping the dark gray curve). Pie charts indicate the proportion of tumor cells with basal (blue) and higher (pink) mutation rates. **c**, Shannon diversity of an MS locus in simulated tumors ($n = 50$) with varying starting mutation rate and switching rate. **d**, Tumor growth time (in arbitrary units) between establishing immune escape and reaching detectable size (the model in **b**), computed from 100 simulated tumors with starting mutation rate $\mu = 120$ mutations/division at increasing lethal mutation frequency (left to right) and mutation rate switching rate (x axis). The P value of a two-sided Wilcoxon test comparing $\beta = 0$ and $\beta = 0.02$ is reported on top of each panel in **c** and **d**. **e**, Average (over 50 replicates) number of eliminated lineages per ten surviving lineages as a function of selection strength, in tumors with no (= 0, blue) and frequent (= 0.02, pink) mutation rate switching. Three independent repeats of simulation and averaging are indicated by circles, triangles and squares. Boxplots: horizontal black line represents median. Lower and upper hinges represent first and third quartiles. Lower and upper whiskers extend to values up to 1.5× interquartile range from the hinge. Outlying points beyond the whisker are plotted individually.

We then evaluated mutation burden and found that each of the clonal PDO lines carrying additional MMR homopolymer mutations had accumulated a significantly greater number of InDel mutations over the 8-week mutation accumulation period compared to the *MLH1*$^{-/-}$ reference (Fig. 4g and Extended Data Fig. 5c). These time course analyses corroborate our LCM WES data and show that in the context of high mutation supply due to MLH1 loss, a single additional MMR hit further increases cellular mutation rate.

Finally, we evaluated homopolymer evolution to test clonal genotypes and population drift over time. Exploiting the variant allele read counts in the WGS data, we found little deviation from reference length for nonmutated MMR homopolymers (MSH6 C8, *MSH3* A8 and *MSH2* A8) within the 8-week timeframe of this experiment, regardless of PDO genotype (Fig. 4h). By contrast, loci showing MMR homopolymer −1 deletion frameshifts at the start of the time course tended to return to wild-type (WT) sequence, thereby restoring gene function in part of the population (Fig. 4h). These temporal insights into MMR homopolymer sequence evolution show their intrinsic hypermutability and reversible nature. Moreover, when isolated from the tumor microenvironment

and immune selection, PDOs appear to experience a cell-intrinsic fitness cost with incremental MMR mutations, presumably due to mounting deleterious mutation burden. Together these data confirm that each incremental MMR homopolymer mutation increases clonal mutation rates, consistent with strong epistatic gene dosage effects in the context of high mutation supply.

## Incremental MMR mutations drive increased clonal diversity at a competing fitness cost

We next set out to analyze the microenvironmental impact of secondary *MSH6* and *MSH3* homopolymer frameshift mutations. We leveraged our spatially deconvoluted LCM analysis by carrying out multiplex IHC labeling of key immune cell populations on serial sections to our LCM slides (Fig. 5a). We labeled CD8, CD4, FoxP3 (Forkhead box P3), CD20, pan-CK, MSH6 and DAPI to trace immune cell infiltration and analyzed 194 regions across 26 tumors. Accurate automated immune cell quantification in tumor cell regions is complicated by extensive overlapping of nuclei in standard tissue sections. In addition, normal immune cell populations express MSH6, necessitating strict separation

of tumor cells from MSH6-positive immune cells. We developed a dedicated FluORescence cell segmentatION (ORION) workflow based on ellipsoidal modeling to accurately quantify immune cell infiltration both within and between MSH6-proficient and MSH6-deficient tumor regions (Fig. 5b,c, Extended Data Fig. 6a–k and Methods). We benchmarked our workflow against eight state-of-the-art cell segmentation tools (Supplementary Table 6 and Methods).

Stratifying regions by MSH6 labeling revealed a clear increase in infiltrating CD8-positive cytotoxic T cells (Fig. 5d), CD20-positive B cells (Fig. 5e), CD4-positive T cells (Fig. 5f) and FoxP3-positive $T_{reg}$ cells (Fig. 5g) in MSH6-deficient tumor regions. We then directly compared MSH6-proficient and deficient regions within individual tumors and again found increased CD8 infiltration in MSH6-deficient regions (Fig. 5h).

Next, we combined our clonally resolved WES analyses with matched immune cell infiltration data and found that CD8 infiltration correlated with mutation burden (Fig. 5i). We then analyzed neoantigen burden again stratifying by $MSH6^{F1088fs}$ and $MSH3^{K383fs}$ mutation status and observed a clear stepwise increase in neoantigen burden across these groups (Fig. 5j). These data corroborate our immune cell infiltration data comparing MSH6-proficient and deficient regions (Fig. 5d–h) and suggest that subclonal neoantigens drive intratumor immune cell infiltration in MMRd CRC.

We hypothesized that the drawbacks of increased neoantigen burden and immune cell infiltration associated with $MSH6^{F1088fs}$ and $MSH3^{K383fs}$ might represent an evolutionary trade-off with the benefits conferred by greater subclonal genetic diversity, ultimately driving clonal selection and immune escape. We therefore queried the clonal diversity of our samples using exonic homopolymers as polymorphic lineage tags (Extended Data Fig. 7a–g). We first examined the interaction between total mutation number and Shannon diversity at coding homopolymers and found a clear positive correlation (Fig. 5k). We then confirmed an increase in Shannon diversity across samples stratified by $MSH6^{F1088fs}$ and $MSH3^{K383fs}$ status (Fig. 5l). Finally, we analyzed sample-specific HLA mutation status using Polysolver and found that broader clonal diversity correlated with a greater number of mutated HLA alleles (Fig. 5m). This result indicates that subclones carrying incremental MMR frameshift mutations are characterized by greater clonal diversity at key immune escape loci, which provides a substrate for immune selection.

We then examined subclonal neoantigen complexity. Recent data from preclinical models[19] and patient cohorts[20] suggest that increased subclonal neoantigen complexity can drive early immune exhaustion and lack of immune checkpoint inhibitor treatment response. We find that subclones carrying either $MSH6^{F1088fs}$ or $MSH3^{K383fs}$ or both InDels show a greater proportion of subclonal predicted neoantigens (Fig. 5n,o). MMRd tumors typically demonstrate increased numbers of InDel neoantigens, some of which may escape nonsense-mediated decay (NMD) and provoke immune recognition[21]. We therefore filtered our subclonal neoantigen InDel calls to identify neoantigen InDels within NMD escape locations (Methods).

Tumor regions with $MSH6^{F1088fs}$ and/or $MSH3^{K383fs}$ were found to have significantly higher numbers of predicted NMD escape neoantigens (Kruskal–Wallis test, $P = 0.0044$; Extended Data Fig. 7h). A subset of these have been experimentally validated to elicit strong CD8 T cell responses in healthy controls and patients[22,23] (Supplementary Table 9 and Extended Data Fig. 7i).

We sought to test whether immune selection favors incremental MMR homopolymer InDels by using immune dN/dS (the ratio of nonsynonymous to synonymous substitutions) analysis. Immune dN/dS measures the ratio of nonsynonymous to synonymous mutations at genomic loci that are exposed to the immune system (immune ON) compared to neutral expectation[24]. This analysis found that tumor regions with $MSH6^{F1088fs}$ or $MSH3^{K383fs}$ InDels showed significant enrichment of nonsynonymous mutations predicted to be exposed to the immune system, while regions without $MSH6^{F1088fs}$ or $MSH3^{K383fs}$ showed no such enrichment (Fig. 5p and Supplementary Note 3). This supports the subclonal selection of MMR homopolymer InDel mutations through linked immune escape variants.

## Adaptive hypermutability accelerates immune escape

Our data indicate that the hypermutable homopolymer sites in MSH6 and MSH3 act as cryptic genetic switches that are unmasked by MSI. The dynamic relationship between hypermutability at these key loci and immune adaptation in MMRd CRC in the face of increasing mutational load is complex. To understand the dynamic impact of mutation rate switching, we extended our previous modeling work[7] and created a model of growing cancer that incorporates mutation rate switching, immune selection, immune escape and deleterious lethal mutations (Fig. 6a,b).

First, we evaluated how mutation rate switching influences population diversity. We simulated the alterations of a single MS locus throughout tumor growth and computed the overall Shannon diversity of this locus for each surviving detectable tumor. We found that higher mutation rates are associated with greater population diversity (Fig. 6c), corroborating observations within our clinical cohort (compare Fig. 5l). Moreover, increased mutation rate switching either exacerbated or dampened population diversity, depending on the mutation rate of the founding cell population (Fig. 6c).

Following the immune escape, tumor clones grow unimpeded by the immunogenicity of rapidly accumulating mutations that characterize hypermutated tumors. However, deleterious mutations, represented in our model by lethal mutations, can still drastically decrease overall fitness (compared with our observations in the PDO model). Because cells experiencing the higher mutation rate regime incur such disadvantageous mutations more frequently, we hypothesized that switching back down to a lower mutation rate is associated with a growth advantage for tumors initiated from these cells. We defined growth time as the time between complete immune escape and the tumor reaching clinically detectable size (Fig. 6b). Notably, we find that higher switching rates lead to a significantly faster tumor development, and this effect becomes more pronounced

**Fig. 7 | Phylogenetic trees reveal *MSH6* homopolymer frameshift reversion events. a**, Cartoon showing workflow for phylogenetic reconstruction. Multiregion whole-exome sequencing data from MSH6-proficient and MSH6-deficient microdissected patches ($n = 22$ tumors, between two and six patches per patient) are used to generate a binary SNV matrix to infer phylogenies using the maximum parsimony method (PAUP package). Scale bars indicate branch length and evolutionary distance expressed as substitution burden. The read length distribution of the *MSH6* and *MSH3* homopolymers is plotted separately (MSIsensor package) and compared against the reference MSH6 and MSH3 IHC of the input sample for verification. Finally, MSH6 labeling is overlaid on the phylogeny to reconstruct homopolymer evolution. **b–d**, Maximum parsimony phylogenetic trees were reconstructed using SNV mutation data with cases UCL_1016 (**b**), UCL_1002 (**c**) and UCL_1018 (**d**) displayed. Branch length is proportional to the number of mutations. Inset shows clinicopathological characteristics. Branches are colored according to MSH6 IHC labeling, with blue indicating MSH6-proficient and pink indicating MSH6-deficient lineages. High-power photomicrographs show MSH6, MSH3 and MSH2 labeling as indicated. Dashed lines indicate the border between proficient and deficient labeling; asterisk indicates the absence of labeling throughout; arrowheads indicate small reverter clones throughout. Trees are labeled with pertinent immune escape mutations. MS length distribution shows MSIsensor output where peak height is proportional to allelic frequency, with beige indicating reference C8 or A8 length, dark gray indicating expanded or contracted alleles and red indicating +3 frameshift. Phylogenetic trees were generated for all $n = 10$ tumors (Extended Data Fig. 10). Yo, year old.

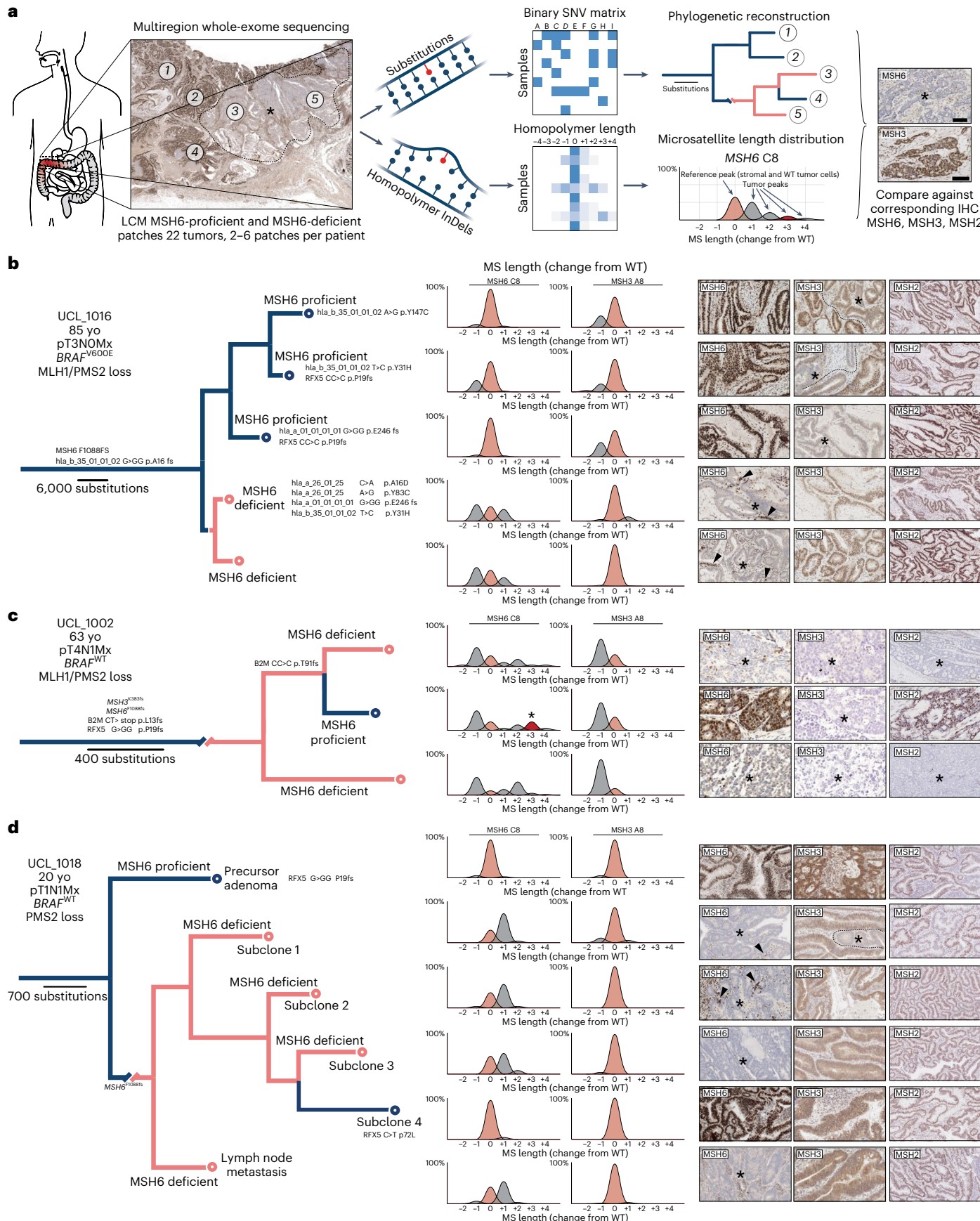

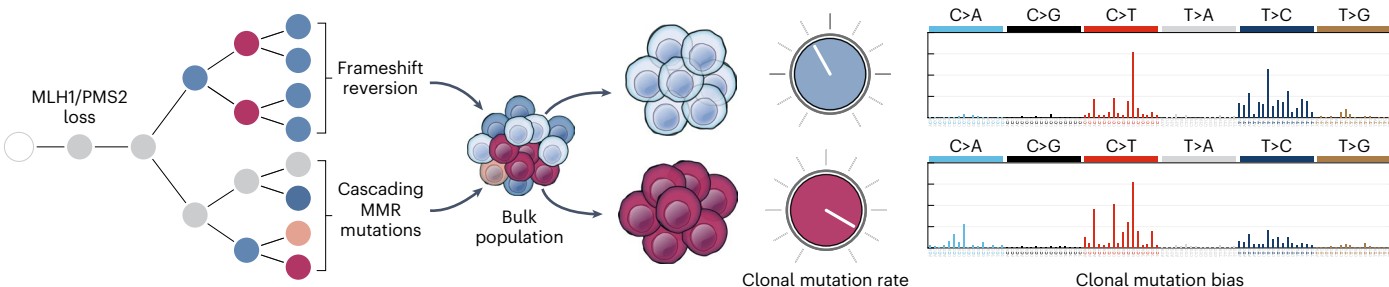

**Fig. 8 | Model illustrating genomic evolutionary trajectories to immune escape in MMRd cancer.** Incremental MMR mutations diversify cellular mutation rate and redirect mutation bias, which expands accessible genotype space and increases population diversity, allowing natural selection to pick immune-adapted variants.

as the likelihood of deleterious mutations increases (Fig. 6d). Switching had no beneficial impact when immune selection was absent, as in this case lineages could reach detectable size without developing immune escape (Fig. 6e).

In sum, our simulations suggest that mutation rate switching drives greater net lineage survival, which becomes more favorable as immune selection grows more stringent (Supplementary Note 4). These data are in line with our LCM/WES and PDO data. Balancing negative selection forces of deleterious mutation and immune extinction in this way suggests that evolving MMRd lineages in vivo might undergo cycles of switching to and from an increased mutation rate regime to shape immune adaptation.

### Tumor phylogenies reveal frameshift reversion following immune escape

We wanted to explore the evolution of immune adaptation in individual MMRd tumors by harnessing the phylogenetic information contained within our multiregion WES data. We first established maximum parsimony phylogenetic trees from our multiregion SNV sequencing data (experimental workflow in Fig. 7a). We then evaluated immune escape alterations by joint inspection of tree topology and MSH6/MSH3 immunolabeling. As an internal control for immunolabeling, we plotted *MSH6* and *MSH3* homopolymer length traces from sequencing data using the MSIsensor package (Extended Data Fig. 9a,b). These plots show a WT reference allele (0 peak, in beige), representing WT tumor alleles plus stromal admixture, and several contracted or expanded tumor alleles depending on *MSH6/MSH3* homopolymer mutation within a sample on either side of the reference allele count (Fig. 7a, workflow).

Phylogenetic analysis confirmed ongoing immune adaptation in growing MMRd tumors with many private immune escape mutations confined to tumor subclones (Fig. 7b and Extended Data Fig. 10). As before, MSH6-deficient subclones carrying immune escape mutations frequently contained scattered nested subclones that had spontaneously restored MSH6 expression (Fig. 7b (arrowheads) and Extended Data Fig. 10), suggesting ongoing evolution of cellular mutability concurrent with subclonal immune adaptation. Our cohort represents a cross-sectional analysis of treatment-naive MMRd tumors, suggesting that MMRd tumors explore multiple immune escape trajectories at time of diagnosis.

We then evaluated the temporal evolution of incremental MMR mutations from these phylogenetic trees. This analysis provided direct in vivo support for *MSH6* homopolymer frameshift switching during tumor evolution. Figure 7c depicts a phylogeny consisting of three terminal branches, two of which derive from MSH6-deficient patches, while the third derives from an MSH6-proficient patch. All patches revealed loss of MSH3 labeling. MSH2 labeling was lost in regions that had lost both MSH6 and MSH3, while it was restored in the MSH6-proficient patch, indicating that simultaneous loss of MSH6 and MSH3 expression leads to complete loss of MSH2 labeling (Extended Data Fig. 9c). Phylogenetic ordering revealed that the MSH6-proficient

lineage derived from the MSH6-deficient clade, providing formal phylogenetic proof of frameshift reversal. Inspection of the *MSH6* homopolymer length distribution showed that the MSH6-proficient lineage carried a +3 insertion (Fig. 7c, asterisk), demonstrating unequivocally that the *MSH6* homopolymer had undergone sequential nucleotide insertion until the reading frame was restored, after which the lineage clonally expanded. Notably, this reverter lineage carried a *B2M* mutation, providing a potential explanation for its selection and clonal expansion. In a second example (Fig. 7d and Extended Data Fig. 9d), phylogenetic ordering again revealed an MSH6-proficient branch that derived from an MSH6-deficient clade. The clade showing homopolymer reversion in this case carried a nonsynonymous C>T mutation in the HLA class II regulatory protein RFX5, which has been classified as pathogenic.

Taken together, these phylogenetic data corroborate our PDO data and model simulations and show that MMRd tumors exploit the inherent hypermutability of the *MSH6* and *MSH3* homopolymer coding sequences to explore diverse immune escape solutions and navigate their immune selection landscape.

## Discussion

The distribution of somatic mutations across cancer genomes is nonuniform and shaped by a complex interplay of factors including replication timing, chromatin compaction and three-dimensional genome organization[25,26]. Here we visualize the spatial clonal architecture of evolving MMRd tumors to allow joint analysis of individual tumor subclones and the immune microenvironment at clonal resolution. We find that subclonal MMRd lineages adapt to immune selection by repurposing hypermutable mononucleotide repeats in the *MSH6* and *MSH3* coding sequences as ON/OFF switches to drive stochastic loss of expression at these loci. Our bulk sample, LCM, PDO and molecular evolution data show that incremental MMR mutations increase cellular mutation rate, redirect mutation bias and increase clonal diversity. This expands accessible genotype space and increases population diversity, allowing natural selection to pick immune-adapted variants (Fig. 8).

Drug selection experiments in MS-stable CRC[27] and nonsmall-cell lung cancer[28,29] have provided evidence for adaptive mutability in response to targeted treatment. Our work adds to this evidence base and reveals that adaptive mutability drives lineage survival during MMRd cancer evolution prior to clinical intervention. Understanding the evolutionary pathways to immune escape in MMRd tumors may allow us to forecast individual responses to immune checkpoint inhibition[30].

## Online content

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

Hamzeh Kayhanian[1,18], William Cross [1,2,18], Suzanne E. M. van der Horst[3,18], Panagiotis Barmpoutis [1,4,18], Eszter Lakatos[5,18], Giulio Caravagna [6], Luis Zapata[7], Arne Van Hoeck [3], Sjors Middelkamp[8], Kevin Litchfield[1], Christopher Steele[1], William Waddingham[1], Dominic Patel [1], Salvatore Milite [6], Chen Jin [4], Ann-Marie Baker [7], Daniel C. Alexander [4], Khurum Khan[9], Daniel Hochhauser[1,9], Marco Novelli[1,10], Benjamin Werner [11], Ruben van Boxtel [3,8], Joris H. Hageman[3], Julian R. Buissant des Amorie[3], Josep Linares[12], Marjolijn J. L. Ligtenberg [13,14], Iris D. Nagtegaal [13], Miangela M. Laclé[15], Leon M. G. Moons[16], Lodewijk A. A. Brosens [15], Nischalan Pillay[1], Andrea Sottoriva [7,17], Trevor A. Graham [7,11], Manuel Rodriguez-Justo [1,10], Kai-Keen Shiu[1,9], Hugo J. G. Snippert [3] ✉ & Marnix Jansen [1,10] ✉

[1]UCL Cancer Institute, University College London, London, UK. [2]Cancer Mechanisms and Biomarker Discovery Group, School of Life Sciences, University of Westminster, London, UK. [3]Oncode Institute, Center for Molecular Medicine, University Medical Center Utrecht, Utrecht, The Netherlands. [4]UCL Centre for Medical Image Computing, Department of Computer Science, University College London, London, UK. [5]Department of Mathematical Sciences, Chalmers University of Technology and University of Gothenburg, Gothenburg, Sweden. [6]Department of Mathematics, Informatics and Geosciences, University of Trieste, Trieste, Italy. [7]Centre for Evolution and Cancer, The Institute of Cancer Research, London, UK. [8]Princess Máxima Center for Pediatric Oncology, Utrecht, The Netherlands. [9]Department of Oncology, UCL Cancer Institute, University College London, London, UK. [10]Department of Pathology, University College London Hospital, London, UK. [11]Centre for Cancer Genomics and Computational Biology, Barts Cancer Institute, Queen Mary University of London, London, UK. [12]HSL-AD, London, UK. [13]Department of Pathology, Radboud University Medical Center, Nijmegen, The Netherlands. [14]Department of Human Genetics, Radboud University Medical Center, Nijmegen, The Netherlands. [15]Department of Pathology, University Medical Center Utrecht, Utrecht University, Utrecht, The Netherlands. [16]Department of Gastroenterology and Hepatology, University Medical Center Utrecht, Utrecht, The Netherlands. [17]Computational Biology Research Centre, Human Technopole, Milan, Italy. [18]These authors contributed equally: Hamzeh Kayhanian, William Cross, Suzanne E. M. van der Horst, Panagiotis Barmpoutis, Eszter Lakatos. ✉e-mail: h.j.g.snippert@umcutrecht.nl; m.jansen@ucl.ac.uk

## Methods

### UCL CRC cohort

All samples were anonymized and processed according to protocols approved by the UCL/UCLH (University College London/ University College London Hospitals) Biobank of Health and Disease Ethical Review Committee (project reference: NC21.18). In line with UK regulations, the research was conducted with project-specific research ethics approval and samples de-identified to the research team, which permitted the research to be conducted without individual patient consent. The samples used in this project were archival material requested from the UCL research tissue biobank. The UCL research tissue bank is registered under REC reference 20/YH/0088 and IRAS (integrated research application system) project: 272816. The biobank was searched to identify MMRd CRCs diagnosed between 2014 and 2018. Of 546 cancers tested by IHC, 88 (16%) showed MMR protein loss. Available FFPE (formalin-fixed and paraffin-embedded) tumor blocks were retrieved, and sections were cut to perform MSH6 IHC using an established protocol. Antibody details and IHC conditions are provided in Supplementary Table 3.

After assessing for tissue quality, 11 ($n = 11/40$, 28%) tumors had subclonal loss of MSH6 in at least one tumor block. A stage- and age-matched cohort of 11 MLH1/PMS2 MMR-D tumors without immunohistochemical loss of MSH6 was also selected as the comparison group. For each tumor, a corresponding normal block from the resection margin was also retrieved. MSH6-labeled slides for each tumor were scanned using a slide scanner (Hamamatsu NanoZoomer).

**LCM.** Tumors with MSH6-deficient subclones ($n = 11$) and those in the MSH6-proficient comparison group ($n = 11$) were taken forward for LCM. Multiregion samples from more than one tumor block were taken where available. Because IHC labeling can affect DNA yield, a bespoke protocol was developed so that adjacent IHC-labeled sections were used to guide the microdissection of thicker sections on LCM membrane slides. Each tumor block was serially sectioned as follows: one 3-μm-thick section onto a glass slide, five 10-μm-thick sections onto polyethylene naphthalate membrane slides (Carl Zeiss AG) and one 3-μm-thick section onto a glass slide. The 3-μm-thick sections underwent IHC against MSH6 and were used to guide microdissection of the thicker sections in between. Membrane slides were pre-treated with 0.01% poly-L-lysine to improve tissue adherence. Mounted sections were baked in an oven at 50 °C for 4 h. The 10-μm-thick sections were deparaffinized and stained with hematoxylin as follows: xylene (10 min, two changes), 100% ethanol (1 min, two changes), 90% ethanol (1 min, one change), rinse in deionized water, Gill's hematoxylin (1 min, one change), rinse gently in running water, 90% ethanol (1 min, two changes), 100% ethanol (1 min, two changes), xylene (1 min, two changes). LCM was performed using the Palm MicroBeam microscope (Carl Zeiss AG). Selected MSH6-deficient and proficient tumor regions approximately 2–3 mm² in area were individually microdissected and collected in 500 μl AdhesiveCap tubes (Carl Zeiss AG). Tissue originating from the same location was pooled across serial sections and processed as one sample. Tissue from the resection margin normal mucosa was also microdissected and used as the germline sample. Each microdissected region was allocated a unique sample number, and the microdissected site was recorded on corresponding scanned slides for future reference.

**IHC.** IHC was performed using the Leica Bond autostainer (Leica Biosystems). Antibody details and conditions are provided in Supplementary Table 5.

**DNA extraction.** In total, 6 μl of proteinase K and 200 μl of lysis buffer (PerkinElmer) were added to microdissected tissue samples and incubated overnight at 56 °C followed by 1 h at 70 °C to reverse formaldehyde cross-links. DNA extraction was completed using the Chemagic

Prepito automated instrument (PerkinElmer), which uses a magnetic particle separation technique. Extracted DNA was quantified using a Qubit fluorometer (Thermo Fisher Scientific) as per the manufacturer's instructions.

**Sanger sequencing.** Validation of frameshift mutation in the C8 coding MS within *MSH6* was performed by PCR, followed by BigDye terminator Sanger sequencing. Oligonucleotides (forward primer TTTTAACAGATGTTTTACTGTGC and reverse primer TCATTAG GAATAAAATCATCTCC), Q5 polymerase master mix (New England Biolabs) and 10 ng of DNA were used in PCR reactions. PCR conditions were 35 cycles of denaturation at 95 °C for 30 s, followed by primer annealing at 60 °C for 1 min, followed by extension at 72 °C for 30 s.

**Sample preparation for whole-exome sequencing.** Acoustic fragmentation of DNA was performed using the Covaris E220 device. In total, 125 ng of sample DNA was inserted into snap-cap microtubes (Covaris) at a total volume of 50 μl. The Covaris device was used as per the manufacturer's guidelines. The following settings were used: duty factor = 10%, peak incident power ($W$) = 175, cycles per burst = 200 and time (seconds) = 300. Fragmented DNA samples were transferred to 1.5 ml Eppendorf tubes.

**FFPE repair.** FFPE repair was performed to minimize the impact of artefactual lesions due to formalin fixation[31] using a validated kit (New England Biolabs, M6630L) and following the manufacturer's protocol. Briefly, 48 μl of fragmented sample DNA was mixed with 3.5 μl of FFPE DNA repair buffer, 3.5 μl of end-prep buffer and 2 μl of FFPE DNA repair mix, and the mixture was incubated at 20 °C for 30 min.

**Library preparation.** Library preparation was performed using the NebNext Ultra II Kit (New England Biolabs) as per the manufacturer's protocol. Briefly, following end repair and A-tailing, adapter ligation was performed by adding 30 μl of ligation master mix, 1 μl of ligation enhancer and 2.5 μl of sequencing adapters, and the reaction mixture was incubated for 15 min at 20 °C. Adapters were diluted 10× as per the manufacturer's guidance. Magnetic bead clean-up of adapter-ligated libraries was performed by adding 87 μl (0.9×) of Ampure XP beads (Beckman Coulter) followed by ethanol washes and elution in 17 μl of 10 mM Tris–HCl. Next, 15 μl of the adapter-ligated library was amplified with ten cycles of PCR by adding 25 μl of Q5 master mix and 10 μl of indexing primers. For sample indexing, NEBNext Multiplex Oligos (E7335) was used, and the indexing primer used was recorded for each sample. Library fragment size was analyzed with a Tapestation device (Agilent Technologies) using high-sensitivity screen tape and also quantified using a Qubit fluorometer (Thermo Fisher Scientific).

**Exome capture.** Exome capture was performed following the manufacturer's protocol in the SeqCap EZ Kit (Roche Sequencing Solutions). In total, 250 ng of library samples from the previous step were pooled in groups of four to give a total mass of 1 μg. The multiplexed library pool was hybridized with SeqCap EZ Prime Exome probes for 16 h at 47 °C. Following hybridization, unbound probes were washed away, and the hybridized DNA was amplified with 14 cycles of PCR followed by 1× Ampure XP bead clean-up and eluted in 33 μl of 0.1× TE (Tris-EDTA) solution. The final captured amplified library was quantified by qPCR using the NEBNext Library Quant Kit for Illumina (New England Biolabs).

**Next-generation sequencing.** Samples were diluted to 2 nM and sequenced in batches of 12 on the NovaSeq instrument (Illumina) using an S1 flowcell with 100 bp paired-end reads as per the manufacturer's instructions.

**Whole-exome sequencing, aligment and variant calling pipeline.** FastQ sequencing files were aligned to the Hg19 reference genome using BWA-mem (version 0.7.7). Aligned sequencing files were converted to BAM files followed by sorting and indexing of reads using SAMtools. Picard Tools was used to mark duplicates, and GATK (version 2.8) was used for local InDel realignment. Picard Tools, GATK (version 2.8) and FastQC were used to produce quality control metrics. SAMtools mpileup (version 0.1.19) was used to locate nonreference positions in tumor and germline samples. Bases with a Phred score of less than 20 or reads with a mapping quality (MQ) of <20 were omitted. MuTect (version 1.1.4) was used to detect SNVs, and results were filtered according to the filter parameter PASS. An SNV was considered a true positive if the VAF was ≥5% and the number of reads in the tumor and germline at that position was ≥20. For InDels, only calls classed as high confidence by VarScan2 and Scalpel were kept to avoid the risk of caller-specific artifacts often observed with InDel calling. Variant annotation was performed using Annovar (version 2016Feb01).

**Purity, ploidy and copy number (CN) estimation.** The Sequenza package was used to derive CN estimates for each sample. We obtained tumor purity and ploidy estimates using the probabilistic parameter search as suggested in the package manual. We included a quality control step in line with a recent publication[32] which examines the somatic SNV allele frequency distribution present in proposed regions of copy change. We found that all inferred copy states matched the expected allele frequency shifts, suggesting our ploidy estimates were correct (allele frequency shifts include peaks at 0.33 and 0.67 in trisomy regions and 0.5 and 1 in regions of copy-neutral LOH (loss of heterozygosity), following correction by tumor content).

**Calculation of cancer cell fraction (CCF).** For each SNV, the CCF was calculated using a previously described formula based on the VAF, tumor purity and allele-specific CN[33]. SNVs across all samples were pooled into four groups according to the *MSH3/MSH6* mutation status of samples. The distribution of CCF for SNVs and predicted neoantigens in each group was plotted as a density plot in R.

**Extraction of read length distribution of MSs.** The MSIsensor package (version 0.6) was run on all samples using tumor and matching normal BAM files as input. Default settings were adjusted to include a minimum homopolymer length of six for distribution analysis. Next, the SciRoKo package (version 3.4) was used to identify all length 6–11 homopolymers within the exome using a BED file of genomic coordinates for the human Hg19 exome. MSIsensor distribution files were next filtered for exonic length 6–11 homopolymers, and the resulting data were used for downstream analysis.

**Identification of *MSH6*[F1088] and *MSH3*[K383] frameshift mutations.** To accurately call frameshifts within *MSH6* and *MSH3* homopolymers, we used the consensus of calls made using MSIsensor (version 0.6) and the variant calling pipeline described above. The MSIsensor-derived MS read length distributions of *MSH6* and *MSH3* homopolymers were extracted. The percentage of reads at each length was calculated for both MSs and tabulated (Extended Data Fig. 10). To call a mutation, a minimum of 5% of reads was required to show instability, with a minimum of 50 reads present. Next, cases identified as mutated using this MSIsensor technique were checked against calls made using the Varscan/Scalpel pipeline described above. Any discrepancies were manually checked using Integrated Genomics Viewer (v2.3) software. Our experimental design ensured that we had sufficient power to account for the expected noise in NGS data at homopolymer sequences. The average sequencing depth at the *MSH3* and *MSH6* loci was 300× and 379×, respectively. The minimum VAF that we accepted (conservatively) as a putative variant in our data was 5%, while the minimum observed VAF was 1.3%. Taking the commonly reported mutation error rate

expectation of 0.01%[34] and minimum observed depth and mutation frequency (150× and 5%, respectively), the statistical power was 0.78. Taking average values (300× and 17%), the power climbs to 0.98. In the best-case example in our data (600× and 35%), the power is 0.99 (analyses performed in G*Power software).

**Shannon MS diversity.** The Shannon diversity index was calculated for all exonic length 6–11 MSs in each sample using the formula:

$$\text{Shannon diversity} = -\sum_{i=1}^{R} (p_i \ln (p_i)),$$

where $p_i$ is the proportion of total reads represented by the $i$th MS length and $R$ is the total number of read lengths present at an MS.

**Phylogenetic reconstruction.** For tumors with more than three samples sequenced, the maximum parsimony method was used to infer phylogenies from the SNV calls. We used the Paup package (http://phylosolutions.com/paup-test/) and parameters as described previously[35]. Briefly, SNV calls were converted into a binary matrix, where 0 equals absence and 1 equals the presence of a mutation, rows relate to a biopsy or the normal sample and columns relate to each variant. The following methods were used for phylogenetic reconstruction: (1) the root function was used to root each phylogeny to the normal sample; (2) the hsearch function was used to perform a heuristic search of available trees, and 1,000 of the shortest trees were output and examined; and (3) the bootstrap function was used to randomly resample the data 10,000 times with replacement, with the proportion of each branch instance reported. The most parsimonious tree was reported for each case, and in this data, there was only ever one best solution. The .tre files generated were viewed using FigTree software (http://tree.bio.ed.ac.uk/software/figtree/) and converted to PDF files.

For tumors with only two or three samples, sequenced parsimony trees cannot be produced. In these cases, the binary matrices were used to make simple inferences about clonality through shared mutation instances. Variants present in all samples were allocated as trunk mutations. For tumors with three biopsies, biopsy pairs with the most shared mutations were placed together on the same clade. Variants unique to each sample formed terminal branches (leaves).

**HLA typing and mutation calling.** The Polysolver package (version 4) was used to perform haplotyping and mutation calling for HLA-A, HLA-B and HLA-C alleles. Germline and tumor sequencing data were supplied in the form of BAM files.

**Mutations in antigen-processing machinery (APM) genes.** APM genes previously reported as undergoing mutation in MMRd cancers were identified from the literature[36]. This created a gene list consisting of *NLRC5*, *RFX5*, *TAP1*, *TAP2*, *CIITA* and *JAK1*. Coding mutations in these genes (frameshift, nonsynonymous SNVs or nonsense mutations) were retrieved from the annotated variant call files. Synonymous mutations were excluded.

**Neoantigen calling.** Neoantigens were predicted using an established pipeline (NeoPredPipe) using patient-specific HLA haplotypes and the NetMHCpan prediction tool[37].

**NMD and identification of experimentally validated neoantigens.** InDel mutations frequently cause premature termination codons (PTCs), which are a target for the NMD pathway, resulting in the degradation of putative neoantigen transcripts. NMD is known to operate less efficiently when PTCs are present in the last exon, penultimate exon within 50 bp of the 3′ exon junction or first exon within the first 200 bp of the coding sequence[21]. InDel variants were annotated using the ANNOVAR package to identify the exon position of the variant.

InDels were classified as follows: first exon within 200 bp of the coding sequence, first exon >200 bp from the start of the coding sequence, middle exon, penultimate exon ≤50 bp of the last exon junction complex or last exon. Neoantigens were labeled as predicted to escape NMD if located in the first exon within the first 200 nucleotides of the coding sequence, last or penultimate within 50 bp of the last exon junction complex[21].

Experimentally validated neoantigens were identified from the literature, from studies where neoantigens had been identified as being recurrently observed in MSI tumors and able to elicit strong CD8 T cell responses in healthy controls and patients[22,23]. These validated neoantigens are listed in Supplementary Table 9. We then searched our neoantigen data to identify the presence of these validated neoantigens in the UCL CRC WES cohort. We found that these validated immunogenic antigens were frequently observed in our cohort, with on average 3.7 and 2.7 per sample in cases with $MSH6^{F1088fs}$ and/or $MSH3^{K383fs}$ versus $MSH6/MSH3$ WT samples, respectively, although this difference was not significant due to the small numbers (Wilcoxon $P$ = 0.39; Extended Data Fig. 7i). These findings support our main findings that increased neoantigen burden observed in the presence of $MSH6$ and $MSH3$ homopolymer frameshifts is validated with immunogenic potential.

**Linear mixed-effect model.** To account for the nonindependence of multiple samplings per patient, a linear mixed-effect model was created to assess the relationship between $MSH6/MSH3$ frameshift status and total mutation burden. Individual variation in mutation burden between tumors was defined as a random effect, and the presence of mutation in $MSH6$ and/or $MSH3$ MSs, age at diagnosis and tumor purity were defined as fixed effects. $P$ values were obtained by likelihood ratio tests of the full model with the effect of $MSH6/MSH3$ status against the null model without the effect of $MSH6/MSH3$ status. The model was created using the R package LME4 as follows:

lmer(MT_burden ~ MSH6_MSH3_status + age + tumor_purity + (1|tumor_ID). Full results of the linear mixed effects model are provided in Supplementary Table 4.

**Mutation signature analysis.** Analysis of mutation signatures was performed using the package Sigprofiler (version 3.1). SNV and InDel data were merged according to the $MSH6/MSH3$ mutation status of samples resulting in the following three groups: samples WT for both $MSH6$ and $MSH3$, samples with either the $MSH6^{F1088fs}$ or $MSH3^{K383fs}$ and samples with both $MSH6^{F1088fs}$ and $MSH3^{K383fs}$. Three de novo SBS (single base substitution) signatures were extracted, and their 96-channel trinucleotide context was plotted. The percentage contribution of each signature according to $MSH6/MSH3$ meta-groups was further plotted. A similar analysis was performed for InDel and double base mutations.

**Immune dN/dS.** Immune dN/dS, defined as the portion of the genome exposed to immune recognition, was calculated using SOPRANO[24] (the code is available at github.com/luisgls/SOPRANO). It estimates dN/dS values in a target region (ON-target) and in the rest of the proteome (OFF-target) using a trinucleotide context correction (SSB192). Here we have used genomic regions that translate to peptides that bind the HLA-A0201 allele as the target region (ON). Only genes with a median expression of more than one fragment per kilobase million (FPKM) were used according to the human expression atlas data (downloaded on 18 October 2018). The file used as the target region can be obtained from github.com/luisgls/SOPRANO.

**MOBSTER clonal deconvolution and mutation rate analysis**
To retrieve in vivo mutation rate estimates, we developed a simple pipeline around MOBSTER, a recently developed computational method that can perform tumor subclonal deconvolution by integrating population genetics and machine learning[38]. This method is able to retrieve an estimate of the tumor mutation rate ($\mu$) from the tail of neutral mutations. To run it, we first pooled somatic variants and absolute CN alterations (CNAs) generated as detailed above. We then used a computational method to map somatic SNVs on top of CN segments and assess the consistency between tumor purity, ploidy and CNAs. We restricted our analysis to SNVs and dropped InDels because the VAF estimates for SNVs are more reliable for assessing the quality of the calls and performing deconvolutions. All the samples we analyzed passed our quality check process.

We then assessed the overall percentage of CNA segments that span the tumor genome and considered the copy state of each segment. This confirmed that the largest chunk of the tumor genome is in a heterozygous diploid state, with a single copy of the major and minor alleles, which is expected from CRCs with MSI[39]. For this reason, we retained only SNVs mapping to diploid segments, which harbor less noise compared to mutations that map to more complex tumor karyotypes. With pooled diploid SNVs, we proceeded to run tumor subclonal deconvolution using raw VAFs and MOBSTER. The tool was run to search for tumors with up to two subclones, with an optional neutral tail; model selection for the number of clonal populations ($k > 0$) and the tail was performed using the routines available in MOBSTER. MOBSTER could estimate the different mixtures of cancer subpopulations in each one of the bulk samples, as well as the neutral tail of somatic SNVs that accrue inside each of the subclonal expansions, if present. Across data for cancer UCL_1014, we observed monoclonal populations ($k = 1$) with a neutral tail, therefore concluding that these data lack evidence of ongoing subclonal positive selection, consistent with patterns of CRC evolution observed earlier[40].

Parameters of the fit for the Power Law Type-I tail available in MOBSTER were then used to retrieve the tumor mutation rate >0. This quantity is canonically expressed in time units of tumor cell doublings—that is, considering discrete time-evolution steps in which all tumor cells divide synchronously—and depends on the size of the analyzed tumor genome. To make it comparable across multiple samples of the same patient and to account for the fact that we used whole-exome data, we normalized by the size of the diploid exome regions in each biopsy of every patient. These gave us the point estimates reported in Main.

We then sought out to build a confidence interval (CI) for $\mu$, adopting a nonparametric bootstrap procedure[41]. In practice, we bootstrapped with repetitions from the mutations available in each sample and built $n$ = 200 datasets per patient. Then we reran the MOBSTER analysis conditioning on retrieving the putative monoclonal architecture ($k = 1$) identified in the main run and recomputed the normalized values for the bootstrap estimate of $\mu$. With the distribution of bootstrapped $\mu$ values, we built a percentile CI corresponding to an $\alpha$-level of 5% by taking the 2.5% and 97.5% empirical quantiles.

**PDO cultures**
The collection of patient tissue for the generation and distribution of organoids has been performed according to the guidelines of the European Network of Research Ethics Committees following European, national and local law. In all cases, patients signed informed consent after ethical committees approved the study protocols.

A surgically resected T2-stage colorectal tumor was obtained from the University Medical Center Utrecht Hospital. Punch biopsies of five different tumor regions and adjacent normal tissue were collected in basal medium (Advanced DMEM/F12 (Invitrogen) supplemented with 1% penicillin–streptomycin (Lonza), 1% HEPES buffer (Invitrogen) and 1% Glutamax (Invitrogen)). The tissues were chopped into ~5 mm fragments and incubated in basal medium supplemented with 1 mg ml$^{-1}$ dispase (Gibco) and 1 mg ml$^{-1}$ collagenase (Merck) for 30 min at 37 °C/5% $CO_2$ and subsequently fragmented through shear stress (pipetting). Of each biopsy, four crypts/tumor fragments were isolated and grown into clonal lines. The residual biopsy was taken into the culture in bulk. Tissue fragments were embedded in Matrigel matrix domes (Corning) and expanded in CRC culture medium (basal medium supplemented

with 20% R-spondin 1 conditioned medium, 10% Noggin conditioned medium, 1× B27 (Gibco), 1.25 mM N-acetyl-L-cysteine (Sigma), 500 nM A83-01 (Tocris Bioscience), 0.5 nM Wnt Surrogate-Fc Fusion protein (U-protein express), 50 ng ml$^{-1}$ recombinant human EGF (Peprotech), 50 ng ml$^{-1}$ human FGF-basic (Peprotech), 100 ng ml$^{-1}$ recombinant human IGF1 (BioLegend), 10 µM Y-27632 (Gentaur) at 37 °C/5% CO$_2$.

At passage 4, genomic DNA was extracted from the 24 clonal lines according to the manufacturer's instructions (QIAamp DNA Micro Kit; Qiagen). WGS libraries were generated using standard Illumina protocols. WGS libraries were sequenced to ~15× genome coverage (2 × 150 bp) on an Illumina NovaSeq 6000 system at the Utrecht Sequencing Facility. The WGS data were processed as described previously (https://github.com/ToolsVanBox/NF-IAP). Briefly, reads were aligned to the human reference genome (GRCh38) using the Burrows-Wheeler Aligner (v0.7.17). After marking duplicates using Sambamba (v0.6.8), variants in the multisample mode were marked by using GATK's HaplotypeCaller (v4.1.3.0). The in-house-developed Somatic Mutations Rechecker and Filtering (SMuRF) tool (v3.0.0) was used to filter somatic variants (https://github.com/ToolsVanBox/SMuRF). Somatic variants with a VAF of less than 0.25, a base coverage of less than five reads, an MQ of less than 55, a GATK phred-scaled quality score (QUAL) < 100 and/or presence in a panel of unmatched normal human genomes were excluded.

In addition, MSH homopolymer loci were genotyped regularly during the culturing stages by targeted locus PCR amplification (NEB Q5 Polymerase), PCR product ligation into pJet1.2 blunt end plasmids (CloneJET PCR Cloning Kit, Thermo Fisher Scientific) and subsequent Sanger sequencing of plasmids isolated from bacterial colonies (Macrogen Europe). The following oligos were used to amplify the MSH homopolymer loci (MSH2 FW: 5′-gattgtatctaagcaactttcc-3′, RV: 5′-ctgacatgctcgtgctatg-3′; MSH3 FW: 5′-gaatcccctaatcaagctgg-3′, RV: 5′-caagaccatctggatctctcc-3′; MSH6 FW: 5′-cagagattgttttcatatcagtg-3′, RV: 5′-cagttgctagaggtcatgaac-3′; IDT DNA).

### PDO time course

Four independent *MLH1*$^{-/-}$ organoid cultures were selected based on MSH mutant status (MSHwt*; CRISPR generation of *MLH1*$^{-/-}$ organoids), *MSH2*$^{+/-}$, *MSH3*$^{+/-}$ and *MSH3*$^{-/-}$; *MSH6*$^{+/-}$). All lines underwent a clonal step (FACS (fluorescence activated cell sorting) sorting of single cells), marking the start of the experiment at passage 6. For each line, we cultured six independent clonal lines (4 × 6 clonal lines 't = 1') and expanded these lines for 9 weeks. After these 9 weeks, one-third of the culture was frozen down (Recovery Cell Culture Freezing Medium; Thermo Fisher Scientific), one-third of the culture was harvested to extract gDNA by the QIAamp DNA Micro Kit following the manufacturer's instructions (Qiagen; shared mutations with VAF = 0.5 are a proxy for the t = 1 clonal cell) and one-third of the culture underwent a second clonal step (FACS sorting of single cells), marking the endpoint of the time course. For each of the 24 't = 1' lines, six subclones were expanded in culture (total of 144 lines 't = 2') for 6 weeks to reach sufficient material for harvesting. In total, half of the culture was frozen down and half was harvested for genomic DNA extraction (shared mutations with VAF = 0.5 are a proxy for 't = 2'). We selected four pairs ('t = 1' versus 't = 2') for all four original tumor genotypes (4 × 4 = 16), except for the *MSH3*$^{-/-}$;*MSH6*$^{+/-}$ variant. We included two additional 't = 2' clones of this genotype to track homopolymer stability within a population over a period of 9 weeks. In total, we generated WGS libraries of 18 isogenic organoid lines using standard Illumina protocols. WGS libraries were sequenced to ~15× genome coverage (2 × 150 bp) on an Illumina NovaSeq 6000 system at the Utrecht Sequencing Facility.

### WGS and read mapping

The time course 15× WGS samples were analyzed with the Hartwig pipeline for somatic variant calling (https://github.com/hartwigmedical/pipeline5), which was hosted on the Google Cloud Platform using Platinum (https://github.com/hartwigmedical/platinum). Details of the full pipeline are described in previous work[42] and in the Hartwig pipeline GitHub page. Briefly, reads were mapped to the reference genome GRCH38 using BWA-mem v.0.7.5a, duplicates were marked for filtering and InDels were realigned using GATK v.3.4.46 IndelRealigner. GATK HaplotypeCaller v.3.4.46 was run to call germline variants in the reference sample. For somatic SNV and InDel variant calling, GATK BQSR was applied to recalibrate base qualities. SNV and InDel somatic variants were called using Strelka v.1.0.14 with optimized settings and postcalling filtering. Structural variants were called using Manta (v.1.0.3) with default parameters followed by additional filtering to improve precision using an internally built tool (Breakpoint Inspector v.1.5). CN calling and determination of sample purity were performed using PURPLE (PURity & PLoidy Estimator), which combines B-allele frequency, read depth and structural variants to estimate the purity of a tumor sample and determine the CN and minor allele ploidy for every base in the genome. The number of somatic mutations falling into the 96 SBS, 78 DBS (doublet bases substitution) and 83 InDel contexts (as described in COSMIC: https://cancer.sanger.ac.uk/signatures/) was determined using the R package mutSigExtractor (https://github.com/UMCUGenetics/mutSigExtractor, v1.23). To obtain the mutational signature contributions for each sample, the mutation context counts were fitted to the COSMIC catalog of mutational signatures using the nnlm() function from the NNLM R package. Ultimately, TMB was determined for all genotypes by subtracting the total mutation burden 't = 2' minus 't = 1'. Statistical tests were performed in GraphPad Prism software and R studio.

### CRISPR generation of *MLH1*$^{-/-}$ organoids

The 'MSHwt' clonal line was derived from paired normal organoids in which we generated an *MLH1*$^{-/-}$ null allele using CRISPR–Cas9 technology, as described previously[43]. In brief, exon 2 of MLH1 was disrupted by the insertion of a puromycin-resistance cassette, using the following gRNA: 5′-AGACAATGGCACCGGGATCAGGG-3′ and Cas9 plasmid (Addgene, 48139) using the NEPA21 Super Electroporator (Nepa Gene) following described conditions[44]. Puromycin-resistant clones were genotyped, and loss of MLH1 protein was confirmed (not shown; Cell Signaling Technology, MLH1 4C9C7).

### Immunofluorescence staining of PDOs

The four selected biopsy clonal lines (*MSHwt*, *MSH2*$^{+/-}$, *MSH3*$^{+/-}$ and *MSH3*$^{-/-}$/*MSH6*$^{+/-}$) were immunostained for MSH3 nuclear protein levels as described previously[45]. In brief, organoids were dislodged from their Matrigel matrix domes by incubation in basal medium supplemented with 1 mg ml$^{-1}$ dispase for 30 min at 37 °C/5% CO$_2$ and pelleted after several washing cycles with basal medium. Organoids were fixed in 4% paraformaldehyde in PBS on ice for 45 min. Fixed organoids were transferred to repellent plates (Greiner Bio-One). Permeabilization, blocking and antibody incubation steps were done in an organoid washing buffer (0.1% Triton X-100 in PBS and −0.2% wt/vol BSA) at 4 °C on a shaker platform. Primary antibody (BD Biosciences; purified mouse anti-human MSH3, 1:100 dilution), secondary antibody Alexa568 anti-mouse (Life Technologies; 1:1,000) and Hoechst. Organoids were mounted in clearing solution (ddH$_2$O, 60% (vol/vol) glycerol and 2.5 M fructose) and imaged on a Zeiss LSM880 confocal laser scanning microscope at ×40 magnification. Images were processed in Fiji software. Hoechst was used as a nuclear reference marker to quantify nuclear MSH3 levels. Statistical analysis was performed in GraphPad Prism software.

### Multiplex immunofluorescence (MIF) of UCL CRC cohort

A panel consisting of MSH6, CD20, FOXP3, CD4, pan-CK (pan-cytokeratin) and CD8 were selected for the MIF assay. Primary antibody details are provided in Supplementary Table 5. The opal (Akoya) MIF automation kit was used, which includes HRP-conjugated (horseradish

peroxidase) secondary antibody, opal fluorophores, DAPI stain, antibody diluents and blocking buffers. The manufacturer's protocol was followed, and immunostaining was performed using the Leica Bond RX autostainer (Leica Biosystems).

**Monoplex optimization.** To optimize the labeling of each marker, monoplex slides were created where tissue sections were labeled with each primary antibody on separate slides. Each primary antibody was assigned an opal fluorophore. Monoplex slides were processed with an appropriate number of antibody stripping steps before and after staining, reflecting the eventual multiplex sequence. Monoplex slides were imaged using the Vectra 3.0 fluorescence microscope, and signal counts were assessed using the Inform software.

**Autofluorescence slide.** A representative tumor section was labeled with pan-CK primary antibody and without opal fluorophore to assess levels of background autofluorescence.

**Library development.** To create a spectral unmixing library, slides were stained with the most abundant marker (pan-CK) and each opal fluorophore individually, resulting in six library slides. Library and autofluorescence slides were imaged on the Vectra 3.0 using all five epi-fluorescence filters (DAPI, FITC, Cy3, Texas red and Cy5). The spectral unmixing library was developed using Inform software.

**Multiplex assay development.** The multiplex assay was run using the optimized conditions developed during monoplex optimization. Timings, temperature settings and reagent concentrations for each step are detailed in Supplementary Table 5. The following steps were performed on the Leica Bond RX autostainer:

1. Deparaffinization using Bond dewax solution
2. Antigen retrieval solution using Bond ER1 or ER2 solution
3. Blocking buffer
4. Primary antibody incubation
5. Opal polymer HRP incubation
6. Opal fluorophore incubation
7. Stripping of antibody complexes using Bond ER1 or ER2 solution
8. Repeat steps two to seven until all primary antibodies are applied
9. DAPI counterstain

Following the immunostaining steps mentioned above, the slides were cover-slipped manually using Diamond Antifade Mountant (Invitrogen) and imaged using Vectra 3.0. High-power images were taken from MSH6-proficient and MSH6-deficient regions of interest (ROI). The size of each ROI was the same in all experiments at $1\,mm^2$.

**ORION cell segmentation workflow.** In this study, we developed ORION, a cell segmentation workflow for multispectral immunofluorescence imaging (see Extended Data Fig. 5 for an overview). ORION, an unsupervised method, uses an established ellipsoidal model[46] to identify individual cells and exclude noise and noncell objects. As the ellipsoidal models do not require labeled data and extensive training procedures, they provide promising results in unannotated multiplex IF datasets that include a high degree of cell shape and intensity variations. To this end, initially, the unmixed spectral signatures undergo a Gaussian filter with a $5 \times 5$ kernel to remove small artifacts. Subsequently, an adaptive thresholding method that performs well in images with foreground and background intensity heterogeneity[47] is applied for the selection of the optimal threshold value for each pixel within its local neighborhood. This requires mean filtering and estimation of the local threshold based on the mean neighborhood pixel intensity. In the resulting $M$ binary image, morphological operations consisting of erosion, dilation and removal of small elements are applied, to suppress small artifacts.

For the separation of touching cells, an improved ellipsoidal modeling approach is performed. Initially, we estimate the distance transformation of the binary image $M$ of $p$ pixels that represents the connected cells and we estimate the regional maxima of this. Given that the number and location of local maxima correspond to those of nuclei, we reject the touching maxima. The remaining maxima comprise the list of candidate seeds. Then, based on the hypothesis that cells can be spatially modeled as ellipsoids $E_C$, the pixels of cells are then modeled using a Gaussian distribution. More specifically, a Gaussian mixture model is applied with the number of clusters $C$ being equal to that of candidate seeds, and the mixture parameters, namely the mean and variance, are estimated using the expectation–maximization (EM) algorithm. For the initialization of the EM algorithm the $k$-nearest neighbor classification using Euclidean distance as the distance metric is used to estimate the initial parameters. The EM is an iterative method consisting of the following two steps: (1) expectation, which computes the likelihood with respect to the current estimates, and (2) maximization (equation (1)), which maximizes the expected log-likelihood (equation (2)) as follows:

$$Q\left(\theta,|,\theta^{(t)}\right) = E_{Z|X,\theta^{(t)}}(\log L\left(\theta;X,Z\right)) \tag{1}$$

$$\theta^{(t+1)} = \arg\max_{\theta} Q(\theta|\theta^{(t)}) \tag{2}$$

where $Q$ is the expected values of the log-likelihood function $\theta$, $X$ is the pixel coordinates, $Z$ is the latent variables and $\theta^{(t)}$ is the current parameters.

Having estimated the ellipsoidal model of cells, we need to reject any erroneous seeds from the candidate list and re-estimate the ellipsoidal models for the remaining seeds. For this study, we developed a new fitness validation criterion taking into account the overall combination of ellipses of candidate seeds. More specifically, the proposed criterion aims to keep the cells well-separated and takes into account the binary areas that are included in the estimated ellipses, the total area of the extracted ellipses, as well as the background area that is included in the estimated ellipses and the overlapping parts of the ellipses of the touching cells. Subsequently, we introduce an intensity-based parameter $W_I$ based on the intensity variance of each estimated ellipse aiming to separate the touching cells with different intensities. In the case that the estimated ellipses fit perfectly to the binary mask $M$, the value of the fitness function tends to be equal to 1. To this end, the number of candidate seeds and the estimated ellipsoidal components are defined by maximizing the following fitness degree of the 2D cell data mask:

$$\left(\frac{A_F - A_B - A_T - W_I}{E}\right) \tag{3}$$

where the total area covered by the estimated ellipses is $E = \sum_{p \in M} E_{C(p)}$, the foreground area of the binary image $M$ that is included in the estimated ellipses is $A_F = \sum_{p=1} M(p)E(p)$, the area of the background area of the binary image $M$ that is included in the estimated ellipses is $A_B = \sum_{p=1}(1 - M(p))E(p)$ and the overlapping parts of the ellipses of the touching cells for the total number of the identified ellipses is defined as $A_T = \sum_{i=1} \sum_{p=1} E_{C_i}(p) \cap E_{C_j}(p), j = 1, j \neq i$. The final segmentation of the clustered cells is performed by applying Bayesian classification that assigns each pixel $p$ to cluster $C_i$ with the maximum posterior probability.

To evaluate the performance of ORION, we conducted tests using three datasets and compared them with eight independent state-of-the-art cell segmentation approaches. More specifically, we used three datasets for this evaluation—two publicly available datasets (datasets A and B) and a subset of the multispectral IF images of MSI tumors created in this study (dataset C). Dataset A included

48 fluorescence images of 1,831 cells, while dataset B included 49 cell nuclei images of 2,178 nuclei in total. Dataset C consisted of 400 cells of multiplex IF images. Furthermore, both traditional (namely Otsu[48], three-step[49], watershed[50], LSBR[51], LLBWIP[52] and RFOVE[53]) and deep learning models (namely U-net[54] and Mask R-CNN[55]) were used for the comparison. Although deep learning models usually achieve higher segmentation accuracy than traditional methods, they require higher computational cost and annotation time. To validate the efficiency of ORION, we used the Jaccard similarity coefficient as well as Dice false positive and Dice false negative values to measure oversegmentation and undersegmentation, respectively. Furthermore, Hausdorff distance and mean absolute contour distance were used to evaluate the contour of detected cells. Moreover, we estimated the true detected rate to determine the ratio of segmented cell number to the total number of annotated cells. Results detailed in Supplementary Table 6 demonstrate that the proposed method outperforms other methods. Also, we note that deep learning models exhibit lower accuracy than ellipsoidal modeling-based algorithms as the number of annotated training examples is limited and the application of transfer learning to datasets with high diversity between single data, such as multiplex IF data, is challenging. Application of both ellipsoidal and deep learning models is consistent with previous experiments[53,54], conducted in the publicly available datasets used for validation in this study.

Using ORION, cell segmentation of different markers was carried out based on DAPI nuclear staining or combining DAPI and cytoplasmic staining due to a lack of clear cell boundaries (for example, CD4+ cells). Because both tumor cells and immune cells may express MSH6, we used colocalization of MSH6 expression with the epithelial marker pan-CK to identify tumor cells and with immune markers (CD8, CD4 and CD20) to identify specific immune cell populations. The experimental results (Supplementary Table 6) show that ORION outperformed eight state-of-the-art approaches that have been used in the past for cell segmentation. Finally, the true detected rate for the three datasets was estimated to be equal to 98.1%.

Following the validation of ORION, we next ran the workflow on the full dataset of 194 multispectral image tiles from 26 tumors. We performed neighborhood analysis[56] to quantify the number of immune cells of each subtype within the vicinity of MSH6-proficient and MSH6-deficient tumor cells. We then used the localized segmented cell centers and a radius $R = 100$ μm to identify the spatial associations with immune cells. We chose this radius as a biologically relevant distance for interaction between tumor cells and immune cells. We counted the number of the different immune cells that were identified within this radius. For each tile, we reported the sum total of immune cells of each subtype identified from neighborhood analysis.

### GEL 100,000 Genomes CRC dataset

**WGS, variant calling, purity and ploidy estimation.** WGS data were generated through a standardized, clinical pathology-accredited, workflow as part of the GEL 100,000 Genomes Project[57]. Briefly, the sequencing data were aligned to the human genome GrCh38 using the Illumina iSAAC aligner and were then subjected to extensive variant calling and quality control processes. The basis of this is the Strelka variant caller plus artifact filtering using a project-wide panel of normal, and population-based filtering using the aggregated gnomAD dataset. We also scrutinized the resultant SNV calls taking a minimum depth cutoff of 50×. We used the package Sequenza to derive tumor purity, ploidy and CN estimates for each sample as described above. In total, 992 CRCs were identified for study using the V8 data release (available as of November 2019).

**Identification of MSI cancers.** Cancers with MSI were detected using the MSIsensor (version 0.6) package and validated using two methods. For the identification of MSI-high cases in the GEL cohort, we used default settings when running MSIsensor. As per the GitHub page given below, the maximal homopolymer size default is 50 bp and the maximal length of microsate is 5 bp. The depth threshold is 20×, and the false discovery rate is 0.05. As per the default settings, an MSIsensor score of >3.5 was used as the cutoff to call a sample MSI high.

MSI-high tumors identified using MSIsensor were then validated using the mutational signature analysis available as part of the GEL V8 release, and we confirmed the enrichment of an MMRd SBS mutation signature. As further validation, cases with available pathological data were confirmed to be MMRd by IHC (histology validation set, $n = 101$, 98% classification accuracy). Tumors with discernible pathogenic POLE or POLD1 exonuclease domain mutations were excluded. The resulting cohort of 217 cases was used for downstream analysis and is referred to here as the GEL CRC MSI cohort.

**Identification of primary MMR defect.** Germline and somatic mutations in the MMR genes (MLH1, PMS2, MSH2, MSH6 and MSH3) were identified by searching the relevant GEL main program tiering data for tier 1 pathogenic mutations. To identify tumors with MLH1 promoter methylation, the presence of somatic BRAF^V600E was used as an indicator. This approach is in keeping with current clinical guidelines where it is recognized that among MMRd colorectal tumors the presence of somatic BRAF^V600E mutation associates strongly with MLH1 promoter methylation[9,58].

**Mutation frequency of *MSH6*/*MSH3* MSs compared with other length 8 coding MSs.** Because *MSH6* and *MSH3* contain a C8 and A8 coding homopolymer, respectively, we were interested in comparing the frequency with which these sites are mutated across the cohort compared to other length 8 homopolymers of the same nucleotide base. The genomic coordinates of all length 8 exonic MSs were obtained using the SciRoKo package. The mutation status of all length 8 coding MSs and the base affected were extracted from the variant call files by filtering for the genomic coordinates of length 8 coding MSs. The percentage of cases with a frameshift mutation in C:G or A:T MSs was calculated separately and compared to the mutation frequency observed for *MSH6* (C8) and *MSH3* (A8) MSs, respectively. Comparisons were made using the chi-squared test.

**Multiple linear regression model.** To determine the contribution of *MSH6*^F1088fs and *MSH3*^K383fs frameshifts to mutation burden in MSI CRCs, multiple linear regression modeling was performed. The presence or absence of frameshift mutations in the coding homopolymers of *MSH6* and *MSH3* together with the mutation status of coding homopolymers in a further 21 genes were used as independent variables in the model. The list of 21 genes were those reported as recurrently mutated in MSI CRC[11] and consisted of *RFX5, MBD4, AIM2, ACVR2A, DOCK3, TGFBR2, GLYR1, OR51E, CLOCK, CASP, JAK1, TAF1B, BAX, MYH11, HPS1, SLAMF1, HNF1A, RGS12, ELAVL3, SMAP1* and *SLC22A9*. We also included tumor purity and age at diagnosis in the model to account for potential confounding. There was no difference in estimated tumor purity between *MSH6* and *MSH3* mutated groups (one-way ANOVA, $P = 0.742$). The model was created using the lm function in R, and the results were plotted as a volcano plot with regression coefficients of contribution to mutation burden versus $-\log_{10}(P)$ of the $t$ statistic. We also ran the model using just *MSH6* and *MSH3* frameshifts as independent variables and obtained estimates of the contribution of these frameshifts individually and combined on the total mutation burden. Results of the model output are provided in Supplementary Table 1.

**Identification of *MSH6*/*MSH3* coding mutations outside of length 8 homopolymers.** Variant call files were searched for all coding *MSH6* and *MSH3* mutations. Data were extracted and tabulated according to the frequency and type of mutation (Fig. 1b and Supplementary Table 2).

**Mutation burden analysis.** Total mutation, SNV and InDel counts for each sample were measured using data from variant call files. Both synonymous and nonsynonymous SNVs were included. Violin and waterfall plots were generated in R using the ggplot package.

**Analysis of MSI cases with confirmed primary MLH1/PMS2 (MutL) deficiency.** To confirm that differences in the primary cause of MMR loss in samples were not confounding results, we restricted our analysis to cases with confirmed MLH1/PMS2 (MutLα) deficiency. We identified tumors with somatic $BRAF^{V600E}$ mutation (indicating $MLH1$ promoter methylation) and also samples with tier 1 pathogenic germline $MLH1$ or $PMS2$ mutations. Somatic $BRAF^{V600E}$ mutations were identified from variant call files and germline mutations from the GEL main program tiering data. Violin plots for total mutation, SNV and InDel burden according to $MSH6$ and $MSH3$ homopolymer mutation status were plotted on this subset of the cohort.

### TCGA dataset
MSI cancers within the TCGA dataset were identified from a previous study[11]. Variant calls for these tumors were downloaded from the National Cancer Institute Genomics Data Commons (GDC) portal (https://portal.gdc.cancer.gov/repository). Cases with a frameshift mutation in the $MSH6$ and $MSH3$ coding MSs were identified from the supplementary data provided in ref. 11. Tumors with discernible mutations in $POLE$ and $POLD$ exonuclease domains were excluded[59]. This resulted in an MSI cohort consisting of the following tumor types: colorectal ($n = 48$), uterine ($n = 67$), stomach ($n = 63$) and esophageal ($n = 3$). Mutation burden plots were created in R using the ggplot package according to the $MSH6$/$MSH3$ mutation status of tumors. To analyze neoantigen counts, data were obtained from ref. 7. Briefly, tumor purity and ploidy were estimated using ASCAT on Affymetrix SNP array data. Samples with purity below 0.4 and ploidy above 3.6 were excluded. This reduced the cohort size to 117 samples (colorectal = 34, uterine = 56 and stomach = 27). Neoantigens were predicted using the Neopredpipe pipeline, as detailed in ref. 37. To analyze MSH3 and MSH6 RNA expression levels, raw RNA counts were obtained, transformed to FPKM and then converted to transcripts per kilobase million (TPM) values using the following formula: $TPM = FPKM/sum (FPKM) \times 10^6$. RNA expression data were available for 127 samples (colorectal ($n = 42$), uterine ($n = 56$) and stomach ($n = 29$)).

### Mathematical model of the effect of stochastic mutation rate switching on tumor growth
Our model was based on our previous stochastic branching process modeling of tumor growth and neoantigen accumulation[7]. The model simulated tumor growth where each cell can either (1) die with a probability inversely proportional to their fitness or (2) divide into two daughter cells that accumulate new mutations according to their respective mutation rate. Cells in hypermutated and ultra-hypermutated states gain a number of mutations in each division sampled from a Poisson distribution with parameters (mutation rate, $\mu$) 6 and 120, respectively. Our previous work has shown that $\mu_{basal} = 6$ (average six exonic mutations gained per division) accurately recapitulates the range of subclonal mutation burdens observed across the TCGA MMRd CRC cohort[7]. Likewise, our MOBSTER data (this manuscript) revealed a 20-fold mutation rate difference in vivo (cf. Fig. 3p–o), and here we modeled $\mu_{high} = 120$ mutations/division. New mutations are either (1) neutral with no effect on cell fitness; (2) antigenic, decreasing the cell's fitness; (3) immune escape mutations that eliminate immune predation and therefore nullify antigen-induced fitness decrease; or (4) lethal, irreversibly decreasing cell fitness regardless of immune escape (Fig. 6a). The probability of a given mutation being nonneutral is defined by $P$(antigen), $P$(escape) and $P$(lethal), respectively. Note that these mutation types are nonexclusive, and a mutation can be, for example, both antigenic and lethal (although with only a

small probability). In addition, at each division, the daughter cells may undergo mutation rate switching with probability $\beta$. $\beta = 0$ corresponds to no switching (mutation rates remain constant), while $\beta > 1/100$ represents frequent switches to or from an ultra-hypermutated state. Each tumor was initiated with a homogeneous population of 100 tumor cells, all in either hypermutated or ultra-hypermutated states, and simulated until elimination (no tumor cells left) or until it reached detectable size (>100,000 cells).

**MS diversity.** We encoded the mutation status of the MS locus as an integer−0 represented a WT allele, −1/+1 represented a single deletion/insertion and so on. Upon division, daughter cells inherited the mutation status of their ancestor. Every new mutation had a probability, $p$(ms), to affect the MS−if they did, the state of the locus was changed from $n$ to $n + 1$ or $n − 1$ with equal probability. At the end of the simulation, the mutation status of all cells was read out and the total Shannon diversity of the population was computed in R (using the package entropy).

**Growth time.** We defined the start of the 'growth period' as the last time point when the population count went below 20 (immune-escaped) cells. The final time was the time point when the population reached 100,000 cells. Growth time was computed as $T$(final) − $T$(growth-start). We chose this measure over $T$(final) as the latter had a very high uncertainty due to the variable time lineages spent before probabilistically acquiring immune escape and initiating unimpeded growth.

**Parameter values.** The following default parameter values were used in all simulations unless indicated otherwise (for example, a range of $P_{lethal}$ values in Fig. 4f) neoantigen probability, $P_{antigen} = 0.1$; immune escape probability, $P_{escape} = 10^{-6}$; lethal mutation probability, $P_{lethal} = 5 \times 10^{-4}$; MS-shifting rate, $P_{ms} = 10^{-3}$ and immune-related selection coefficient, $s = -0.8$ (representing moderate selection).

### Statistical analysis
The Kruskal–Wallis test was used to test for a difference in the distribution of three or more groups with post hoc pair-wise comparisons performed using the Wilcoxon unpaired test. A $P$ value of less than 0.05 was considered significant. Multiple linear regression was performed using the Lm package, and linear mixed-effect modeling was performed using the LMER package in R. For correlation analysis, Pearson and Spearman rank correlation was used to assess for linear and monotonic relationships, respectively. The chi-squared test was used to compare the distribution of categorical variables. Statistical analyses were performed in R (version 3.6.2).

### Reporting summary
Further information on research design is available in the Nature Portfolio Reporting Summary linked to this article.

### Data availability
Details of access to all datasets used in this study are listed below:
1. To access the WGS data from the GEL CRC dataset, researchers must apply to become a member of either the GEL Research Network (www.genomicsengland.co.uk/research/academic) or as a Discovery Forum industry partner (www.genomicsengland.co.uk/research/research-environment) and follow the detailed instructions. This dataset is under controlled access as per the terms and conditions of the GEL Clinical Information Partnership (Available at: https://files.genomicsengland.co.uk/documents/GeCIP-Rules_29-08-2018.pdf).
2. Variant calls from the TCGA whole-exome sequencing dataset can be retrieved from the Genomics Data Commons website (https://portal.gdc.cancer.gov/).
3. Whole-exome sequencing data from the UCL CRC cohort and WGS data for the PDO work are available at the European Genome Phenome

archive under the following accession codes: EGAS50000000218 and EGAS50000000297, respectively.

## Code availability

Software packages used for this publication are listed below. The study did not use any custom code or packages.

1. Tumor phylogenetic reconstruction: Paup (https://paup.phylosolutions.com/)
2. Generation of phylogenetic trees: Figtree (version 1.4.4; https://github.com/rambaut/figtree/releases)
3. Neoantigen calling: Neopredpipe (https://github.com/MathOnco/NeoPredPipe)
4. Hompolymer read length distribution analysis: MSIsensor (version 0.6; https://github.com/ding-lab/msisensor/blob/master/README_msisensor.md)
5. Mutation signature analysis: SigProfiler (version 3.1; https://github.com/AlexandrovLab)
6. Immune dN/dS analysis: SOPRANO (https://github.com/luisgls/SOPRANO)
7. Subclonal deconvolution and mutation rate analysis: MOBSTER (https://github.com/caravagnalab/mobster)

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

## Acknowledgements

This research was made possible through access to the data and findings generated by the 100,000 Genomes Project. The 100,000 Genomes Project is managed by Genomics England (a wholly owned company of the Department of Health and Social Care). The 100,000 Genomes Project is funded by the National Institute for Health Research and the National Health Service (NHS) England. The Wellcome Trust, Cancer Research UK and the Medical Research Council have also funded research infrastructure. The 100,000 Genomes Project uses data provided by patients and collected by the NHS as part of their care and support. We thank all the patients who contributed samples to this study. I. Bjedov is gratefully acknowledged

for discussions regarding bacterial frameshift switching. Support was provided to the Jansen Lab by the National Institute for Health Research, the University College London Hospitals Biomedical Research Center and the Cancer Research UK University College London Experimental Cancer Medicine Center. We also acknowledge the Pathology, Genomic and Microscopy Translational Technology Platforms (TTP) at the UCL Cancer Institute. H.K. is supported by a Cancer Research UK clinical research training fellowship (542093). M.J. is supported by a Cancer Research UK Clinician Scientist Fellowship (A22745). M.J. and H.K. receive funding from Bowel Research UK (553856) and Rosetrees Trust (M670 and 100045). B.W. is supported by the Barts Charity Lectureship (grant MGU045) and a UKRI (United Kingdom Research and Innovation) Future Leaders Fellowship (grant MR/V02342X/1). A.S. was supported by the Wellcome Trust (202778/B/16/Z) and Cancer Research UK (A22909 and A26815). T.A.G., supporting E.L., was supported by Cancer Research UK (A19771 and DRCNPG-May21_100001) and the Wellcome Trust (202778/Z/16/Z). G.C. is supported by AIRC (Italian Foundation for Cancer Research) under MFAG 2020 number 24913. We thank O. Kranenburg (University Medical Center Utrecht) and U-PORT (UMCU) for their support and the funding received by HJGS from the ERC starting grant (803608) and NWO VIDI (09150172010017). The Oncode Institute is partly financed by the Dutch Cancer Society.

## Author contributions

M.J., H.K. and H.J.G.S. conceived the project and designed the experimental strategy. H.K. performed molecular pathology wet lab experiments. K.L. performed variant calling of sequencing data. H.K. performed downstream bioinformatic analysis. P.B., C.J. and D.A. performed image analysis of MIF data. S.E.M.v.d.H., A.V.H., S.M., R.v.B., J.H.H., J.R.B., M.M.L., L.M.G.M., L.B. and H.J.G.S. performed and analyzed PDO experiments. E.L. and T.G. performed mathematical modeling experiments. G.C. and S.M. performed MOBSTER clonal deconvolution and mutation rate analysis. L.Z. performed immune dN/dS analysis. D.P. contributed to MIF experiments. J.L., M.N. and M.R.J. contributed to IHC experiments and interpretation. C.S. provided advice on statistical analysis. W.W., K.K., D.H., A.B., M.N., B.W., M.J.L., I.N., N.P., A.S. and M.R.J. were involved in the interpretation of the data. M.J., H.K., S.E.M.v.d.H, A.V.H., H.J.G.S., P.B., G.C., E.L., W.C. and K.K.S. contributed to writing the manuscript. All authors edited and approved the final manuscript.

## Competing interests

The authors declare no competing interests.

## Additional information

**Extended data** is available for this paper at https://doi.org/10.1038/s41588-024-01777-9.

**Correspondence and requests for materials** should be addressed to Hugo J. G. Snippert or Marnix Jansen.

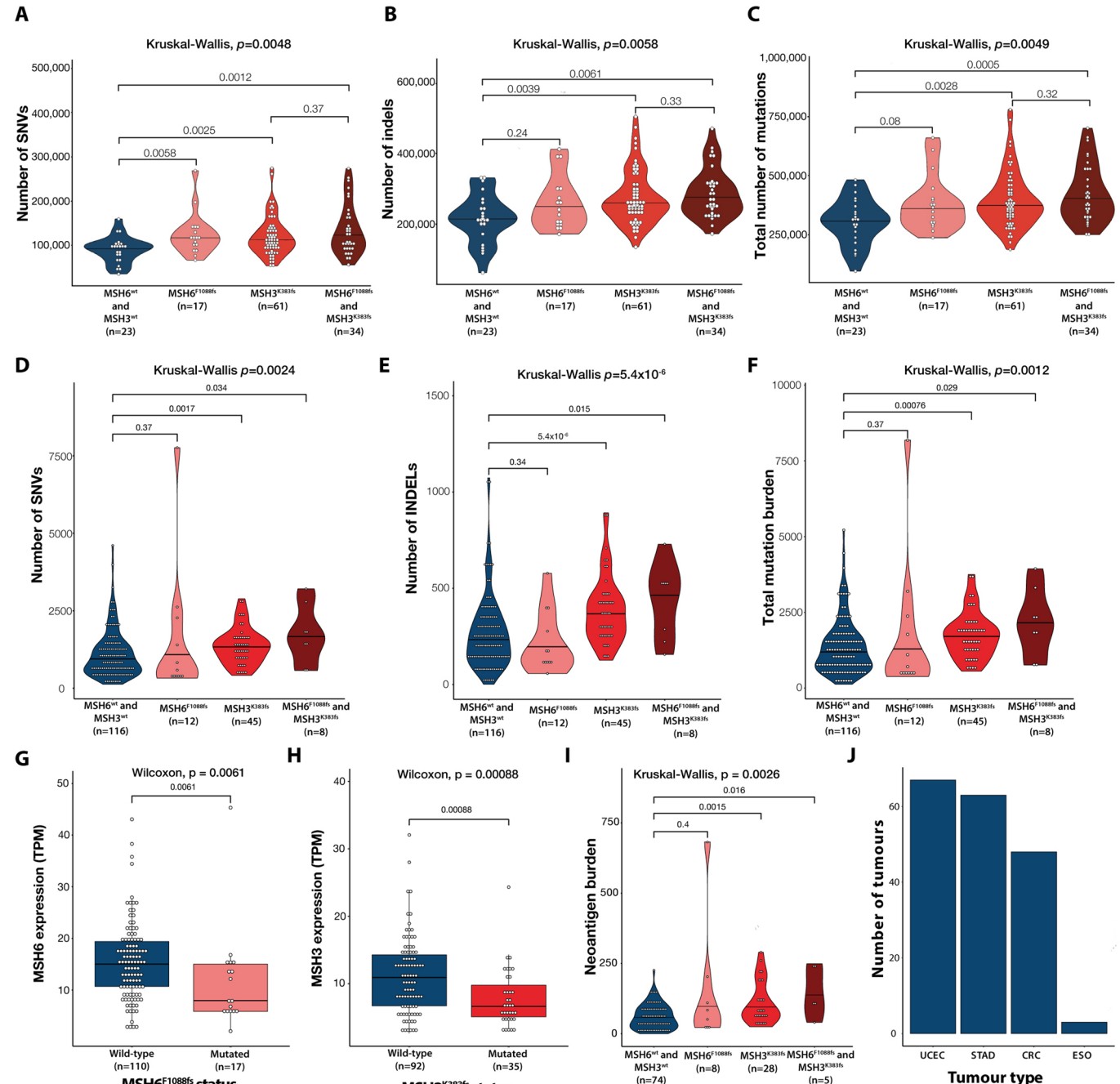

**Extended Data Fig. 1 | Mutation burden is increased in presence of incremental *MSH6^F1088fs* and *MSH3^K383fs* across cohorts.** (a–c) Genomics England MSI CRC cohort subset for cases with confirmed MutL (MLH1/PMS2) loss as primary cause of MMRd. Violin plots show SNV (a), InDel (b) and total mutation burden (c) in tumors according to presence of secondary *MSH6*^F1088 and *MSH3*^K383 frameshifts. (d–j) TCGA MSI validation cohort. (d–f) Violin plots display SNV (d), InDel (e) and total mutation burden (f) according to presence of secondary *MSH6*^F1088 and *MSH3*^K383 frameshifts. (g,h) RNA expression of MSH6 (g) and

MSH3 (h) is reduced in tumors with *MSH6* and *MSH3* frameshifts, respectively. Two-sided Wilcoxon test reported. (i) Neoantigen burden according to *MSH6*^F1088 and *MSH3*^K383 frameshifts. (j) Number of tumors according to tumor type making up the TCGA MSI cohort. UCEC = uterine corpus endometrial carcinoma, STAD = stomach adenocarcinoma, CRC = colorectal adenocarcinoma and ESO = esophageal adenocarcinoma. Boxplots: horizontal black line represents median. Lower and upper hinges represent 1st and 3rd quartiles. Lower and upper whiskers extend to values up to 1.5× interquartile range from the hinge.

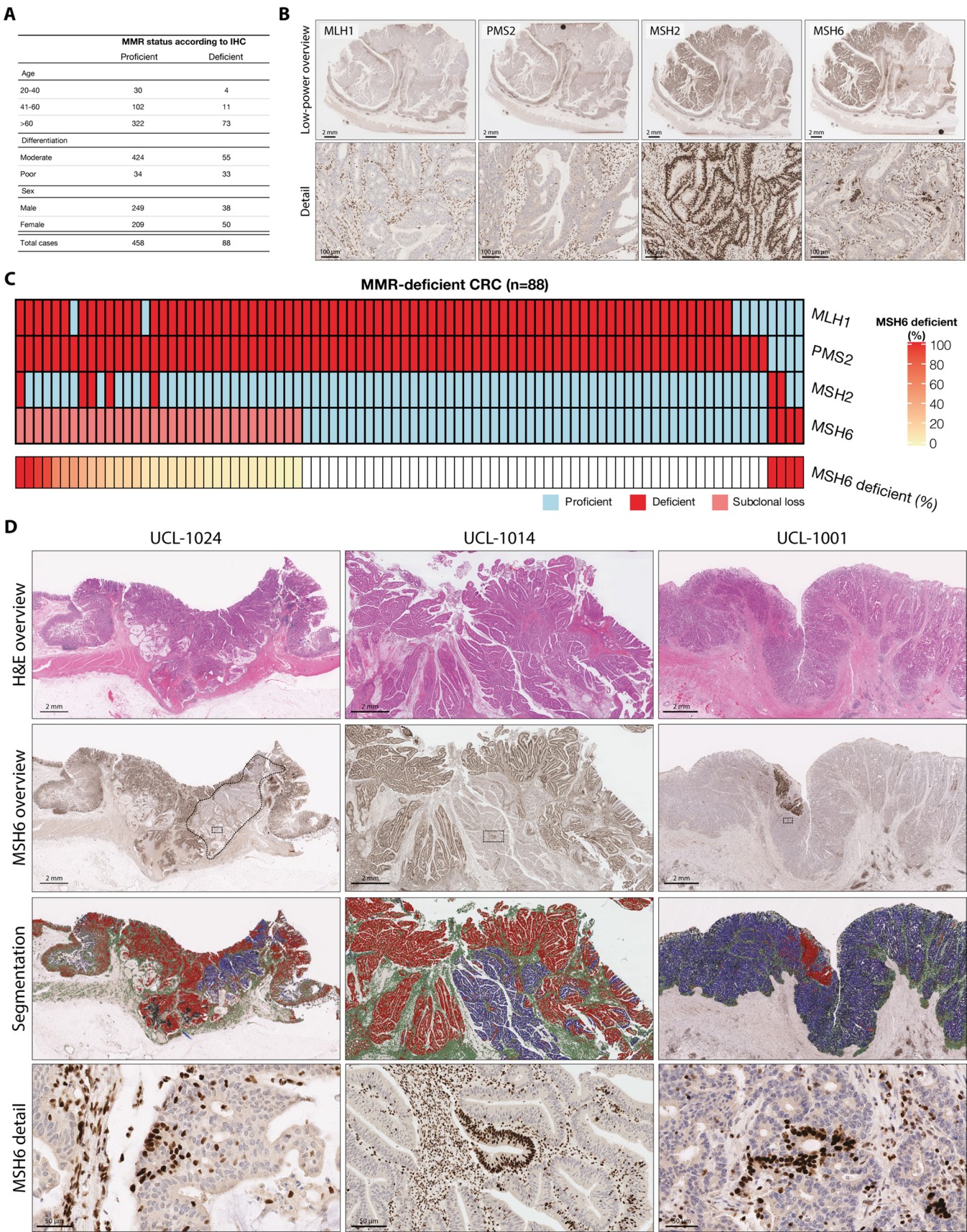

**Extended Data Fig. 2 | See next page for caption.**

**Extended Data Fig. 2 | MSH6 immunolabeling reveals nested proficient reversion subclones within deficient regions.** (**a**) Summary table for the cohort of n = 546 consecutive CRCs used in this study. (**b**) Representative MMR IHC of polypoid tumor with complete loss of MLH1/PMS2 and subclonal loss of MSH6. High-power detail image shows islands of MSH6 reversion (arrowheads, bottom right panel) within MSH6-deficient subclone. (**c**) Breakdown of the pattern of MMR protein loss in n = 88 MMRd tumors. n = 32 tumors had subclonal MSH6 loss, and the estimated percentage of deficient tumor cells observed is displayed in the heatmap. (**d**) n = 3 example tumors with MSH6 subclonal loss. For each tumor H&E staining overview, MSH6 immunohistochemistry overview, IHC segmentation (red is MSH6-proficient, blue is MSH6-deficient and green is stroma) and detail images are shown.

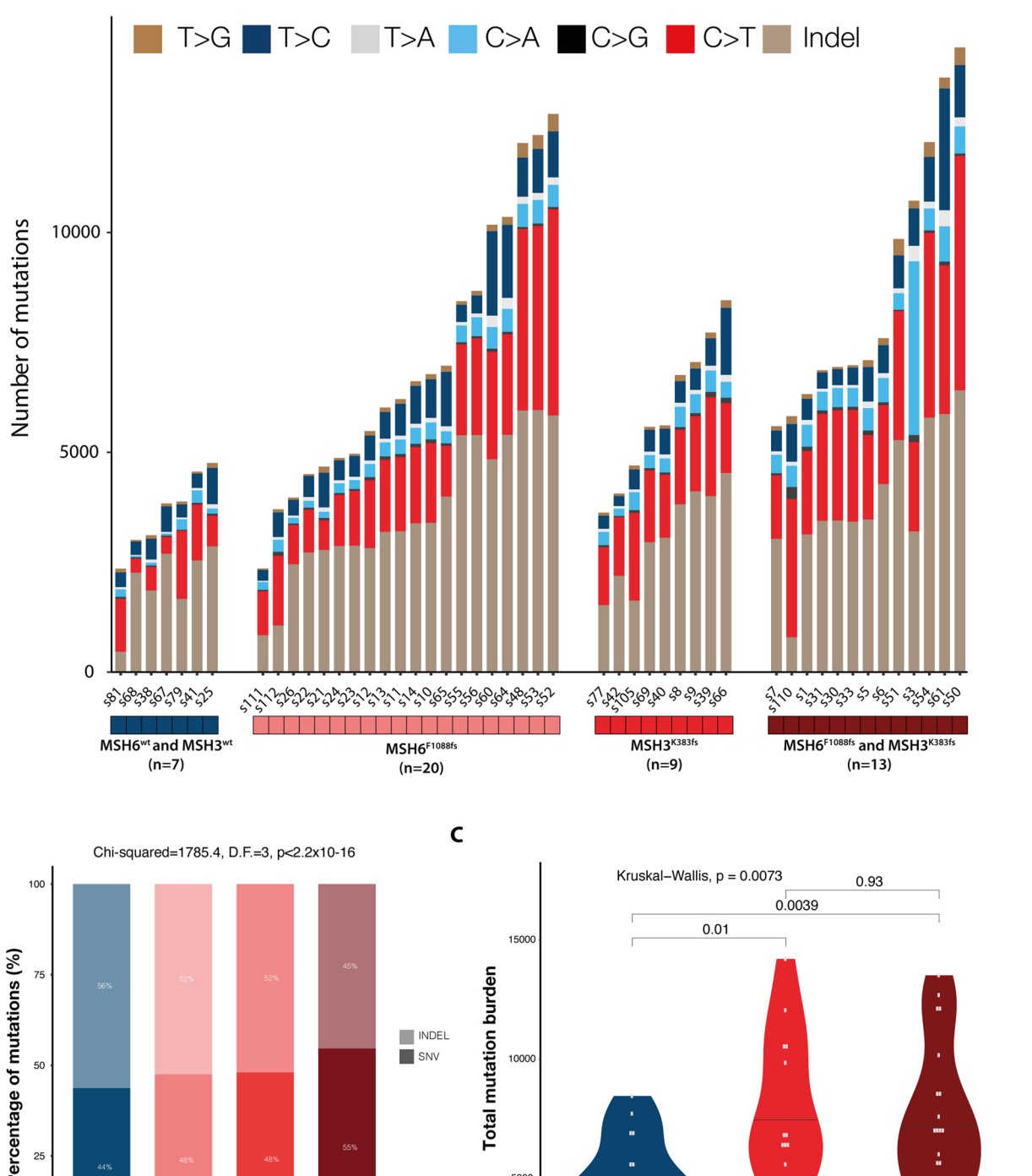

**Extended Data Fig. 3 | Extended data for UCL colorectal cancer cohort mutation bias and burden plots.** (**a**) Mutation bias on a per sample basis with absolute numbers of each mutation type. (**b**) Proportions of SNVs and InDels per mutation group. (**c**) Mutation burden according to MSH6 allelic status.

A

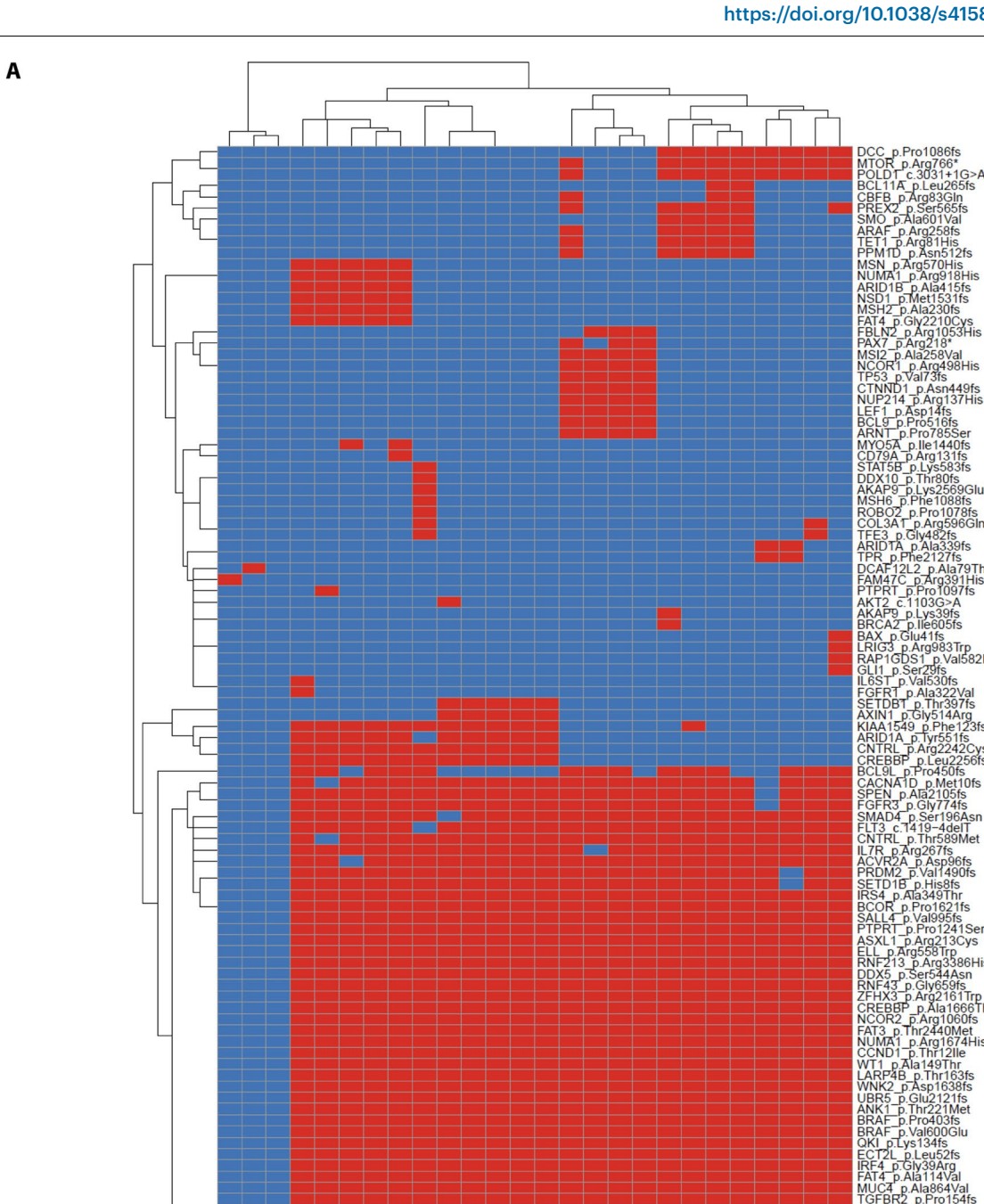

**Extended Data Fig. 4 | Patient-derived organoids mutation clustering. (a)** Heatmap and unsupervised clustering of patient-derived organoids, both bulk and clonally derived samples. The first three columns from the left are normal mucosal samples (labeled 'normal'), and the remainder are tumor samples (labeled 'tumor PDOs').

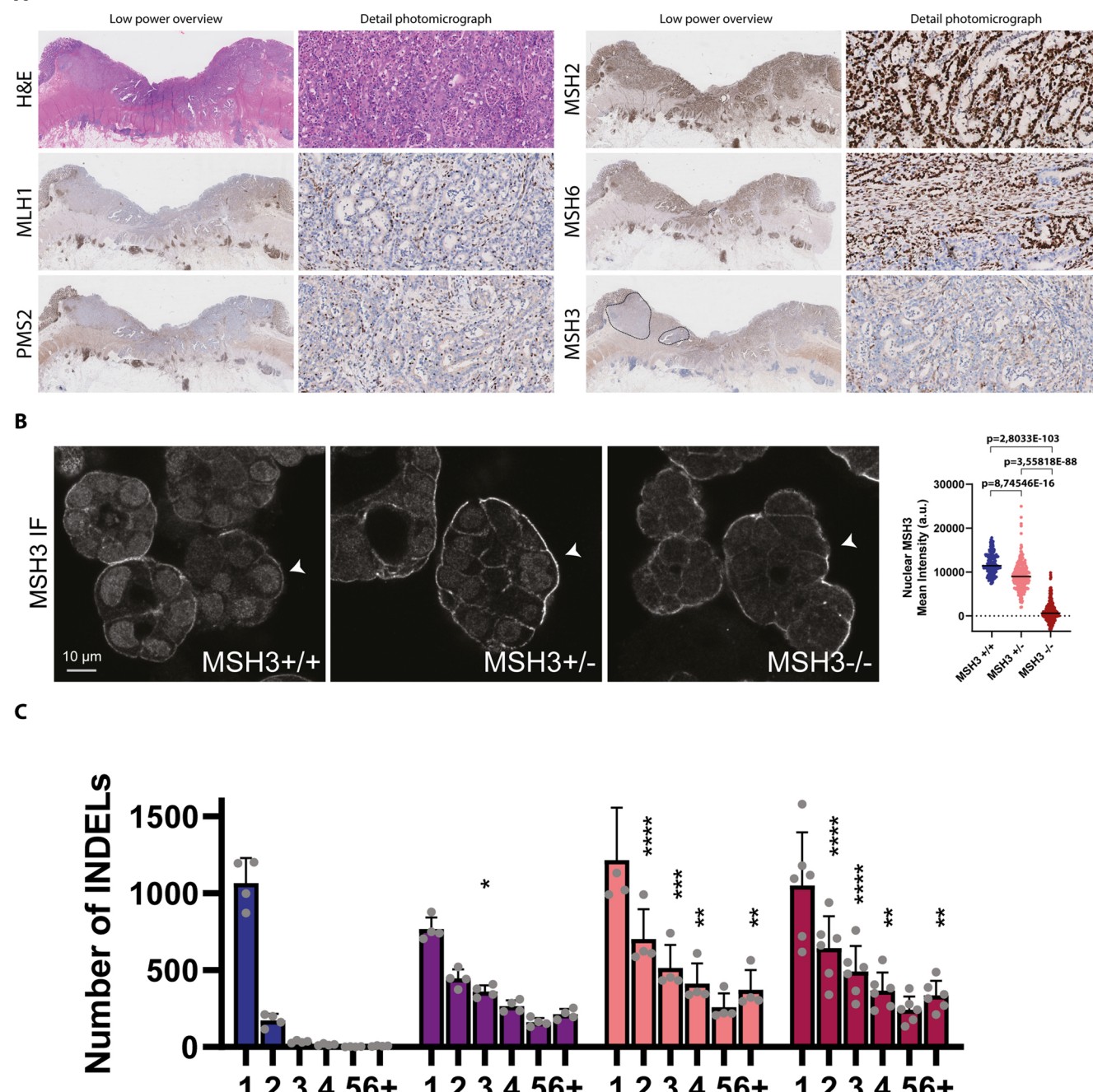

**Extended Data Fig. 5 | Patient-derived organoids validation. (a)** Sections of the mirror block to the tumor block shown in Fig. 4a, inset. Left columns show low power overview, and right columns show high-power detail photomicrographs. Rows show H&E, MLH1, PMS2, MSH2, MSH6 and MSH3. MLH1 and PMS2 show no labeling throughout the tumor bed, as expected. MSH2 is retained throughout. MSH6 shows several small foci of loss (example dashed region shown), and MSH3 shows two larger clones which show complete loss of labeling (dashed regions). PDO genotyping confirms biallelic MSH3 homopolymer InDels, n = 1 for all immunohistochemistry. **(b)** MSH3 immunofluorescence comparing immunolabeling in MLH1$^{-/-}$ MSH3$^{+/+}$, MLH1$^{-/-}$ MSH3$^{+/-}$ and MLH1$^{-/-}$ MSH3$^{-/-}$ PDOs.

For each MSH3 genotype, 230 cells were measured from 30 organoids (2 replicates of 15 organoids/condition per IF round). Nuclear intensity counts show a stepwise decrease (one-way ANOVA, MSH3$^{+/+}$ vs MSH3$^{+/-}$: 8,74546E-16, MSH3$^{+/-}$ vs MSH3$^{-/-}$: 3,55818E-88, and MSH3$^{+/+}$ vs MSH3$^{-/-}$: 2,8033E-103). Nb. Plasma membrane labeling is background. **(c)** InDel mutation bar chart showing frequency of indicated InDels across PDO genotypes. Barplot represents all InDels accumulated over the course of 8 weeks per genotype (MLH1$^{-/-}$, n = 4; MLH1$^{-/-}$ MSH2$^{+/-}$, n = 4; MLH1$^{-/-}$ MSH3$^{+/-}$, n = 4; MLH1$^{-/-}$ MSH3$^{-/-}$ MSH6$^{+/-}$, n = 6). Error bars show SD of the mean, two-way ANOVA (* $p < 0.05$, ** $p < 0.01$, *** $p < 0.001$, **** $p < 0.0001$).

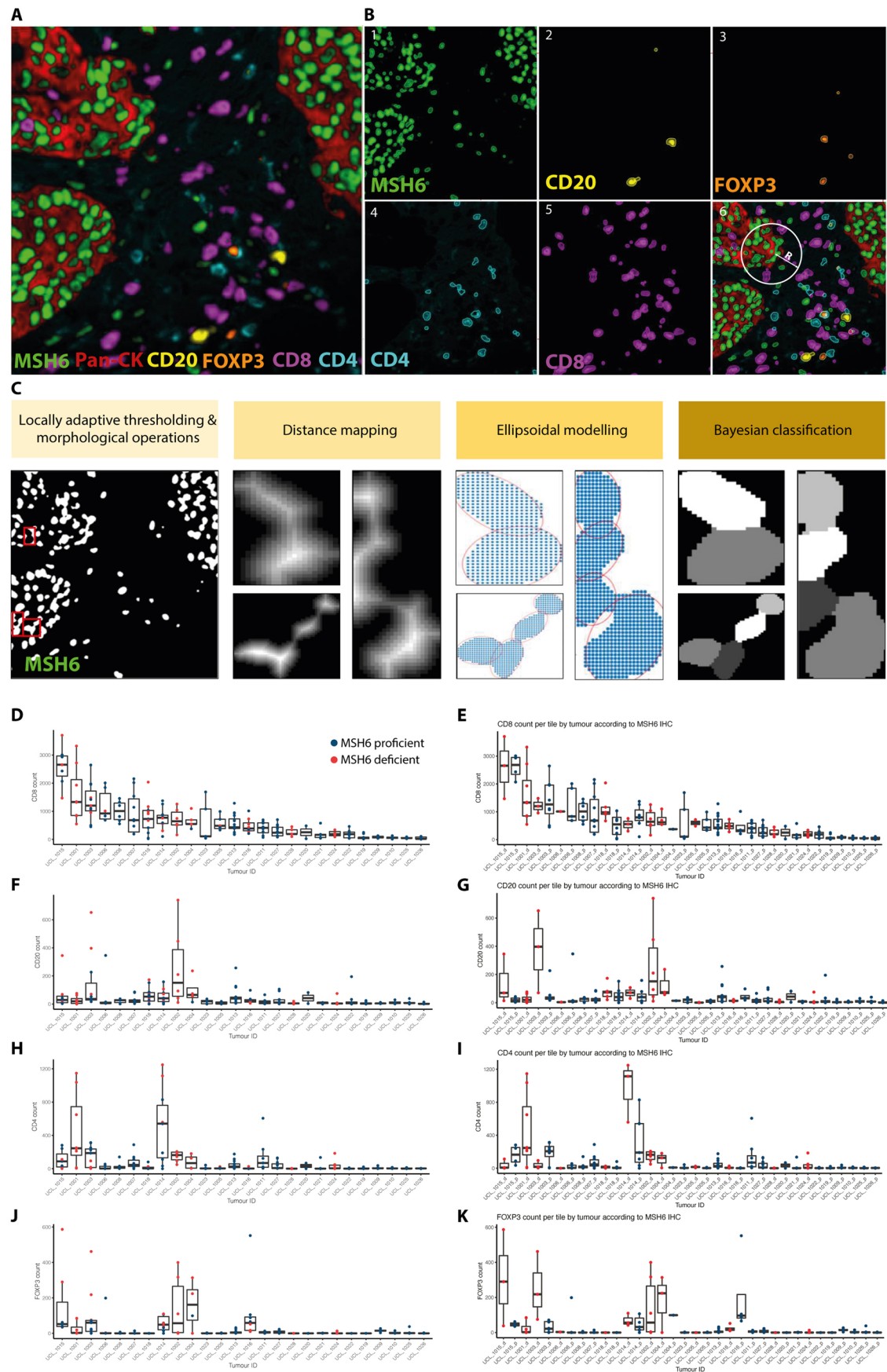

**Extended Data Fig. 6 | See next page for caption.**

**Extended Data Fig. 6 | ORION (FluOResence cell segmentatION) workflow. (a)** Multiplex IF image of example tumor labeled for MSH6, pan-CK, CD8, CD4, CD20 and FOXP3. **(b)** Isolation of MSH6, CD20, FoxP3, CD4 and CD8 spectral signals. **(c)** Example of the ORION main workflow steps. The workflow includes locally adaptive thresholding of isolated spectral signals, estimation of distance maps and local maxima, ellipsoidal modeling of cells and Bayesian classification for the identification of cells. Neighborhood analysis is used for the identification of tumor–immune interactions estimating the number of immune cells within radius R of each MSH6-proficient or deficient tumor cell (shown in **b**_6). ORION workflow was performed in n = 26 tumors producing n = 194 imaged tiles. **(d,f,h,j)** Immune infiltration levels for CD8 (**d**), CD20 (**f**), CD4 (**h**) and FOXP3 (**j**) cells in tumors per imaged tile. **(e,g,i,k)** Data presented with MSH6-proficient and deficient tiles from the same tumor were presented as separate plots side by side. Dots are colored according to MSH6 expression status of tumor in the imaged tile. In total, n = 194 multispectral imaged tiles were assessed across 26 independent tumors. Boxplots: horizontal black line represents median. Lower and upper hinges represent 1st and 3rd quartiles. Lower and upper whiskers extend to values up to 1.5× interquartile range from the hinge.

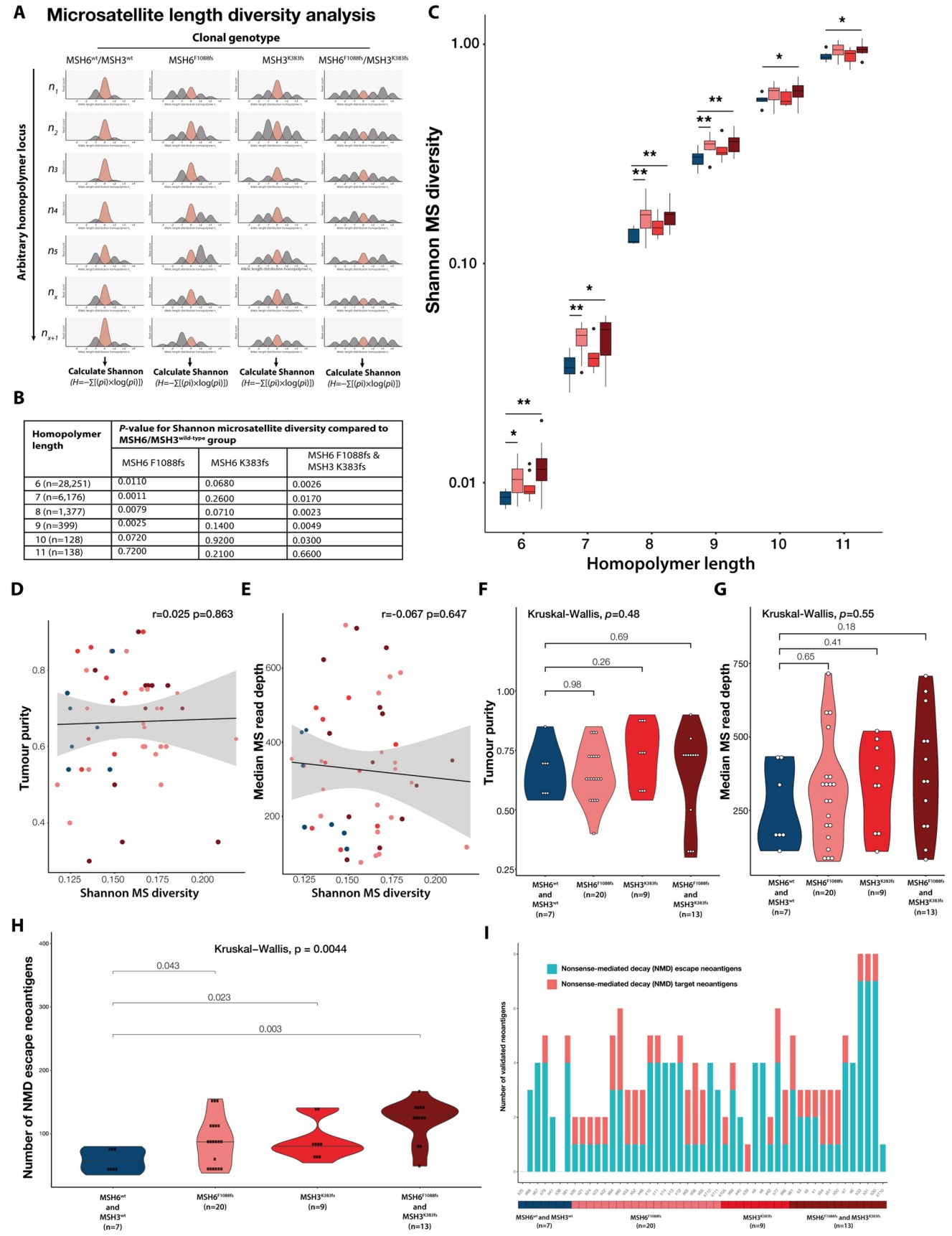

**Extended Data Fig. 7 | See next page for caption.**

**Extended Data Fig. 7 | Microsatellite diversity in tumor samples according to homopolymer length and neoantigen analysis. (a)** Cartoon illustrating microsatellite (MS) length diversity analysis. We hypothesized that increasing clonal mutation burden would be reflected in increasing clonal diversity. In this analysis, microsatellites are used as lineage tags to interrogate population structure. $MSH6^{F1088fs}$ and $MSH3^{K383fs}$ frameshift status from left to right and seven arbitrary homopolymers from top to bottom. Plots show read count for each microsatellite length, beige is the wild-type reference allele and gray are deviations from the reference. Cartoon shows progressive population heterogeneity with $MSH6^{F1088fs}$ and $MSH3^{K383fs}$ frameshifts. **(b,c)** Homopolymer length **(b)** versus Shannon microsatellite diversity **(c)** in samples grouped according to $MSH6$ and $MSH3$ frameshift status. $MSH6^{F1088fs}$ and $MSH3^{K383fs}$ frameshifts result in increased microsatellite diversity, but effect size decreases at longer homopolymer lengths. Asterisks indicate significance according to two-sided Wilcoxon test (* $p < 0.05$, ** $p < 0.005$) **(d,e)** No correlation between Shannon MS diversity and tumor purity or median microsatellite read depth. Gray shaded error band represents 95% confidence interval for linear regression line. **(f,g)** No difference in tumor purity **(f)** or read depth at microsatellite **(g)** sites between samples according to $MSH6/MSH3$ grouping. **(h)** Counts for neoantigens predicted to escape NMD in samples grouped according to $MSH6^{F1088fs}$ and $MSH3^{K383fs}$ mutation status. **(i)** Bar chart shows counts of validated immunogenic neoantigens across groups (see list Supplementary Table 9 for a list of validated neoantigens). Cyan shows validated neoantigens that are predicted to escape nonsense-mediated decay, and pink shows validated neoantigens that are not predicted to escape nonsense-mediated decay.

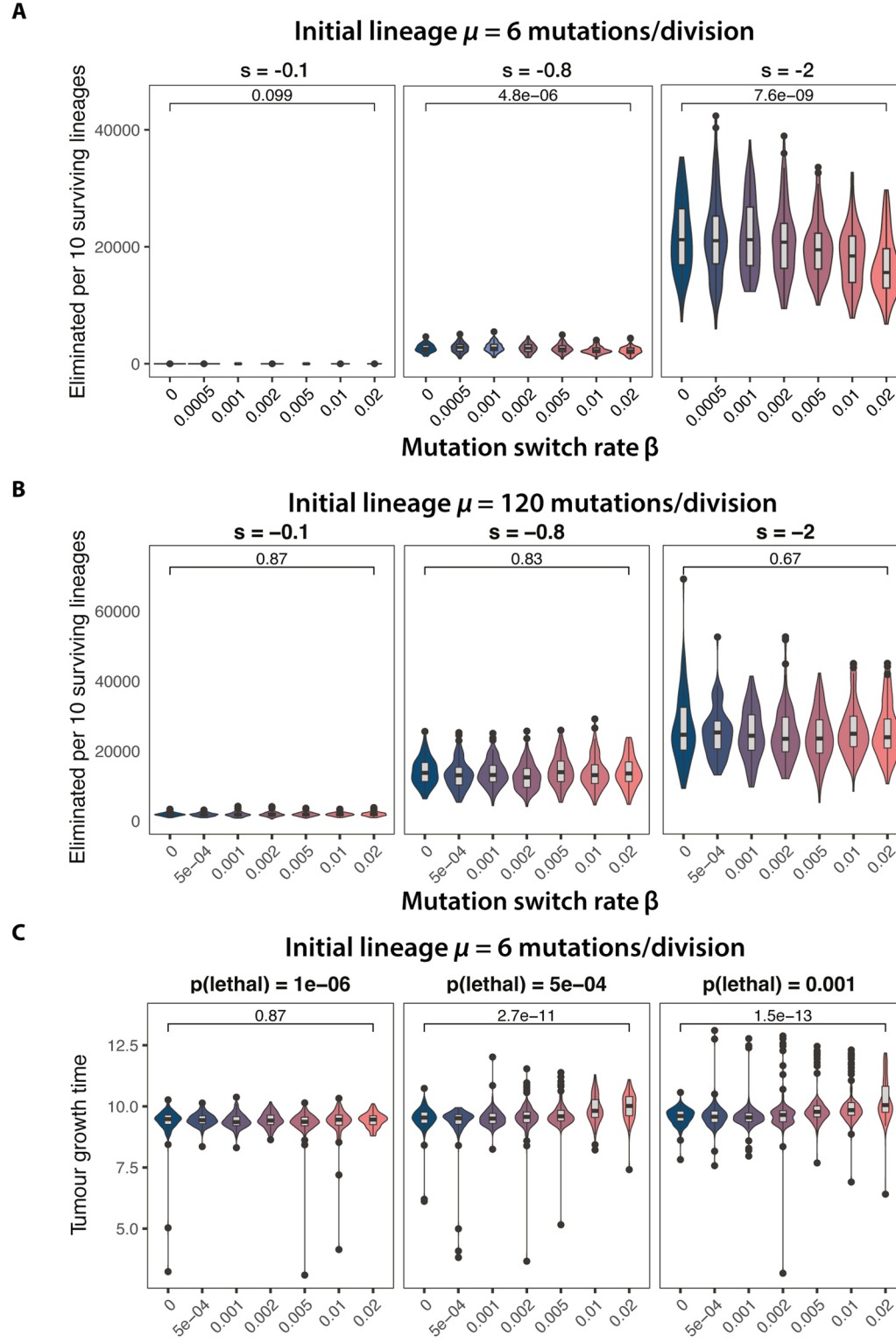

**Extended Data Fig. 8 | Extended data for mathematical model of stochastic mutation rate switching. (a)** Number of lineages eliminated per 10 surviving lineages, computed from n = 100 simulated tumors at increasing immune selection (left to right) and switching rate (x axis). Tumors are initiated with $\mu$ = 6 mutations/division. **(b)** Number of lineages eliminated per 10 surviving lineages, computed from n = 100 simulated tumors at increasing selection rate (left to right) and mutation rate switching rate (x axis). Tumors are initiated with $\mu$ = 120 mutations/division. **(c)** Tumor growth time (in arbitrary units) between establishing immune escape and reaching detectable size, computed from n = 100 hypermutated simulated tumors at increasing lethal mutation rate (left to right) and mutation rate switching rate (x axis). The p-value of a two-sided Wilcoxon test comparing $\beta$ = 0 and $\beta$ = 0.02 is reported on each panel. Boxplots: horizontal black line represents median. Lower and upper hinges represent 1st and 3rd quartiles. Lower and upper whiskers extend to values up to 1.5× interquartile range from the hinge. Outlying points beyond the whisker are plotted individually.

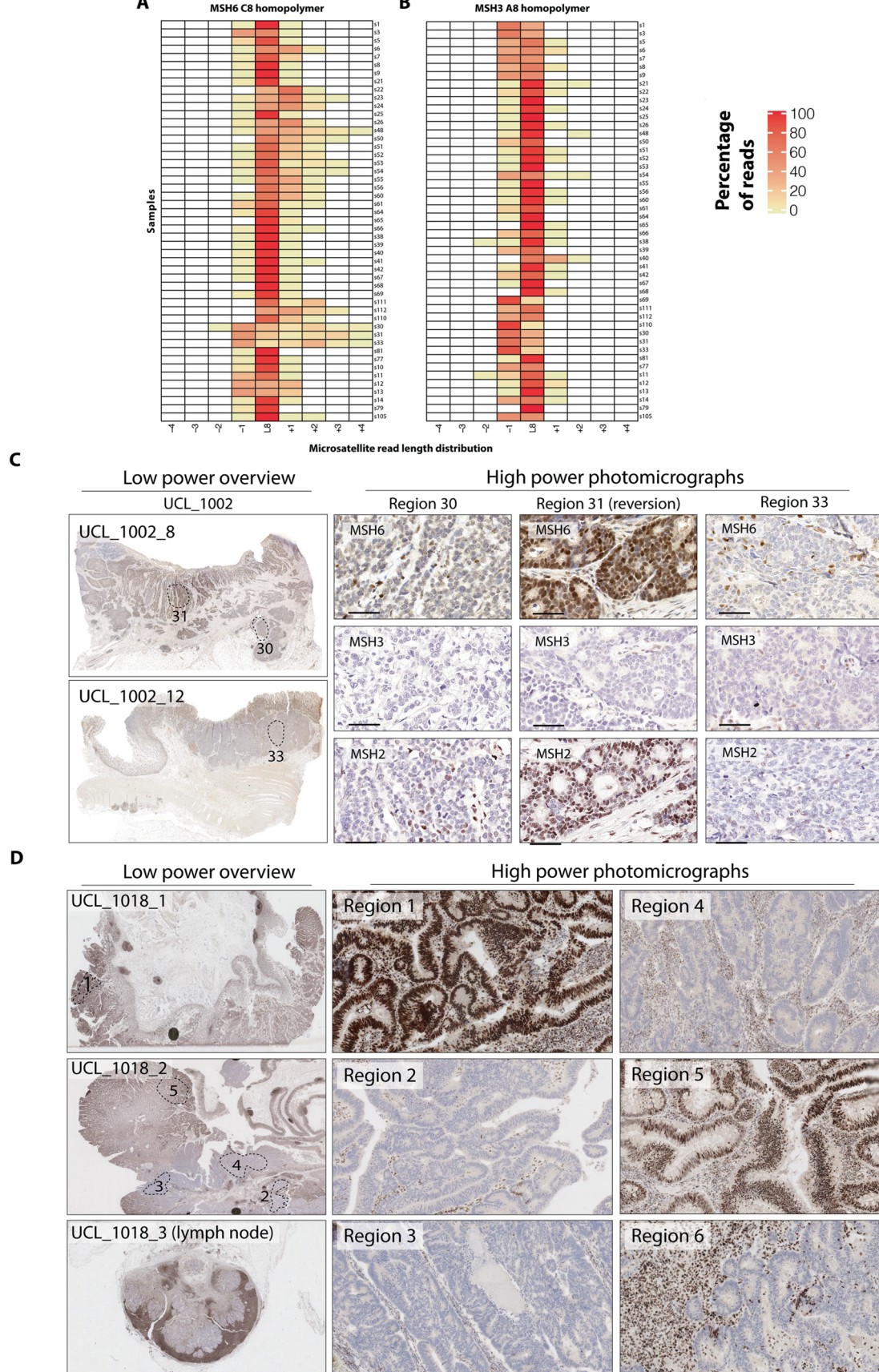

**Extended Data Fig. 9 | Microsatellite length distribution heatmap and MSH6 IHC. (a,b)** Microsatellite length distribution heatmap for the *MSH6* C8 (**a**) and *MSH3* A8 (**b**) homopolymers in all samples from UCL WXS cohort (n = 49). (**c,d**) MSH6 IHC for cases shown in main Fig. 7 (n = 2 independent tumors). (**c**) Tumor UCL_1002 and (**d**) tumor UCL_1018.

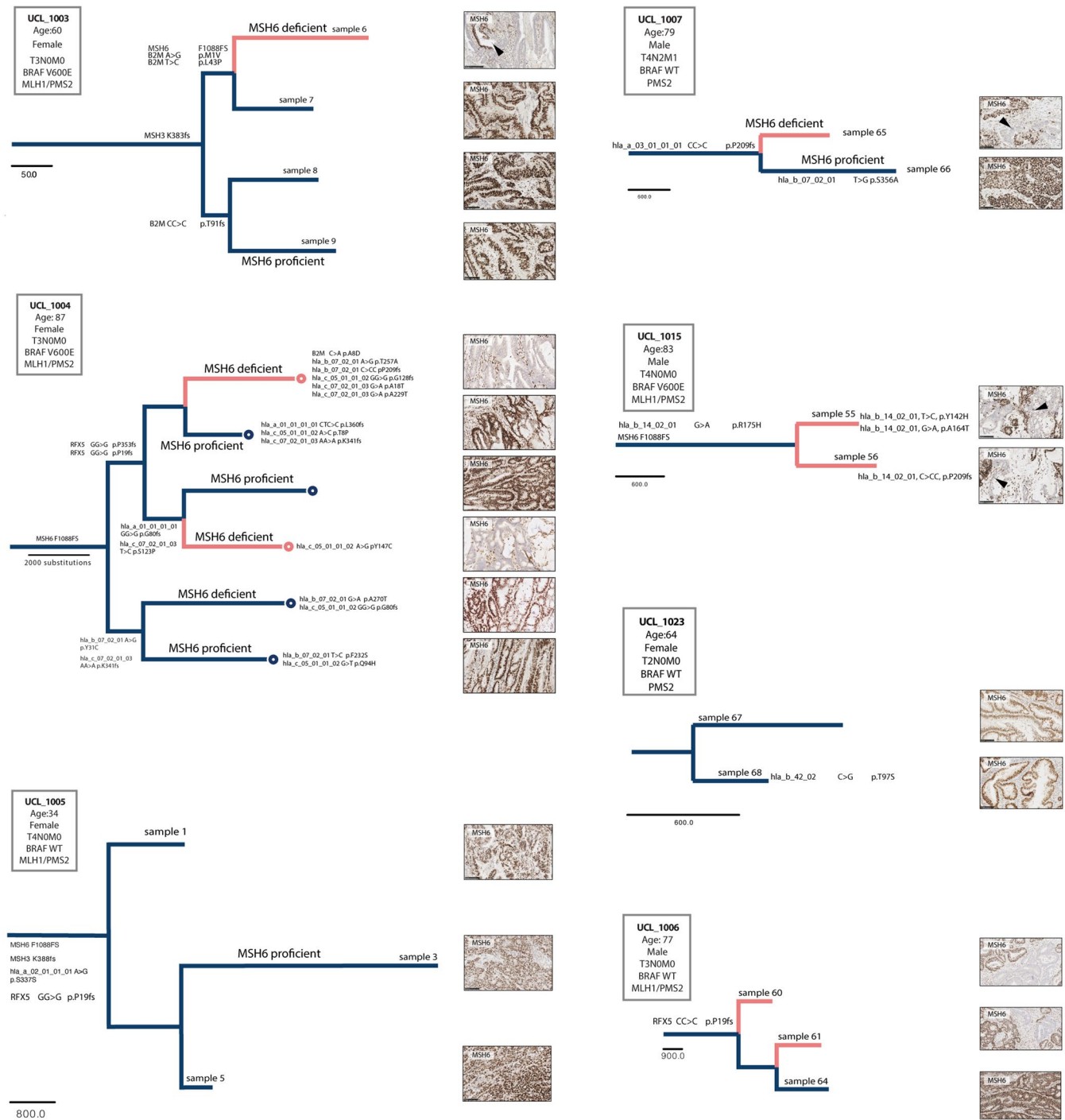

**Extended Data Fig. 10 | Extended data for phylogenetic trees.** Phylogenetic trees not shown in main figure 7 included here. Phylogenetic trees were generated for all *n* = 10 tumors (see also Fig. 7).

# Reporting Summary

## Statistics

For all statistical analyses, confirm that the following items are present in the figure legend, table legend, main text, or Methods section.

| n/a | Confirmed | |
|---|---|---|
| ☐ | ☒ | The exact sample size (*n*) for each experimental group/condition, given as a discrete number and unit of measurement |
| ☐ | ☒ | A statement on whether measurements were taken from distinct samples or whether the same sample was measured repeatedly |
| ☐ | ☒ | The statistical test(s) used AND whether they are one- or two-sided *Only common tests should be described solely by name; describe more complex techniques in the Methods section.* |
| ☐ | ☒ | A description of all covariates tested |
| ☐ | ☒ | A description of any assumptions or corrections, such as tests of normality and adjustment for multiple comparisons |
| ☐ | ☒ | A full description of the statistical parameters including central tendency (e.g. means) or other basic estimates (e.g. regression coefficient) AND variation (e.g. standard deviation) or associated estimates of uncertainty (e.g. confidence intervals) |
| ☐ | ☒ | For null hypothesis testing, the test statistic (e.g. *F*, *t*, *r*) with confidence intervals, effect sizes, degrees of freedom and *P* value noted *Give P values as exact values whenever suitable.* |
| ☒ | ☐ | For Bayesian analysis, information on the choice of priors and Markov chain Monte Carlo settings |
| ☒ | ☐ | For hierarchical and complex designs, identification of the appropriate level for tests and full reporting of outcomes |
| ☒ | ☐ | Estimates of effect sizes (e.g. Cohen's *d*, Pearson's *r*), indicating how they were calculated |

*Our web collection on statistics for biologists contains articles on many of the points above.*

## Software and code

Policy information about availability of computer code

| Data collection | Alignment of FASTQ sequencing files: BWA-mem (version 0.7.7) (http://bio-bwa.sourceforge.net/) |
|---|---|
| | Sorting and indexing of BAM files: SAMtools (http://www.htslib.org/) |
| | Marking duplicate reads and local INDEL realignment: PICARD tools, GATK (version 2.8) (https://broadinstitute.github.io/picard/) |
| | Detection of somatic variants: MuTect (version 1.1.4) (https://software.broadinstitute.org/cancer/cga/mutect_download) |
| | Detection of INDELs: VarScan2 (version 2.3.8) (http://varscan.sourceforge.net/) |
| | Detections of INDELs: SCALPEL (http://scalpel.sourceforge.net/index.html) |
| | Annotation of variants: ANNOVAR (https://annovar.openbioinformatics.org/en/latest/user-guide/download/) |
| | Tumour purity, ploidy and copy number estimation: Sequenza (https://cran.r-project.org/web/packages/sequenza/vignettes/sequenza.html) |
| | Identification of exonic homopolymers: SciRoKo (version 3.4) (https://kofler.or.at/bioinformatics/SciRoKo/Download.html) |
| | HLA haplotyping and mutation calling: Polysolver (Version 4) (https://software.broadinstitute.org/cancer/cga/polysolver) |
| Data analysis | Tumour phylogenetic reconstruction: Paup (https://paup.phylosolutions.com/) |
| | Generation of phylogenetic trees: Figtree (version 1.4.4) (https://github.com/rambaut/figtree/releases) |
| | Neoantigen calling: Neopredpipe (https://github.com/MathOnco/NeoPredPipe) |
| | Hompolymer read length distribution analysis: MSIsensor (version 0.6) (https://github.com/ding-lab/msisensor/blob/master/README_msisensor.md) |
| | Mutation signature analysis: SigProfiler (Version 3.1) (https://github.com/AlexandrovLab) |
| | Immune dN/dS analysis: SOPRANO (https://github.com/luisgls/SOPRANO) |
| | Subclonal deconvolution and mutation rate analysis: MOBSTER (https://github.com/caravagnalab/mobster) |

For manuscripts utilizing custom algorithms or software that are central to the research but not yet described in published literature, software must be made available to editors and reviewers. We strongly encourage code deposition in a community repository (e.g. GitHub). See the Nature Portfolio guidelines for submitting code & software for further information.

# Data

Policy information about availability of data

All manuscripts must include a data availability statement. This statement should provide the following information, where applicable:

- Accession codes, unique identifiers, or web links for publicly available datasets
- A description of any restrictions on data availability
- For clinical datasets or third party data, please ensure that the statement adheres to our policy

1. Whole genome sequencing data from the Genomics England colorectal cancer dataset can be accessed by application through the Genomics England Clinical Interpretation Partnership (https://www.genomicsengland.co.uk/about-gecip/joining-researchcommunity/).
2. Variant calls from the TCGA whole exome sequencing dataset can be retrieved from the Genomics Data Commons (GDC) website (https://portal.gdc.cancer.gov/).
3. Whole exome sequencing data from the UCL colorectal cancer cohort has been deposited to the European Genome Phenome archive under accession code:

# Research involving human participants, their data, or biological material

Policy information about studies with human participants or human data. See also policy information about sex, gender (identity/presentation), and sexual orientation and race, ethnicity and racism.

| Reporting on sex and gender | Reported data in line with UK HTA data and tissue privacy laws |
|---|---|
| Reporting on race, ethnicity, or other socially relevant groupings | As above |
| Population characteristics | Populations characteristics in this study: Mismatch repair deficient colorectal cancer Adult (age >18) |
| Recruitment | UCL WXS cohort: Participants were recruited from the UCL/UCLH biobank of health and disease archive according to Institutional Review Board approved protocol (Project Reference Number NC21.18). No bias to report. Genomics England WGS cohort: The participants were recruited across 13 NHS Genomic Medicine Centres and written informed consent was obtained from the participants. No bias to report. |
| Ethics oversight | Approval provided by UCL/UCLH Biobank of health and disease ethics review committee (Project Reference Number NC21.18) |

Note that full information on the approval of the study protocol must also be provided in the manuscript.

# Field-specific reporting

Please select the one below that is the best fit for your research. If you are not sure, read the appropriate sections before making your selection.

☒ Life sciences  ☐ Behavioural & social sciences  ☐ Ecological, evolutionary & environmental sciences

For a reference copy of the document with all sections, see nature.com/documents/nr-reporting-summary-flat.pdf

# Life sciences study design

All studies must disclose on these points even when the disclosure is negative.

| Sample size | In the Genomics England WGS cohort, we analysed all available colorectal tumours identified as microsatellite instable (MSI) resulting in a total of n=217 tumours. In the TCGA cohort, we analysed microsatellite instable (MSI) tumours in the colorectal, uterine, stomach and esophageal datasets resulting in a total of n=181 tumours. In the UCL WXS cohort, we identified all mismatch repair deficient (MMRd) colorectal tumours diagnosed between 2014 to 2018 resulting in a total of n=88 tumours with loss of at least one MMR protein. After assessing for tissue quality and tumour block availability, we identified n=11/40 (28%) tumours with subclonal loss of MSH6 on a background of clonal MLH1/PMS2 loss. A stage and age matched cohort of n=11 MH1/PMS2 deficient tumours without evidence of MSH6 loss was used as the control group. Due to the limited sample availability no sample size calculation was performed upfront. |
|---|---|
| Data exclusions | In the Genomics England cohort we excluded n=6 tumours with discernible pathogenic POLE or POLD1 mutations. In the TCGA cohort we excluded n=9 tumours with discernible pathogenic POLE or POLD1 mutations. |
| Replication | For analyses of human tumour datasets, replication is not relevant as these specimens cannot be replicated. |

| Replication | For the patient derived organoid (PDO) analysis a minimum of four independent pairs of parent and daughter PDO lineages were assessed for each of the four MMR genotypes. |
| --- | --- |
| Randomization | There was no therapy or intervention in this study, therefore randomisation is not relevant. |
| Blinding | There was no therapy or intervention in this study, therefore randomisation is not relevant. |

# Reporting for specific materials, systems and methods

We require information from authors about some types of materials, experimental systems and methods used in many studies. Here, indicate whether each material, system or method listed is relevant to your study. If you are not sure if a list item applies to your research, read the appropriate section before selecting a response.

## Materials & experimental systems

| n/a | Involved in the study |
| --- | --- |
| ☐ | ☒ Antibodies |
| ☐ | ☒ Eukaryotic cell lines |
| ☒ | ☐ Palaeontology and archaeology |
| ☒ | ☐ Animals and other organisms |
| ☒ | ☐ Clinical data |
| ☒ | ☐ Dual use research of concern |
| ☒ | ☐ Plants |

## Methods

| n/a | Involved in the study |
| --- | --- |
| ☒ | ☐ ChIP-seq |
| ☒ | ☐ Flow cytometry |
| ☒ | ☐ MRI-based neuroimaging |

## Antibodies

| Antibodies used | MSH6: Clone: EP49, Company: Agilent, Catalog: M364601-2<br>MLH1: Clone: ES05, Company: Agilent, Catalog number: M364001-2<br>PMS2 Clone: A16-4 Company: BD Sciences, Catalog number: 556415<br>MSH2 Clone: FE11 Company: Agilent, Catalog number: M363929-2<br>MSH3 Clone: 611390 Company: BD Sciences, Catalog number: 611390<br>CD20: Clone:L26, Company: Agilent, Catalog number: M0755<br>FOXP3: Clone: D608R, Company: Cell Signalling Technology, Catalog number: 12653<br>CD4: Clone: 4B12, Company: Agilent, Catalog number: M7310<br>PANCK: Clone: AE1/3, Company: Agilent, Catalog number: M3515<br>CD8: Clone: 4B11, Company: Agilent, Catalog number: M7103 |
| --- | --- |
| Validation | All antibodies were validated using positive and negative control tissue. For the MMR proteins MLH1, PMS2, MSH2 and MSH6, these have been extensively validated by HSL-AD which runs the clinical diagnostic laboratory at our centre and we used the same antibody clones and IHC conditions as validated for clinical samples. For immune cell markers CD8, CD4, FOXP3 and CD20, these were validated using lymph nodes tissue and normal colonic mucosa. |

## Eukaryotic cell lines

Policy information about cell lines and Sex and Gender in Research

| Cell line source(s) | Patient derived (identity not disclosed) |
| --- | --- |
| Authentication | N/A |
| Mycoplasma contamination | Routinely tested |
| Commonly misidentified lines<br>(See ICLAC register) | N/A |

# Plants

Seed stocks

*Report on the source of all seed stocks or other plant material used. If applicable, state the seed stock centre and catalogue number. If plant specimens were collected from the field, describe the collection location, date and sampling procedures.*

Novel plant genotypes

*Describe the methods by which all novel plant genotypes were produced. This includes those generated by transgenic approaches, gene editing, chemical/radiation-based mutagenesis and hybridization. For transgenic lines, describe the transformation method, the number of independent lines analyzed and the generation upon which experiments were performed. For gene-edited lines, describe the editor used, the endogenous sequence targeted for editing, the targeting guide RNA sequence (if applicable) and how the editor was applied.*

Authentication

*Describe any authentication procedures for each seed stock used or novel genotype generated. Describe any experiments used to assess the effect of a mutation and, where applicable, how potential secondary effects (e.g. second site T-DNA insertions, mosiacism, off-target gene editing) were examined.*

