## [Peer Review File · Nature Genetics]

Peer Review Information

Manuscript Title: Homopolymer switches mediate adaptive mutability in mismatch repair-deficient colorectal cancer

Corresponding author name(s): Dr Marnix Jansen, Dr Hugo (JG) Snippert

Reviewer Comments & Decisions:

Decision Letter, initial version:
--

4th Apr 2022

Dear Dr Jansen,

Your Article, "Mutation rate evolution drives immune escape in mismatch repair deficient cancer" has now been seen by 3 referees. You will see from their comments copied below that while they find your work of considerable potential interest, they have raised quite substantial concerns that must be addressed. In light of these comments, we cannot accept the manuscript for publication, but would be very interested in considering a revised version that addresses these serious concerns.

We hope you will find the referees' comments useful as you decide how to proceed. If you wish to submit a substantially revised manuscript, please bear in mind that we will be reluctant to approach the referees again in the absence of major revisions.

To guide the scope of the revisions, the editors discuss the referee reports in detail within the team, including with the chief editor, with a view to identifying key priorities that should be addressed in revision and sometimes overruling referee requests that are deemed beyond the scope of the current study. In this case, the reviewers are supportive of the goals of the work, but the main limitation is that lack of more direct evidence that the mutations in MSH3/6 are driving hypermutation, immune sensitivity, and immune escape. They have made suggestions for experiments aimed at fortifying these aspects of the work and while you are strongly encouraged to follow their guidance, we will leave it to you decide how best to address these points experimentally. I would say that the same applies to Reviewer #2's comments regarding the need for single cell analysis. While we agree that excluding the stroma as a potential source of MSH3/6 is important, you are free to demonstrate this robustly using other methods.

All other comments should be addressed; through new analyses or experimentation where possible, or textually where appropriate. Please do not hesitate to get in touch if you would like to discuss these issues further.

If you choose to revise your manuscript taking into account all reviewer and editor comments, please highlight all changes in the manuscript text file. At this stage we will need you to upload a copy of the manuscript in MS Word .docx or similar editable format.

*2) If you have not done so already please begin to revise your manuscript so that it conforms to our Article format instructions, available here. Refer also to any guidelines provided in this letter.

Please be aware of our guidelines on digital image standards.

[redacted]

If you wish to submit a suitably revised manuscript we would hope to receive it within 6 months. If you cannot send it within this time, please let us know. We will be happy to consider your revision so long as nothing similar has been accepted for publication at Nature Genetics or published elsewhere. Should your manuscript be substantially delayed without notifying us in advance and your article is eventually published, the received date would be that of the revised, not the original, version.

Nature Genetics is committed to improving transparency in authorship. As part of our efforts in this direction, we are now requesting that all authors identified as 'corresponding author' on published papers create and link their Open Researcher and Contributor Identifier (ORCID) with their account on the Manuscript Tracking System (MTS), prior to acceptance. ORCID helps the scientific community

achieve unambiguous attribution of all scholarly contributions. You can create and link your ORCID from the home page of the MTS by clicking on 'Modify my Springer Nature account'. For more information please visit please visit www.springernature.com/orcid.

Thank you for the opportunity to review your work.

Sincerely,

Safia Danovi
Editor
Nature Genetics

Referee expertise:

Referee #1: hypermutator tumours

Referee #2: immunogenomics, immune escape, tumour evolution

Referee #3: colorectal cancer geneomics incl. hereditary syndromes, pre-cancer

Reviewers' Comments:

Reviewer #1:

Remarks to the Author:

The authors describe association between MSH6 and MSH3 secondary mutations in dMMR tumors and their association with increased mutational burden, areas (clones) where that occurs in the tumors, increased immune infiltrates in these areas and prove that using multiple platforms including specific signatures and phylogenetic trees. The authors should be commended for this work. However, these data do not support the major claims of the authors throughout the paper.

Major general concerns:

1. Although the authors show very nicely throughout the paper that the MSH6/3 indels are associated with hypermutation, and increased immune infiltrates, this does not prove the claim that this drives either immune sensitivity or escape. For that, it would be important to see of these cancers, which responded to immune checkpoint blockade, and in failing tumors, which were the mutations which were seen at recurrence/progression. For example, disappearance of MSH6/3 sub clones in the relapsed tumor may suggest that indeed, the immune infiltrates are causing loss of these ultra-hypermutant clones or vice versa.
2. Similarly, the evolution of clones in the mathematical model and the analysis can be just a matter of tumor fitness and additional mutations causing disappearance of these clones. The cause and effects with immune surveillance is not clear.
3. I would suggest to tone down the claims through the paper and focus on the tumor mutational burden and subclonal fitness and evolution.

Specific comments:

1. The MS-indel mutations in MSH6/3 are presented as drivers and the data in Figure 1 (and other

figures after that) just suggest association between these alterations and increased TMB. Since MSI accumulate over time in a stochastic fashion in MMR tumors, this can be just a marker of ongoing mutation accumulation and not a driver of increased mutagenesis. Similarly, other associations of mutations with neoantigens are known and not specific. These do not prove the driving forces of these mutations. Lastly, it is unclear why MS-ndel mutation in MSH6 is a stronger driver of dMMR than missense mutation in the same gene.

2. The authors should be commended for the great work using LSM for clonal analysis. Can the authors look at mutations in MSH6/3 in these areas as driving loss of expression?

3. One should not over interpret the data in Fig 1G-H. These are not the same as bacterial mutations but rather association of MSH6 loss when MS-indel occurs in the exome.

4. Can the authors compare their signatures A, B and C with the known signatures which are common in dMMR which are Cosmic 6, 14,15 and the ones which are more specific to MUTS alfa which are 1 and 5? This may contribute to their claims in Figure 2.

5. Figures 5 and 6 are repetitive to the claims in the initial figures and do not add to the hypothesis that immune evasion is the main reason for the intratumoral heterogeneity in MSH6/3. Wonder if these should come before Figure 3 as the immune microenvironment can come as a final suggestion of immune involvement in clonal expansion of these cancers.

6. As such, Figure 4 can go to the supplements.

Reviewer #2:

Remarks to the Author:

Review of Kayhanian et al.

MUTATION RATE EVOLUTION DRIVES IMMUNE ESCAPE IN MISMATCH REPAIR-DEFICIENT CANCER

This study examines the contribution of mutation rate evolution to immune escape in mismatch repair deficient colorectal cancer. The authors mapped the clonal structure of MMRD CRC and show that MMRD mutability coevolves with neoantigen selection to drive diversification and immune escape. They find that MSI modulates subclonal DNA repair at MSH6 and MSH3. They find that MMRD helps generate intraheterogeneity by adapting mutation rate to immune selection. This is a nice study which demonstrates the importance of neoantigens and immune regulation in MMR tumor growth and provides a mechanism for regulated hypermutability. Specific comments follow.

Comments:

1. The application of conserved models for regulated hypermutability in cancer cells is a nice approach.

2. Homopolymer sequence is notoriously difficult to deconvolute using bulk genomic sequencing despite purity corrections and may be confounded by lack of power due to depth of sequencing. The authors need to provide power calculations to show they have sufficient power to detect the range of subclonal alleles.

3. in Fig. 1 bulk data, it is not clear that the authors controlled for phasing of alleles. Were both alleles of MSH3 and 6 affected? One would need to show that both alleles would be affected if the association between MSH3 and 6 changes are causative for mutation generation.

4. Fig 1G. The authors claim to have found restored MSH6 function. This could come from stromal cells. The authors have not ruled this out. Single cell spatial sequencing is needed to determine whether this claim is valid.

5. Developing ORION is commendable however the authors should validate the generalizability of the interpretation by using other spatial workflows. For example, “adaptive thresholding” and ellipsoidal modeling has a tendency to introduce biases.
6. The neoantigen burden estimation in Fig. 3 is incomplete. MSI functions by introducing indels, which can generate stop codons that cause nonsense mediated decay. The investigators need to take the into account. Furthermore, there is no validation of any of the neoantigen findings. The authors should validate some of the candidate neoantigens. Dextramer staining is a reasonable approach for this.
7. It is unclear whether the immune infiltrate is near the MSH loss subclones are functional. The authors should use functional markers for T cell activity to provide this information.
8. none of the data in fig 3k-L account for allelic zygosity. One would assume you need both alleles inactivated. Is this true? Without that, the data is not interpretable.
9. Fig 3N-P. the interpretation here is dependent on arbitrary definition of what is subclonal or clonal NA.
10. The analysis ignores immune evasion mechanisms like anti-PD1. Do the subclones have differential expression of immune checkpoints? This would surely affect immunosurveillance.
11. Fig. 5. The authors interpret the data as MSH 6 and 3 homopolymer diversity as driving immune escape solutions but the latter is not well supported. Only two cases are shown that support this contention. Furthermore, the nonsyn mut in RFX5 is of uncertain consequence. Is this a functional mutation? Data here is lacking. How many reverter lineages has immune escape mechanisms?

Reviewer #3:

Remarks to the Author:

The manuscript titled ‘Mutation rate evolution drives immune escape in mismatch repair-deficient cancer’ by Kayhanian/Jansen et al is an original contribution submitted to Nature Genetics for consideration. In brief, Authors have performed detailed mutational analysis from WGS data in MMRd colorectal cancers from the Genomic England cohort (N=217) and profiled frameshift mutations (indels) impacting coding microsatellites. Two specific somatic mutations within coding microsatellite tracts located in MSH3 (A8) and MSH6 (C8) have been correlated with increases in mutational rate, indels, and therefore neoantigen generation. The vast majority of MMRd CRC demonstrate the acquisition of somatic mutations in either of these two microsatellite tracts or even simultaneously. An independent validation of the genomics results has been performed in a limited number of TCGA samples of other tumors known to display MMRd. Then, Authors used an institutional cohort of CRC cases to perform a detailed topographic multiplex protein expression analysis of MSH6 and immune cell markers via IF coupled with specific MS analysis of the A8 and C8 repeats. Subsequent genomic analysis of subclones captured by LCM and stratified by somatic MSH6/MSH3 frameshift mutational status showed specific mutational signatures and accumulation of C>T changes. Protein expression analysis demonstrated that variegated MSH6 loss corresponds with the presence of indels in the C8 tract. In addition, MMRd CRC with MSH6 loss present with active immune infiltration observed via multiplex-IF for several cell types (CD4 and CD8 T cells, Tregs and B cells) and neoantigen accumulation. Detailed clonal analysis suggested that subclones switch on/off the immune response by selecting these clones which could contribute to the development of immune tolerance in general. This manuscript includes sophisticated mathematical modeling on growth advantage depending on the acquisition of these somatic mutations (switch from MMRd to p) and their contribution to neoantigen/immune activation.

Overall, this is a well-executed work that develops further the pre-existing concept that secondary somatic mutations targeting coding microsatellites play an important role in MMRd carcinogenesis. Authors have done an impeccable job in two main areas: systems biology and molecular pathology analysis. However, the data presented here is mainly associative and does not provide final functional proof that these two specific mutations in MSH6 and MSH3 are responsible for the mutation accumulation and the immune switch, although it is more than plausible that this is the case, and it constitutes a solid hypothesis to be conclusively proved. Therefore, a request for addition to this manuscript (major critique) will be to provide proof using an in vitro (an ideally ex vivo) systems model that these two mutations contribute to accumulation of somatic indels. The most valid platform to perform this would be patient-derived organoids of a MMRd clone (it could be MLH1/BRAF-mt clone) that went to be CRISPR'ed with the two mutations independently and then combined. Genomic studies should prove accumulation of indel mutations and IHC of the organoids should prove the somatic inactivation of both proteins.

Moderate critiques in the execution and presentation of this work are the following:

*Genomic analysis highlight both the contribution to mutation accumulation by both MSH6 and MSH3 frameshift mutations. However, it seems like the Authors abandon later the MSH3 frameshift mutation for their subsequent molecular pathology analysis, thus diminishing the value of the MSH3 somatic inactivation. Authors should provide the alternative set of experiments presented in Figure 3 for clones stratified by MSH3 status.

*Connected to the previous point, do Authors observe a similar variegated pattern towards re-switch of MSH3 clones as they do for MSH6 (similar to what is displayed in Fig 1G-H for MSH6)?

*Clonality analysis (Fig 5). Here, Authors resume the assessment of both MSH3 and MSH6 from a functional and genomics standpoint by pairing IHC with microsatellite length analysis (using MSIsensor outputs). However, it seems that the clonal analysis is only driven by the MSH6 status, and it is not clear the contribution of the MSH3 data. In addition, the statement in the texts highlights that there is always loss of MSH3, which is evident in the IHC. However, while some of the clones match the loss of MSH3 with presence of indels in the A8 repeat, there are clones that show no instability with loss of the MSH3 expression, which is difficult to understand (e.g. see two MSH6 deficient clones in case UCL1016). Also, the MSH2 staining needs to be displayed back-to-back in the main figure and considered for the clonal analysis. The main question here is, are there clones within these tumors that are fully proficient for MSH6/MSH3 and MSH2, thus proving the restoration of the MMR mechanism as suggested in the text of the paper? The current version of the results is neither clear nor backing this notion.

Minor critiques to bring to Authors' attention are the following:

*The title of the paper is generic and addressing MMRd tumors in general. The data from TCGA does not offer separate analyses for tumors of different origin. There is evidence in the literature that there is some degree of exclusivity of certain microsatellite mutations for certain types of tissue (Colorectum vs Endometrium versus Urothelium) See <https://www.nature.com/articles/nm.4191> to back this statement and as an important reference to be incorporated. Perhaps, the title needs to be more specific towards CRC.

*The previous reference proves that there is substantial genomic information available within TCGA to run a separate analysis per tumor site to prove that the same FS mutations occur and are responsible for the increase in mutation burden in other MMR-deficient tumor types. This will ultimately prove that the mechanism described in this paper is not exclusive of colorectal cancers and generic for all MMRd tumors.

*Clinical details on the samples analyzed are insufficient for all cohorts (GEL, TCGA and UCL). Tables

need to be more specific on the ages, gender, and stages of the different cohorts analyzed. The mini-Table displayed in Fig S2A is a good model, but it should incorporate information on MSI status. In addition, distinction of sporadic MSI-H cases vs Lynch Syndrome cases should be made in Figures and Tables.

*What is the MSI sensor threshold that Authors have used to define MSI status? This needs to be added to the methods description.

*Regarding the methodology for the quantification of the immune cell types, can Authors provide statements in the methods on the size of the ROIs considered? Were the counts normalized per size of the ROI? Or are these raw counts?

*In Fig 1A the unit in the X-axis is missing. In Figs 1D, E, F units in the Y-axis are missing.

*Fig 1C, mutation burden should be provided as number of mutations per megabase. It is now a consensus in the field that hypermutant tumors display at least 10 mutations per megabase (which is equivalent to MMR-deficient/MSI-H tumor). In addition, it will be easier if authors can define the MMRd status in the covariates as: sporadic MSI-H (MLH1 methylated/BRAF mutated), Lynch Syndrome (germline MMR mutations) or somatic MMRd (double hits in MMR genes/unusual but possible).

*Fig 2A. The additive effect of MSH3 and MSH6 mutations is not clear. The Authors need to discuss it in the paper (discussion section).

*Fig 2D. How do the Authors explain the concept that indels do not increase but decrease in the proportion of somatic inactivation of MSH6 and MSH3? It seems counterintuitive.

*Fig 3. The title of the figure is misleading as the content of the data does not prove the effect of MSH3 on immune infiltration.

*Legend of Fig S1 has a typographical error 'secodnary'

*Fig S2. What is the meaning of the green color in the micrographs? Please, include it in the legend.

*Fig S6. Can the cell count per tumor for the MSH6-deficient/proficient clones be presented in separate box plots back-to-back? It is hard to compare individually if the counts are higher among MSH6-deficient clones as suggested in the text.

*Fig S7. There is a bit of discrepancy on the examples of the microsatellite lengths in double wt clones (5th clone from top to bottom seems to be more unstable than stable and resembles more the deficient clones in Fig 5 than the proficient ones) and in mutated clones showed in this figure when compared to the clonality data displayed in Figure 5. It will be a good idea to clarify what is the criteria in terms of the MSI sensor data (height of the peaks?) to call a clone unstable as there are MSH6 proficient clones in Figure 5 with sizeable peaks for insertions and still being classified and nominated as MSH6-proficient (e.g. UCL-1002 second clone)

*Fig S10. The labeling of the protein for the IHC is missing in the first and third case. It is supposed to be MSH6, but it will be nice to have it labelled to confirm it.

Author Rebuttal to Initial comments

Reviewer #1:

Expertise: hypermutator tumours

The authors describe association between MSH6 and MSH3 secondary mutations in dMMR tumors and their association with increased mutational burden, areas (clones) where that

occurs in the tumors, increased immune infiltrates in these areas and prove that using multiple platforms including specific signatures and phylogenetic trees. The authors should be commended for this work. However, these data do not support the major claims of the authors throughout the paper.

Major general concerns:

1. Although the authors show very nicely throughout the paper that the MSH6/3 indels are associated with hypermutation, and increased immune infiltrates, this does not prove the claim that this drives either immune sensitivity or escape.

For that, it would be important to see of these cancers, which responded to immune checkpoint blockade, and in failing tumors, which were the mutations which were seen at recurrence/progression. For example, disappearance of MSH6/3 sub clones in the relapsed tumor may suggest that indeed, the immune infiltrates are causing loss of these ultra- hypermutant clones or vice versa.

We would like to thank the reviewer for their kind comments regarding the breadth of our analyses.

Our analyses emphasise mechanistic insight from cross-sectional multiregion sampling of treatment-naïve tumours. We show in multiple contexts (bulk WGS samples, LCM WXS samples and patient-derived organoids) that secondary MMR homopolymer mutations increase clonal diversity and widen the repertoire of immune escape variants for selection to act upon. We provide functional evidence through immune microenvironment analyses as well as through detailed molecular evolution studies. Our work shows that the mismatch repair machinery progressively breaks down during microsatellite instable cancer progression. The new time course data in clonal patient-derived organoids we now add to the manuscript further corroborate these conclusions.

To address the reviewer's concerns with regard to immune escape we have now re-examined our mutation bias analyses from LCM exome sequencing studies. Specifically, we now report the 96 trinucleotide mutation spectra comparing patches with additional MMR mutations (either MSH6 homopolymer indels, MSH3 homopolymers indels, or both) to regions without

such additional mutations. We hypothesised increased mutational exposure in trinucleotide contexts previously

linked to MutS exposure^{1,2,3} and indeed these analyses confirm exactly this hypothesis showing increased exposure to CCT>CAT and NCG>NTG mutagenic activity, as shown below (revised manuscript Fig 3F-H).

We then examined whether this shift in global mutation bias is reflected in the local mutation bias in immune escape targets. This approach has similarly been used by other teams that have investigated adaptive mutability in the context of APOBEC mutagenesis.^{4,5} We focused on a small cohort of genes involved in (neo)antigen presentation (B2M, JAK1, CITAA and others). The results of these analyses are shown on the next page and now reported in Fig 3I-K (revised manuscript).

Together with our immune dNdS analyses reported in Fig 8A (revised manuscript) which

show unambiguously expansion (dNdS>1) of immunogenic targets in clones carrying secondary MMR homopolymer indels, we feel these results make a cogent argument that these secondary mutations act as an ON/OFF switch driving increased mutability and immune escape.

¹ <https://www.biorxiv.org/content/10.1101/2021.04.14.437578v1>

² <https://www.science.org/doi/10.1126/sciadv.abg4398>

³ <https://www.nature.com/articles/s43018-021-00200-0>

⁴ <https://www.nature.com/articles/s41586-023-06303-1>

⁵ <https://www.biorxiv.org/content/10.1101/2020.12.18.423280v4>

Second, we have explored further clinical translation of these results. We note that paired data from pre-treatment and post-treatment MSI CRCs as the reviewer suggests is currently not available and would be extremely challenging to generate for the scope of this work. As an alternative, however, we were able to explore data provided from a collaborator who has recently published on the topic of immunotherapy response.⁶ In this data multiregion pre-treatment samples were available from 16 cases of MSI CRC that were later treated with immunotherapies; 11 patients had durable responses (defined as PFS>12months). We reanalysed the raw data of this study on our variant calling pipelines to ensure level comparability in particular with regard to indel calling.

Reassuringly, analysis of these data confirm that 1) patients with secondary homopolymer indels show increased SNV burden (next page, left) and 2) this translates to increased Shannon microsatellite diversity (next page, middle) as also seen in our institutional cohort of treatment- naïve stage II and stage III MMRd CRCs. These results corroborate our data and show that MMRd CRCs carrying secondary MMR homopolymer indels show greater intratumour clonal heterogeneity which acts as the substrate for immune selection and escape.

Our analysis of progression free survival (PFS) in relation to microsatellite diversity shows that patients without secondary MMR homopolymer indels (next page, right) show a tight positive correlation between microsatellite diversity and PFS suggesting that immune selection drove outcome in these patients. By contrast this relationship is lost in patients with secondary homopolymer indels, suggesting that in these cases intratumour heterogeneity and immune

6 [https://www.gastrojournal.org/article/S0016-5085\(21\)03178-4/fulltext](https://www.gastrojournal.org/article/S0016-5085(21)03178-4/fulltext)

selection no longer drives outcome as PFS is comparatively short regardless of microsatellite Shannon diversity.

We respectfully feel though that at this these data are not sufficiently mature and indeed require, as the reviewer suggests, paired pre and post treatment samples to deduce causality. Whilst these data are therefore in line with our main hypothesis, given the caveats involved

we present them here for reviewing purposes only.

We hope the additional analyses presented above reassure the reviewer in this regard.

2. Similarly, the evolution of clones in the mathematical model and the analysis can be just a matter of tumor fitness and additional mutations causing disappearance of these clones. The cause and effects with immune surveillance is not clear.

Our mathematical model includes two processes that substantially decrease clonal fitness and lead to the elimination of clonal lineages. These processes are: 1) the accumulation of a high neoantigen burden, and 2) acquisition of a lethal mutation (which becomes more likely with increased burden). We designed our model so that these two processes can be queried independently through setting the parameters s (immune selection strength, with 0 indicating no selection and -2 for strong selection) and p_{lethal} (probability of gaining a lethal mutation). We found that tumour growth consisted of two phases, (i) first governed by immune elimination up until immune escape was acquired, (ii) then subsequent growth slowed down by a proportion of subclones eliminated through lethal mutations.

To show that immune surveillance is the dominant force during this first phase, we simulated tumours at varying selection strength s (Figures 6E revised manuscript). We observed that the number of eliminated lineages increased with increasing immune selection. The capacity

to switch mutation rate carried no evolutionary advantage at very weak immune selection, but was associated with significantly fewer eliminated lineages (i.e. greater survival/fitness) as immune surveillance increased. We have reproduced Fig S9A below to illustrate this point – with increasing immune selection s (right panel) a far greater number of lineages are eliminated (left violin in blue), as expected. However, increasing mutation switch rates β (0 to 0.02 in incremental steps) allows faster selection of immune-adapted lineages and greater survival (right violin in orange). This supports our empirical data that immune selection is the underlying mechanism behind lineage elimination and that mutation rate switching provides fitness advantage.

Our organoid data now confirm that in the absence of immune selection during *in vitro* culture secondary MMR homopolymer mutations are not favoured and gradually disappear from the population. It is important to emphasise that MMR homopolymer mutations undergo secondary selection and the negative selection of clones carrying these mutations *in vitro* is a consequence of the deleterious mutations they provoke. We feel that the patient-derived organoid data now included in this revised manuscript further highlight the added value of our model simulations.

3. I would suggest to tone down the claims throughout the paper and focus on the tumor mutational burden and subclonal fitness and evolution.

We thank the reviewer for this comment. We have now amended the text to stress that further studies will require longitudinal data to assess the co-evolution of immune and mutation rate dynamics, in particular with regard to outcome prediction.

Specific comments:

1. The MS-indel mutations in MSH6/3 are presented as drivers and the data in Figure 1 (and other figures after that) just suggest association between these alterations and increased TMB. Since MSI accumulate over time in a stochastic fashion in MMR tumors, this can be just a marker of ongoing mutation accumulation and not a driver

of increased mutagenesis.

We thank the reviewer for this important comment. We submit three lines of evidence to show that these microsatellite indel mutations act as drivers of mutation rate/mutation bias, rather than due to increased background mutation rate:

1. Our multiple linear regression analysis (Figure 1C revised manuscript) finds that frameshifts in other commonly mutated homopolymers (BAX, TGFBR2 etc) do not associate with increased mutation burden. Only frameshifts in MSH6 and MSH3 showed a significant correlation with increased mutation burden. This suggests that these mutations are not just bystanders but are causative.
2. We also find that tumours with MSH6 and MSH3 frameshifts have a qualitative change in mutation bias with increased C>T and C>A contribution as observed in our microdissected whole exome sequencing data (figure 2D-F). This shift in mutation bias reflects the known mutational profile associated with MutS loss and is in keeping with the additional roles of MutS in repair of spontaneous deamination of methylated cytosine (Sanders) and oxidative damage (Nik-Zainal) that has been recently described in the literature.
3. We have now included further experiments in clonally derived patient-derived organoids allowing us to recapitulate this experiment in independent repeats. These data exactly corroborate our *in vivo* data from bulk datasets as well as from LCM WXS.

Overall, we feel this is conclusive that these secondary MMR mutations are functional in the context of preceding MLH1 loss.

Similarly, other associations of mutations with neoantigens are known and not specific. These do not prove the driving forces of these mutations.

Our neoantigen analysis (Figure 5J revised manuscript) shows that the increased mutation burden observed in tumours with MSH6/3 frameshifts is associated with an increased neoantigen burden, as expected. It follows that tumours with greater neoantigen burden will experience greater immune selection pressure. We go on to show that this is indeed the case with tumour regions with MSH6 and MSH3 frameshifts displaying increased

immune infiltration (Figures 5D-H revised manuscript) and immune escape events (figure S8). In light of the reviewer's concerns, we have now expanded our neoantigen analysis to also take into account the impact of nonsense-mediated decay (NMD). We show that tumours with MSH6 and MSH3 frameshifts also show an elevated frequency of neoantigens predicted to escape NMD (see below and query 6 reviewer 2) and also find that a number of these neoantigens have been previously described as validated neoantigens capable of eliciting a host immune response.^{7,8,9}

Lastly, it is unclear why MS-indel mutation in MSH6 is a stronger driver of dMMR than missense mutation in the same gene.

The MSH6 C8 indel causes a frameshift mutation which disrupts the downstream open reading frame likely targeting the transcript for nonsense-mediated decay. This is corroborated by decreased transcript abundance in tumours carrying this variant (Supplementary Figure 1G-H). Our IHC and LCM analyses convincingly show complete depletion of protein expression in subclones with this alteration (see also below). Indeed,

the corresponding MSH6^{F1088fs} entry in ClinVar is marked as pathogenic (<https://www.ncbi.nlm.nih.gov/clinvar/RCV000410401/>) and patients carrying this alteration in the germline demonstrate Lynch syndrome. The C8 indel is significantly enriched in MMRd tumours, because it occurs at a hypermutable homopolymer site.

Missense mutations may not have a similar disruptive impact. To directly address the reviewer's query, we looked for other coding MSH6 mutations both in our microdissected WES dataset and in the Genomics England (GEL) whole genome sequencing (WGS) dataset. In both datasets we find, as expected, that the vast majority of MSH6 coding mutations occur at the C8 homopolymer (see Fig 1D revised manuscript). Amongst MSH6-deficient samples in the WES dataset we found

⁷ <https://www.nature.com/articles/s41467-020-17526-5>

⁸ <https://www.nature.com/articles/s41467-020-18514-5>

⁹ <https://doi.org/10.1016/j.cell.2020.11.004>

that in all cases the MSH6 C8 frameshift was present, whilst a co-occurring coding SNV of the MSH6 gene was present in only 2 samples. This is detailed in the table below. This indicates that the main driver of loss of MSH6 expression in these samples is frameshift at the C8 homopolymer.

In the larger GEL WGS cohort, 60% (n=81/134 MMRd cases) of all coding MSH6 mutations were frameshifts at the C8 homopolymer. We did observe 27 missense mutations in the MSH6 gene (see Supplementary table S2B of our manuscript), but these were not recurrent events instead occurring at different sites across the cohort and some of these again coincided with C8 mutations.

In summary, although we do not exclude the possibility that at least some missense MSH6 mutations outside of the C8 homopolymer might also be pathogenic and drive an increase in mutation rate – in fact we feel this is likely –, these mutations are much less common than C8 frameshift mutations and hence we cannot reliably query this within the available data. We have now made this explicit within our manuscript. We hope this reassures the reviewer in this regard.

Sample ID	MSH6 IHC status	MSH6 C8 FS present	MSH6 C8 FS VAF	MSH6 coding SNV present	MSH6 SNV type	Genomic coordinate	Reference	Alternate	SNV VAF
s110	Deficient	Yes	0.268	No					
s30	Deficient	Yes	0.41	No					
s33	Deficient	Yes	0.28	No					
s6	Deficient	Yes	0.4	Yes	non-synonymous	48032070	G	A	0.06
s5	Deficient	Yes	0.14	No					
s61	Deficient	Yes	0.22	No					
s112	Deficient	Yes	0.57	No					
s55	Deficient	Yes	0.3	No					
s56	Deficient	Yes	0.29	No					
s65	Deficient	Yes	0.06	No					
s12	Deficient	Yes	0.279	No					
s13	Deficient	Yes	0.315	No					
s48	Deficient	Yes	0.49	Yes	non-synonymous	48025850	G	A	0.27
s52	Deficient	Yes	0.22	No					
s53	Deficient	Yes	0.17	No					
s60	Deficient	Yes	0.31	No					
s22	Deficient	Yes	0.71	No					
s23	Deficient	Yes	0.62	No					
s24	Deficient	Yes	0.62	No					
s26	Deficient	Yes	0.55	No					

2. The authors should be commended for the great work using LCM for clonal

analysis. Can the authors look at mutations in MSH6/3 in these areas as driving loss of expression?

We thank the reviewer for their kind support of our work. We present several lines of evidence in the manuscript to show that these frameshift mutations are associated, as expected, with complete loss of protein expression. Our exome sequencing data allows detailed quantitative insight into the distribution of microsatellite lengths within the MSH6 and MSH3 homopolymers. Here, C8 and A8 are the wild-type lengths and either expansions or contractions are abnormal. LCM target regions were selected based on MSH6 protein expression (proficient or deficient) and we therefore have an orthogonal internal control for our sequencing data. These regions will also include a small admixture of wild-type stromal alleles. In the manuscript figures (Fig 7 in the revised manuscript) these wild-type alleles on the allele length distribution plots are shown in beige, whilst the abnormal alleles are shown in grey. Please see the example below.

In the example in the top panel we know that the region has lost MSH6 expression and indeed in the allele length distribution we find two prominent (tumour alleles) grey peaks on either side of the stromal reference peak in beige. This corroborates the IHC data. In the bottom panel of the example above we know that the region has retained MSH3 expression. Indeed only the reference length is retrieved in the sequencing data.

The heatmaps for the microsatellite length distribution are shown in Fig S11 and the raw counts are available in the Supplementary tables datasheet.

Turning to our data we find in the microdissected WES cohort that all regions with loss of expression of MSH6 had evidence of a biallelic frameshift in the MSH6 C8 homopolymer. Since this is a coding homopolymer, a frameshift here changes the downstream amino acid sequence resulting in a non-functional protein and loss of protein expression at immunohistochemistry. An excellent example is the tree shown for patient UCL_1002 shown above. In the top clade both

MSH6 and MSH3 protein expression is lost and indeed prominent contracted and expanded tumour peaks are found (note that the stromal reference peak remains visible in the length distribution). In the MSH6 proficient clade we find that a prominent +3 tumour allele peak appears, correlating with reestablishment of reading frame and restoration of protein expression (Nb. the actual protein will have one additional amino acid; the impact on function is unknown but the sequence is translated). Similarly, for MSH3 we find that regions with loss of expression had a frameshift in the MSH3 A8 homopolymer and no other missense mutations were observed.

Overall, we feel that these experiments unequivocally show that frameshift mutations at the coding homopolymers in MSH6 and MSH3 indeed drive loss of expression.

3. One should not over interpret the data in Fig 1G-H. These are not the same as bacterial mutations but rather association of MSH6 loss when MS-indel occurs in the exome.

We thank the reviewer for this comment. We have now made this more explicit in the discussion to our paper appropriately addressing the differences.

4. Can the authors compare their signatures A, B and C with the known signatures which are common in dMMR which are Cosmic 6, 14, 15 and the ones which are more specific to MutS alfa which are 1 and 5? This may contribute to their claims in Figure 2.

We thank the reviewer for this comment.

In our revised manuscript we have removed these *de novo* signatures as we felt that this analysis distracted from more relevant downstream target, namely genes involved in immune escape/neo- antigen presentation, as also suggested by reviewer 2.

We have carried out the analysis requested by the reviewer for reviewing purposes and queried known MMR signatures in our targeted WES data. The known cosmic MMRd signatures are indeed observed in these samples, as expected. We find that samples with MSH6 and MSH3 frameshifts have increased contribution of cosmic SBS signature 15 (see figure below), which is a MMRd signature defined by a strong C>T peak in a GCG context. This is entirely in line with our finding that secondary MSH6/3 mutations drive a change in mutation bias.

We present these data for reviewing purposes although we would be happy to include this if the reviewer wishes.

5. Figures 5 and 6 (cf. Figures 7 and 8 in revised manuscript) are repetitive to the claims in the initial figures and do not add to the hypothesis that immune evasion is the main reason for the intratumoral heterogeneity in MSH6/3. Wonder if these should come before Figure 3 as the immune microenvironment can come as a final suggestion of immune involvement in clonal expansion of these cancers.

The final figures to our paper show independent phylogenetic evidence that the C8 homopolymer erodes over time out-and-in of reading frame (acting as a genetic on/off switch) and demonstrate quantitative molecular evolution analyses supporting selection of C8 mutations with increased immune selection and increased mutation rates from the accumulation of subclonal passengers.

Our goal in this paper is to emphasise that mutation rates are under selection during MMRd tumour evolution and modulated by incremental MMR mutations, which is why we have reserved these key data points to the final figures. With the addition of the organoid experiments as key mechanistic experiments we now feel that the immune microenvironment are best presented in tandem with these.

6. As such, Figure 4 can go to the supplements.

Simulation studies are a key feature of statistical genetics approaches to interrogate dynamic evolutionary systems. Our analysis reveals that mutation rate evolution introduces a heavy genomic penalty even after immune evasion, which drives counterselection of clones with lower genomic mutation rates. Our PDO experiments (Fig 4 revised manuscript) and the phylogenetic analyses (Fig 7 revised manuscript) corroborate this dynamic tumour evolution *in vitro* and *in vivo*, respectively. For this reason we strongly feel that the mathematical model is a key element and should be retained within the body of the manuscript. If the reviewer and the editor nonetheless feel that is it better shown in the supplements then we will follow their guidance.

Reviewer #2:

Expertise: immunogenomics, immune escape, tumour evolution

This study examines the contribution of mutation rate evolution to immune escape in mismatch repair-deficient colorectal cancer. The authors mapped the clonal structure of MMRD CRC and show that MMRD mutability co-evolves with neoantigen selection to drive diversification and immune escape. They find that MSI modulates subclonal DNA repair at MSH6 and MSH3. They find that MMRD helps generate intratumour heterogeneity by adapting mutation rate to immune selection. This is a nice study which demonstrates the importance of neoantigens and immune regulation in MMR tumor growth and provides a mechanism for regulated hypermutability.

Specific comments follow.

Comments:

1. The application of conserved models for regulated hypermutability in cancer cells is a nice approach.

We thank the reviewer for their kind words.

2. Homopolymer sequence is notoriously difficult to deconvolute using bulk genomic sequencing despite purity corrections and may be confounded by lack of power due to depth of sequencing. The authors need to provide power calculations to show they have sufficient power to detect the range of subclonal alleles.

We thank the reviewer for raising this important point. Indeed, we agree that the known noise of NGS experiments, particularly in relation to homopolymer stretches, combined with the fact that some subclonal variants are relatively low frequency means that experimental power is an issue to be particularly mindful of. Our experimental design was such that the average sequencing depth at the MSH3 and MSH6 loci was 300X and 379X, respectively. The minimum VAF that we accepted (conservatively) as a putative variant in our data was 5%, though in reality the minimum observed VAF is 1.3% with median frequencies of both MSH3 and MSH6 mutations at 20%.

Taking the commonly reported mutation error rate expectation of 0.01%¹⁰ and taking minimum observed depth and mutation frequency (150X and 5%, respectively), the statistical power is 0.78.

¹⁰ <https://pubmed.ncbi.nlm.nih.gov/30026539/>

Taking average values (300X and 17%) the power climbs to 0.98. In the best-case example in our data (600X and 35%) the power is 0.99 (analyses performed in G*power software).

It is also worth noting that we did consider tumour purity and sequencing depth as potential confounders in the analysis as our linear mixed effects model corrects for tumour purity (manuscript Table S4). We also found that the observed differences between MSH6/MSH3 mutated groups was not impacted by variation in sequencing depth (Supplementary figure S7D- G).

We hope that together these analyses reassure the reviewer with regards to experimental

power. We have now also briefly highlighted this in the methods to our manuscript.

3. In Fig. 1 bulk data, it is not clear that the authors controlled for phasing of alleles. Were both alleles of MSH3 and 6 affected? One would need to show that both alleles would be affected if the association between MSH3 and 6 changes are causative for mutation generation.

We thank the reviewer for allowing us to comment on this point, in particular because our patient- derived organoid data have yielded important new insights in this regard.

Indeed, as the reviewer states, the bulk WGS data presented in Fig 1 do not allow allelic phasing. This is because of the nature of the hypermutable sites (mutations in *cis* or in *trans* are indistinguishable) combined with the *absence* of a nearby common SNP or an orthogonal control such as matched IHC. Clonal mixing of progressive erosion events further compounds this issue (a -2 and a -1 erosion event could be nested events in *cis* or separate events in *trans*).

Our WES analyses presented in Fig 3, however, do allow us to interrogate this, because the MSH6 IHC analysis acts as orthogonal control to assess the allelic status of the locus when analysed in conjunction with the sequencing data. Interestingly, we find that loss of one MSH6 allele shows a dose-dependent increase in clonal mutation burden ($p=0.01$ for loss of one allele compared to MSH6 wild-type and $p=0.0039$ for loss of both alleles compared to MSH6 wild-type, see violin plot below, now included as Supplementary figure 3C).

Our mutation accumulation experiments in patient-derived organoids (now included in Fig 4 revised version of this manuscript) further corroborate this point and show that isogenic lineages with secondary hemizygous (i.e. mono-allelic) MutS mutations show increased mutation rates.

We note in this context that there is a solid body of evidence from studies in cell lines and animal models indicating that loss of a single MMR allele – particularly in the context of high baseline mutation supply – can diminish repair capacity in a dose-dependent manner.^{11,12,13,14,15,16} For instance, Marra et al¹⁴ found that hemizygous (+/-) MSH2 cells were on average four times more tolerant to temozolomide killing than wild-type cells.

Together, our experiments suggest that in the context of high mutation supply (in this case truncal MLH1/PMS2 loss) the MMR pathway does not behave like a classic two-hit tumour suppressor and can be sensitive to haplo-insufficient dosage effects. Such dosage-effects might further ‘fine tune’ mutability in evolving MMRd tumours. (Nb. In normal tissues, where background mutation supply is low, loss of a single allele is not penetrant because DNA repair demand does not outstrip hemizygous repair capacity.)

11 <https://www.nature.com/articles/1204449>

12 <https://www.sciencedirect.com/science/article/pii/S1568786421001348?via%3Dihub>

13 <https://onlinelibrary.wiley.com/doi/full/10.1002/humu.22605#humu22605-bib-0018>

14 <https://pubmed.ncbi.nlm.nih.gov/11416201/>

15 <https://pubmed.ncbi.nlm.nih.gov/21538690/>

16 <https://pubmed.ncbi.nlm.nih.gov/9751765/>

This is a topic for further study and pending further detailed dissection using for example the full range of permuted compound CRISPR KO cell lines, we have included these data in our manuscript with appropriate caveats.

4. Fig 1G. The authors claim to have found restored MSH6 function. This could come from stromal cells. The authors have not ruled this out. Single cell spatial sequencing is needed to determine whether this claim is valid.

We thank the reviewer for this comment. We should point out firstly that we provide direct phylogenetic evidence in Fig 7 of our manuscript to corroborate the conclusions drawn in Fig 1G (Fig 2G in revised manuscript). These lineage trees show reversal of reading frame through gold standard phylogenetic approaches based on thousands of mutations from WES. Our patient- derived organoids demonstrate further absence of reversion of homopolymer reading frame when organoids are cultured in absence of immune selection.

The immediate purpose of the experiment in Fig 2 is to show that IHC labelling faithfully records the allelic status of the MSH6 locus, which is pivotal to our LCM experiments reported in Figure 3. In Figure 1G (and Supplementary figure S2D) we show examples of MSH6-deficient tumour regions with nested clones that have restored MSH6 expression. Whilst infiltrating stromal cells are present throughout these tumours, these cells have a morphologically different appearance from tumour cells and can be clearly distinguished on the IHC images from tumour cells. All images and LCM target regions were reviewed by a board-certified pathologist (lead contact MJ). The reviewer is correct that DNA extracted from these LCM regions will inevitably contain a (small) admixture of stromal cells which have a normal wild-type length at the MSH6 C8 homopolymer. However, our LCM technique enriched for tumour cells and the Sanger sequencing experiments show that nested MSH6-proficient regions have a normal wild-type length at the C8 homopolymer, whilst the MSH6-deficient region has an insertion event (internal control).

To further interrogate this point, however, we have now carried out additional multiplex immunofluorescence experiments (figure below), showing that within an MSH6-deficient region we observe small nested epithelial clones/ribbons – and indeed individual tumour

cells - with restored MSH6 expression. Here, we have used pan-cytokeratin as an epithelial marker which is positive in all tumour cells, but negative in immune cells. We can be confident therefore that labelled cells with restored MSH6 expression are tumour cells rather than stromal cells.

Multiplex immunofluorescence image of MLH1/PMS2-deficient CRC labelled for MSH6, Pan-CK, CD8, CD4, CD20 and FoxP3. (i) MSH6-proficient tumour region, (ii-iii) MSH6-deficient tumour regions. Magnified images show that within MSH6-deficient regions, isolated tumour cells have restored MSH6 expression.

5. Developing ORION is commendable; however, the authors should validate the generalizability of the interpretation by using other spatial workflows. For example, “adaptive thresholding” and ellipsoidal modeling have a tendency to introduce biases.

We would like to thank the reviewer for this comment. We first validated ORION using 3 independent datasets. This consisted of two publicly available datasets and a subset of the multispectral IF images of MSI tumours created in this study.

We then compared the performance of ORION against 8 state-of-the-art cell segmentation approaches. This included both traditional and deep learning models as detailed in Supplementary table S6. We have now emphasised these extensive validation studies in the main text.

ORION is based on an established unsupervised ellipsoidal model¹⁷ that does not require labelled data, extensive training or parameter optimization. Therefore, ellipsoidal models are capable of providing reliable segmentation results in unannotated multiplex IF datasets that include a high degree of heterogeneity. Furthermore, as the multiplex IF images exhibit considerable foreground and background intensity variations, the employed adaptive thresholding method aims to improve accuracy by reducing the sensitivity of errors in cell boundary detection. We note that deep learning models exhibit lower accuracy than ellipsoidal modelling-based algorithms as the number of annotated training examples is limited and the application of transfer learning to datasets with high diversity between single data such as multiplex IF data is challenging. Results of the application of both ellipsoidal and deep learning models are consistent with previous experiments conducted in the publicly available datasets used for validation of this study.^{18,19}

The description of the ORION and the analysis of the results have been updated accordingly in the supplementary methods in subsection "ORION cell segmentation workflow" and in Table S4.

6. The neoantigen burden estimation in Fig3 (*cf. Fig 5 in revised manuscript*) is incomplete. MSI functions by introducing indels, which can generate stop codons that cause nonsense-mediated decay. The investigators need to take this into account. Furthermore, there is no validation of any of the neoantigen findings. The authors should validate some of the candidate neoantigens. Dextramer staining is a reasonable approach for this.

The reviewer raises an important point in highlighting that frameshift mutations frequently introduce premature termination codons (PTCs) which are targets of nonsense mediated decay (NMD) resulting in degradation of candidate neoantigen mRNAs. We have now refined our neoantigen analysis to take into account the rules of NMD. NMD is known to

operate less efficiently when PTCs are present in either the last exon, the penultimate exon within 50 nucleotides of the 3' exon junction, or first exon within the first 200 nucleotides of the coding sequence. We have therefore filtered our neoantigen calls to identify neoantigens that fall within these NMD escape locations. This analysis shows that tumour regions with MSH6 and/or MSH3 frameshifts have significantly higher numbers of NMD escape neoantigens (violin plot below).

17 Barmoutis, P., Di Capite, M., Kayhanian, H., Waddingham, W., Alexander, D.C., Jansen, M. and Kwong, F.N.K. Tertiary lymphoid structures (TLS) identification and density assessment on H&E-stained digital slides of lung cancer. Plos one 16.9 (2021): e0256907.

18 C. Panagiotakis, A. Argyros, Region-based Fitting of Overlapping Ellipses and its application to cells segmentation. Image and Vision Computing 93, 103810, (2020).

19 Ibtehaz, N., & Rahman, M. S. MultiResUNet: Rethinking the U-Net architecture for multimodal biomedical image segmentation. Neural Networks 121, 74-87, (2020).

Separately, we have further validated our predicted neoantigens by searching the literature for experimentally validated neoantigens that are recurrently observed in MSI tumours and

known to elicit strong CD8 T-cell responses in healthy controls and patients.^{20,21,22} These validated neoantigens are listed in the table below. We then searched our neoantigen data and found that these validated neoantigens are indeed frequently observed in our cohort. Here we include neoantigens that are both predicted to escape and act as targets of NMD, noting that NMD efficiency is not 100% in the context of a cancer.

²⁰ <https://www.nature.com/articles/s41467-020-17526-5>

²¹ <https://www.nature.com/articles/s41467-020-18514-5>

²² <https://doi.org/10.1016/j.cell.2020.11.004>

Gene	Chr	Genomic coordinate	Homopolymer	Exon position
SLC35F5	2	114500277	T10	other
SEC31A	4	83785565	T9	other
SLC22A9	11	63149671	A11	other
TTK	6	80751897	A9	last

SETD1B	12	122242658	C8	other
RNF43	17	56435161	C7	other
ASTE1	3	130733047	T11	last
TTK_2	6	80751910	A7	last
LTN1	21	30339206	A11	other
TGFBR2	3	30691872	A10	other
MYH11	16	15802687	C8	penultimate_50bp

Table. Details of experimentally validated neoantigens obtained from literature (Supp table 9)

Although numbers are small, we see an overall trend of increased numbers of validated NMD escape neoantigens in samples with MSH6 and/or MSH3 frameshifts (see figure below).

7. It is unclear whether the immune infiltrates near the MSH loss subclones are functional. The authors should use functional markers for T cell activity to provide this information.

We have performed additional IHC for markers of T cell activity / exhaustion: granzyme B, PD-1, PD-L1. In line with previous studies, these data show a corona of PD-L1 around the tumour bed (likely macrophages, not shown), but little discernible PD-L1 labelling in the tumour core. Similarly, granzyme and PD-1 show punctate labelling throughout the tumour bed (see figures this page and next, lymphoid follicles shown as internal control), but no appreciable differences between MSH6-proficient and MSH6-deficient clones.

We will point out that these data should not be taken to indicate definitively that these

infiltrates are functional as this would require dedicated flow studies on explant culture material which is currently only possible on bulk material without access to information on lineage make-up.

Again, we provide these data for reviewing purposes. Should the reviewer ask us to refer to this in the main text then we are happy to do so, of course with appropriate caveats.

8. None of the data in Fig 3K-L account for allelic zygosity. One would assume you need both alleles inactivated. Is this true? Without that, the data is not interpretable.

Please see your query above (point 3) where we address this point with regards phasing of alleles in bulk sample data Fig 1.

In short, our time course experiments in clonally derived patient-derived organoids in combination with our LCM/WXS data indicate that – in the context of high mutation supply – mono-allelic MutS mutations behave haplo-insufficient with regards cellular DNA repair demand and thus drive increased mutation accumulation. Together, these data suggest a more complex model requiring the full range of allelic combinations in isogenic cell lines

that are not tumour derived in presence/absence mutagenic agents to accurately measure mutation rate and mutation bias. Although such mutagenesis studies are beyond this manuscript, here we have included these PDO time course experiments in our manuscript with appropriate caveats.

We have also further analysed the VAF plots of 20 LCM patient samples all of which have a detected frameshift in MSH6 (red line and header text - below plots). All samples are confirmed diploid at the MSH6 locus. The reviewer will appreciate that 6 of these cases (s112, 48, 22, 23, 24, 26) display evidence of a bi-allelic frameshift at MSH6, which causes the corrected VAF of the

frameshift mutation to be higher than expected compared to if it were a hemizygous mutation ($p < 0.001$). The remainder are not significantly different and all fall within VAF 0.5, suggesting that the MSH6 alteration is hemizygous. These cases show retention of MSH6 labelling.

In light of this in Fig 3K-L we have stratified by presence of MSH6 and/or MSH3 indel homopolymer mutation (that is, 4 groups) as the number of permutations when stratifying by all possible allelic states (9 groups) would be underpowered and we would be at risk of a type II error. We feel our approach is justifiable.

9. Fig 3N-P. The interpretation here is dependent on arbitrary definition of what is subclonal or clonal NA.

We thank the reviewer for this query. We have used a validated package, PALIMPSEST, to unambiguously call variants as clonal or subclonal according to their cancer cell fraction (CCF).²³ For each SNV, the cancer cell fraction (CCF) was calculated using the variant allele fraction, tumour purity, copy number of the locus. The package determines the 95% confidence interval of the VAF of each variant using a binomial test. A mutation is considered subclonal if the upper boundary of the 95% confidence interval was <0.95 or is otherwise

considered clonal. This workflow has been used extensively before, for instance to identify subclonal driver mutations.²⁴

10. The analysis ignores immune evasion mechanisms like anti-PD1. Do the subclones have differential expression of immune checkpoints? This would surely affect immunosurveillance.

We thank the reviewer for this comment.

We would concur with the reviewer that evidence so far indicates that tumour cells in general might employ multiple immune evasion mechanisms simultaneously. Our data reveal one evolutionary immune escape pathway open to hypermutator cancers, but this is not to the exclusion of others (for example cGAS-STING²⁵).

To test for differential expression of immune checkpoints, we compared the expression of PD-L1 across the four types of TCGA CRC, UCEC and STAD dMMR cancers. (Nb. Since the Genomics England WGS cohort has no paired RNA sequencing data, we could only test for immune checkpoint expression in TCGA cohorts.)

While there was a slight trend for increased PD-L1 expression, we found no significant difference between the groups, neither when considering all dMMR cancers nor in CRCs separately (figures top and bottom panel below, respectively). Even double wild-type and double mutant (MSH6/3 wt/wt vs mut/mut) were not significantly different ($p=0.12$).

These data corroborate our IHC data for PD-L1 shown above.

We observed the same for CTLA-4 expression (Kruskal-Wallis test $p=0.45$, not shown).

23 <https://www.ncbi.nlm.nih.gov/pmc/articles/PMC5670220/>

24 <https://pubmed.ncbi.nlm.nih.gov/25877892/>

25 <https://pubmed.ncbi.nlm.nih.gov/33338427/>

Figure: PD-L1 expression between tumour types within dMMR tumours from the TCGA CRC, UCEC and STAD cohorts (top panel) and from TCGA CRC cohort alone (bottom panel).

11. Fig. 5. The authors interpret the data as MSH 6 and 3 homopolymer diversity as driving immune escape solutions, but the latter is not well supported. Only two cases are shown that support this contention.

We can understand the reviewer's comment here that of the three tumour phylogenies in Figure 5, only two patient phylogenies show examples where the MSH6-deficient clone has additional immune escape solutions.

However, when we look across the whole cohort, we find that samples with MSH6 or MSH3 frameshifts have a significantly higher number of HLA or antigen processing machinery genes. Importantly, these additional mutations in HLA or antigen processing machinery genes show mutation bias consistent with secondary MutS mutations (Fig 3I-K).

We had previously included this data as Fig S8, but we have now moved this figure to the main figures to emphasise this key point. We thank the reviewer for highlighting this.

Furthermore, the non-synonymous mutation in RFX5 is of uncertain consequence. Is this a functional mutation? Data here is lacking.

In the manuscript (Figure 7) we report an RFX5 C>T p.P72L mutation, which appears to be rare and possibly therefore not represented in the COSMIC or ICGC databases. Our assessment that this variant is pathogenic is based on the SIFT and POLYPHEN evolutionary conservation methods, which both returned "deleterious".

We have now included this information in the main text.

How many reverter lineages have immune escape mechanisms?

We examined 10 tumours with dense multiregion sampling and using phylogenetic analysis we found two instances of reversion and both had evidence of immune escape events using stringent calling of pathogenic variants as per the above. We will point out that the complexities of this analysis mean that we were limited to sampling larger (reverter) clones with sufficient input for WXS. A targeted analysis using a smaller bait panel restricted to homopolymer regions and immune escape targets may allow us to revisit this point and target smaller regions. At present this is technically complex.

Reviewer #3:

Expertise: colorectal cancer genomics incl. hereditary syndromes, pre-cancer

Remarks to the Author:

The manuscript titled '*Mutation rate evolution drives immune escape in mismatch repair-deficient*

cancer by Kayhanian/Jansen et al is an original contribution submitted to Nature Genetics for consideration. In brief, authors have performed detailed mutational analysis from WGS data in MMRd colorectal cancers from the Genomic England cohort (N=217) and profiled frameshift mutations (indels) impacting coding microsatellites. Two specific somatic mutations within coding microsatellite tracts located in MSH3 (A8) and MSH6 (C8) have been correlated with increases in mutational rate, indels, and therefore neoantigen generation. The vast majority of MMRd CRC demonstrate the acquisition of somatic mutations in either of these two microsatellite tracts or even simultaneously. An independent validation of the genomics results has been performed in a limited number of TCGA samples of other tumors known to display MMRd. Then authors used an institutional cohort of CRC cases to perform a detailed topographic multiplex protein expression analysis of MSH6 and immune cell markers via IF coupled with specific MS analysis of the A8 and C8 repeats. Subsequent genomic analysis of subclones captured by LCM and stratified by somatic MSH6/MSH3 frameshift mutational status showed specific mutational signatures and accumulation of C>T changes. Protein expression analysis demonstrated that variegated MSH6 loss corresponds with the presence of indels in the C8 tract. In addition, MMRd CRC with MSH6 loss present with active immune infiltration observed via multiplex-IF for several cell types (CD4 and CD8 T cells, Tregs and B cells) and neoantigen accumulation. Detailed clonal analysis suggested that subclones switch on/off the immune response by selecting these clones which could contribute to the development of immune tolerance in general. This manuscript includes sophisticated mathematical modeling on growth advantage depending on the acquisition of these somatic mutations and their contribution to neoantigen/immune activation.

Overall, this is a well-executed work that develops further the pre-existing concept that secondary somatic mutations targeting coding microsatellites play an important role in MMRd carcinogenesis. Authors have done an impeccable job in two main areas: systems biology and molecular pathology analysis.

However, the data presented here is mainly associative and does not provide final functional prove that these two specific mutations in MSH6 and MSH3 are responsible for the mutation accumulation and the immune switch, although it is more than plausible that this is the case, and it constitutes a solid hypothesis to be conclusively proved.

Therefore, a request for addition to this manuscript (major critique) will be to provide proof using an in vitro (an ideally ex vivo) systems model that these two mutations contribute to

accumulation of somatic indels. The most valid platform to perform this would be patient-derived organoids of a MMRd clone (it could be MLH1/BRAF-mt clone) that went to be CRISPR'ed with the two mutations independently and then combined. Genomic studies should prove accumulation of indel mutations and IHC of the organoids should prove the somatic inactivation of both proteins.

We thank the reviewer for their kind and supportive comments.

We have taken the reviewer's comments to heart and through a collaboration with the lab of Dr Hugo Snippert developed a series of MLH1/PMS2-deficient isogenic patient-derived organoid lines (PDOs) carrying various combinations of MSH6 and/or MSH3 homopolymer mutations.

The results of these extensive new experiments are described in new Figure 4 (revised manuscript).

In short, these data corroborate our *in vivo* observations and show that loss of one or more MutS alleles in the context of MLH1 loss increases and skews mutation bias. Importantly, mutation accumulation experiments confirmed the hypermutable nature of these homopolymers and showed that these slowly drift back into reading frame in the absence of immune selection.

We feel that the addition of these data indeed convincingly demonstrates that the secondary homopolymer mutations in MSH3 and MSH6 drive accelerated mutation accumulation. Our dedicated time course experiments in clonal PDO lineages are key to this conclusion.

Moderate critiques in the execution and presentation of this work are the following:

- 1. Genomic analyses highlight both the contribution to mutation accumulation by both MSH6 and MSH3 frameshift mutations. However, it seems like the authors abandon later the MSH3 frameshift mutation for their subsequent molecular pathology analysis, thus diminishing the value of the MSH3 somatic inactivation. Authors should provide the alternative set of experiments presented in Figure 3 for clones stratified by MSH3 status.**

The multiplex immunofluorescence work in Figure 3 (now Figure 5 in revised manuscript) was

carried out with clinical grade antibodies to MSH6, AE1/3 (pancytokeratin), CD8, CD4, CD20 and FoxP3. This meant we could reliably optimise the exact order with which antibodies were best probed and stripped during Opal (Akoya) labelling, because the expected signal of each individual antibody is well-characterised. Unfortunately, and despite trying several commercially available MSH3 antibodies, it was not possible to reliably include MSH3 in our multiplex IF work-up because none of the antibodies available to us was sufficiently specific to survive harsh stripping and reprobing conditions. Poor performance of this antibody in panels quickly becomes obstructive in computational analysis of multiplex labelling data. At this point in time we therefore cannot reliably use this MSH3 antibody in multiplex panels – however we can use it on its own in tissue sections in routine pathology analysis (cf. query below).

We note that our results in Figure 3 (now Figure 5) do include genomic data for MSH3 where appropriate (Figure 3I-O). We hope that this reassures the reviewer in this regard.

2. Connected to the previous point, do authors observe a similar variegated pattern towards re-switch of MSH3 clones as they do for MSH6 (similar to what is displayed in Fig 1G-H for MSH6)?

Indeed, we have found evidence for this switch-like behaviour for MSH3. The figure below shows a small MSH3 positive clone (dashed area) nested within a larger MSH3 negative clone. The inset shows abrupt transition from complete lack of nuclear labelling to positive nuclear labelling (arrow), which is characteristic of clonal transitions.

Our organoid data, in particular the gold standard single-cell expanded isogenic derivatives, further corroborate this point as these PDO lines drift back to wild-type reading frame over time in culture. We feel this shows conclusively that these homopolymers are (1) highly mutable, (2) their allelic status controls cellular mutability, and (3) immune selection favours higher mutation rates.

3. Clonality analysis Fig 5 (cf. Fig 7 revised manuscript). Here, authors resume the assessment of both MSH3 and MSH6 from a functional and genomics standpoint by pairing IHC with microsatellite length analysis (using MSIsensor outputs). However, it seems that the clonal analysis is only driven by the MSH6 status, and the contribution of the MSH3 data it is not clear. In addition, the statement in the texts highlights that there is always loss of MSH3, which is evident in the IHC. However, while some of the clones match the loss of MSH3 with presence of indels in the A8 repeat, there are clones that show no instability with loss of the MSH3 expression, which is difficult to understand (e.g. see two MSH6 deficient clones in case UCL1016).

The reviewer is correct that the selection of clones for LCM excision was driven by MSH6 IHC analysis. MSH3 status was determined from sequencing reads and correlated with IHC results. We therefore label individual clones on the tree according to MSH6 IHC status because our LCM

approach was guided by MSH6 IHC status.

In each case loss of immunolabelling for MSH6 as well as for MSH3 correlates with significant presence of non-reference length alleles on the homopolymer length distribution, which was not seen in cases where immunolabelling is retained. The homopolymer sequencing data thus closely track IHC data.

With respect to UCL1016, the two MSH6 deficient clones (bottom clades) show significant -1 and +1 peaks on either side of the reference peak (stroma), whilst MSH3 homopolymer length distribution for these samples shows no major peaks and immunolabelling is accordingly retained.

To make this unambiguous we have now introduced asterisks in regions of IHC loss for MSH6 as well as for MSH3 and drawn dashes around immune-negative clones.

4. Also, the MSH2 staining needs to be displayed back-to-back in the main figure and considered for the clonal analysis. The main question here is, are there clones within these tumors that are fully proficient for MSH6/MSH3 and MSH2, thus proving the restoration of the MMR mechanism as suggested in the text of the paper? The current version of the results is neither clear nor backing this notion.

Many thanks for this comment. We have now added MSH2 IHC which we agree is useful here in confirming that combined MSH6 and MSH3 loss leads to MSH2 loss.

Of the 3 tumours in Figure 7, UCL_1002 shows lineages with loss of both MSH6 and MSH3. In addition, the MSH6-proficient clone here shows restored MSH6 expression through progressive insertion events at the MSH6 homopolymer resulting in a +3 event. Accordingly, the MSH2 IHC shows that the combined MSH6/MSH3 deficient clones are also deficient for MSH2, whilst the clone with restored (+3 insertion) MSH6 expression also restores MSH2 expression. We treat this as unequivocal evidence of reversion *in vivo*.

Minor critiques to bring to Authors' attention are the following:

1. The title of the paper is generic and addresses MMRd tumors in general. The data from TCGA does not offer separate analyses for tumors of different origin. There is

evidence in the literature that there is some degree of exclusivity of certain microsatellite mutations for certain types of tissue (colorectum vs endometrium versus urothelium) See <https://www.nature.com/articles/nm.4191> to back this statement and as an important reference to be incorporated. Perhaps, the title needs to be more specific towards CRC.

We thank the reviewer for this comment. We agree that there is a degree of tumour type specificity for homopolymers that are recurrently mutated in MSI cancers and that this is described in literature. However, for MSH6 and MSH3 frameshifts we find that these are present across all the MSI tumour types. We do address this in our analysis of the TCGA data in Supplementary figures S1D-J where the analysis included all MSI tumours. For further clarification we now also provide table below detailing the breakdown of cases according to tumour type and presence or absence of MSH6/MSH3 homopolymer frameshift. Further functional dissection in other MMRd cancers is beyond the scope of this paper.

Given that the emphasis in this paper is indeed on colorectal cancer we have amended the title accordingly. The manuscript referred to has now been cited.

Tumour type	MSH6/MSH3 wild type	MSH6 ^{F1088fs}	MSH3 ^{K383fs}	MSH6 ^{F1088fs} and MSH3 ^{K383fs}
Colorectal	32	3	12	1
Uterine	47	9	9	2
Stomach	34	0	24	5
Esophageal	3	0	0	0

Breakdown of tissue type of TCGA MSI tumours according to MSH6 and MSH3 frameshift mutation status.

2. The previous reference proves that there is substantial genomic information available within TCGA to run a separate analysis per tumor site to prove that the same FS mutations occur and are responsible for the increase in mutation burden in other MMR-deficient tumor types. This will ultimately prove that the mechanism described in this paper is not exclusive of colorectal cancers and generic for all MMRd tumors.

As discussed above, we have completed this analysis of the TCGA data in figure S1 D-J. Indeed, this analysis suggests that our genomic mechanism applies across the board to all mismatch repair-deficient cancers.

3. Clinical details on the samples analyzed are insufficient for all cohorts (GEL, TCGA and UCL). Tables need to be more specific on the ages, gender, and stages of the different cohorts analyzed. The mini-table displayed in Fig S2A is a good model, but it should incorporate information on MSI status. In addition, distinction of sporadic MSI-H cases vs Lynch Syndrome cases should be made in Figures and Tables.

Genomics England Cohort: We are unable to provide further details on age and gender as this would be against the data access agreement in place to utilise this data. We have provided germline status where available in Figure 1C (see also your query 7 wrt definitive classification Lynch/somatic in these cases).

UCL cohort: We have included age groupings in table S3. Regrettably, we cannot add more specific details on age and gender as that could put a patient at risk of being identified. We have added an extra column to list pathogenic germline MMR mutations.

TCGA cohort: We have added an additional table (table S7) to include age and gender information.

4. What is the MSIsensor threshold that authors have used to define MSI status? This needs to be added to the methods description.

For identification of MSI high cases in the GEL cohort, we used default settings when running MSIsensor. As per the Github page below, the maximal homopolymer size default is 50bp and the maximal length of microstate is 5bp. The depth threshold is 20X and the false discovery rate 0.05. As per the default settings an MSIsensor score of >3.5 was used as the cutoff to call a sample MSI high.

We have added additional details to the methods as suggested.

https://github.com/ding-lab/msisensor/blob/master/README_msisensor.md

5. Regarding the methodology for the quantification of the immune cell types, can authors provide statements in the methods on the size of the ROIs considered? Were the counts normalized per size of the ROI? Or are these raw counts?

This has been amended. The size of each ROI was the same in all experiments at 1mm². The cell counts provided are raw counts.

6. In Fig 1A the unit in the X-axis is missing. In Figs 1D, E, F units in the Y-axis are missing.

This has been amended. We have emphasised these are raw mutation counts (cf. your next query).

7. Fig 1C, mutation burden should be provided as number of mutations per megabase. It is now a consensus in the field that hypermutant tumors display at least 10 mutations per megabase (which is equivalent to MMR-deficient/MSI-H tumor). In addition, it will be easier if authors can define the MMRd status in the covariates as: sporadic MSI-H (MLH1 methylated/BRAF mutated), Lynch Syndrome (germline MMR mutations) or somatic MMRd (double hits in MMR genes/unusual, but possible).

We address these queries in turn.

Mutation unit: There is significant variation between studies with regard to how mutation burden is expressed, depending on whether targeted resequencing panels, WXS or WGS data used as input. A commonly used definition in the clinical setting is to only include coding mutations in TMB calculations as these are expected to be relevant in terms of neoantigen generation. However, for the purpose of our linear regression model (now Fig 1C) we were interested in total mutation counts across the genome rather than a normalised (and potentially misleading) count in a subsetted exome from our rich WGS data. To emphasise this point we have kept this consistent throughout.

On the figure panels we have now stressed these are raw mutation counts.

MMRd status: The GEL data do not provide direct access to whether a patient is clinically registered as either Lynch or sporadic (either MLH1 methylated or bi-allelic somatic) nor do we

have access addition to methylation data. This means that for the majority of patients one can of course make educated guesses as to a patient's Lynch status, but more complex mutation combinations inevitably occur. Given that our emphasis here is not on comparing these clinical categories, we decided to report the mutation categories 'as is' rather than attempting to further classify as Lynch, sporadic or unknown.

8. Fig 2A. The additive effect of MSH3 and MSH6 mutations is not clear. The authors need to discuss it in the paper (discussion section).

We have emphasised this in the discussion.

9. Fig 2D. How do the authors explain the concept that indels do not increase but decrease in the proportion of somatic inactivation of MSH6 and MSH3? It seems counterintuitive.

Whilst both SNV and indels increase in absolute numbers, loss of MSH6 drives a *proportionally higher* increase in number of SNVs (in particular C>T and C>A events) resulting in a proportional decrease in indels.

The absolute counts are shown in Fig S3. Below we show these two plots side-by-side for the reviewer's benefit. We are happy to include this in the main figures if the reviewer so wishes. We have emphasised this point also in the main text itself.

10. Fig 3. The title of the figure is misleading as the content of the data does not prove the effect of MSH3 on immune infiltration.

The top panels (A to H, now Fig 5) all show MSH6 protein expression on multiplex IHC in relation to immune cell infiltration. (Nb. As detailed above we could not involve MSH3 in this analysis because unfortunately the MSH3 antibody labelling was not sufficiently specific to survive repeated probing/stripping cycles.) The lower panels (I to O) all show analyses based on mutation data from LCM/WXS experiments carried out on the same regions that were probed by multiplex IHC using consecutive slides. In this way we have detailed insight into MSH3/MSH6 mutation status in relation to CD8 infiltration as shown in Fig 5I. For this reason, we politely disagree and feel that the legend title (MSH6 and MSH3 homopolymer frameshift mutations accelerate clonal HLA diversity at the cost of increased neoantigen burden and immune cell infiltration) adequately describes the data in the figure.

Nb. We did not further stratify the MSH6 proficient/deficient regions shown in the top panels by MSH3 mutation status, because strictly speaking these are genotyping data from different experiments (multiplex IHC vs WXS) and different (but consecutive) sections.

11. Legend of Fig S1 has a typographical error 'secodnary'

This has been amended. Many thanks for spotting this typo.

12. Fig S2. What is the meaning of the green color in the micrographs? Please, include it in the legend.

The green areas represent stromal regions. This has been amended in the legend.

13. Fig S6. Can the cell count per tumor for the MSH6-deficient/proficient clones be presented in separate box plots back-to-back? It is hard to compare individually if the counts are higher among MSH6-deficient clones as suggested in the text.

This has been amended and included as Fig S6E-H (also shown below).

14. Fig S7. There is a bit of discrepancy on the examples of the microsatellite lengths in double wt clones (5th clone from top to bottom seems to be more instable than stable and resembles more the deficient clones in Fig 5 than the proficient ones) and in mutated clones showed in this figure when compared to the clonality data displayed in Figure 5. It will be a good idea to clarify what is the criteria in terms of the MSIsensor data (height of the peaks?) to call a clone instable as there are MSH6 proficient clones in Figure 5 with sizeable peaks for insertions and still being classified and nominated as MSH6-proficient (e.g. UCL- 1002 second clone)

We apologise if this was not sufficiently clear. We aimed to illustrate our approach to using homopolymer lengths as a proxy for population diversity (Shannon microsatellite diversity). Fig S7A is a cartoon to illustrate our workflow for the analysis presented in Fig 5 (now Fig 7) of the main figures and the remainder of the panels shown in Fig S7, in particular Fig S7C.

We have now adapted the legend to underscore this point and have amended Fig S7A as per the below:

Microsatellite length diversity analysis

Clonal genotype

UCL-1002 in Figure 5 is a highly illustrative case which shows an MSH6-proficient subclone which has evolved from an MSH6-deficient ancestor lineage (phylogenetic tree based on WXS data, left panel). The homopolymer length data for MSH6 independently corroborate this by showing progressive expansion of homopolymer length until a +3 reading frame is reached (middle) resulting in restoration of protein expression as shown on the IHC (right panels).

15. Fig S10. The labelling of the protein for the IHC is missing in the first and third case. It is supposed to be MSH6, but it will be nice to have it labelled to confirm it.

Many thanks for spotting this omission. This has been amended.

Decision Letter, first revision:

9th Jan 2024

Dear Dr Jansen,

Thank you for submitting your revised manuscript "Homopolymer switches mediate adaptive mutability in mismatch repair-deficient colorectal cancer" (NG-A59618R). It has now been seen by Reviewers #2,3 and their comments are below. Please note that Reviewer #2 was asked to comment on your response to Reviewer #1 who was unable to re-review.

The reviewers find that the paper has improved in revision, and therefore we'll be happy in principle to publish it in Nature Genetics, pending minor revisions to comply with our editorial and formatting guidelines.

Sincerely,

Safia Danovi
Editor
Nature Genetics

Reviewer #2 (Remarks to the Author):

I was asked to evaluate the authors responses to R1 comments. The authors provide nice additional analyses and it would appear that R1's comments are addressed now.

Major concerns.

1. The data presented is reasonable to address these concerns
2. Addressed
3. Addressed

Specific comments:

1. Addressed
2. Addressed
3. Addressed
4. Ok
5. Ok
6. Addressed

Reviewer #3 (Remarks to the Author):

The Authors have adequately addressed and provided valid responses to my queries.

Final Decision Letter:

25th Apr 2024

Dear Dr Jansen,

I am delighted to say that your manuscript "Homopolymer switches mediate adaptive mutability in mismatch repair-deficient colorectal cancer" has been accepted for publication in an upcoming issue of Nature Genetics.

Your paper will be published online after we receive your corrections and will appear in print in the next available issue. You can find out your date of online publication by contacting the Nature Press Office (press@nature.com) after sending your e-proof corrections.

Before your paper is published online, we shall be distributing a press release to news organizations worldwide, which may very well include details of your work. We are happy for your institution or

funding agency to prepare its own press release, but it must mention the embargo date and Nature Genetics. Our Press Office may contact you closer to the time of publication, but if you or your Press Office have any enquiries in the meantime, please contact press@nature.com.

Please note that *Nature Genetics* is a Transformative Journal (TJ). Authors may publish their research with us through the traditional subscription access route or make their paper immediately open access through payment of an article-processing charge (APC). Authors will not be required to make a final decision about access to their article until it has been accepted. Find out more about Transformative Journals

Authors may need to take specific actions to achieve compliance with funder and institutional open access mandates. If your research is supported by a funder that requires immediate open access (e.g. according to Plan S principles) then you should select the gold OA route, and we will direct you to the compliant route where possible. For authors selecting the subscription publication route, the journal's standard licensing terms will need to be accepted, including [a href="https://www.nature.com/nature-portfolio/editorial-policies/self-archiving-and-license-to-publish](https://www.nature.com/nature-portfolio/editorial-policies/self-archiving-and-license-to-publish). Those licensing terms will supersede any other terms that the author or any third party may assert apply to any version of the manuscript.

If you have not already done so, we invite you to upload the step-by-step protocols used in this manuscript to the Protocols Exchange, part of our on-line web resource, natureprotocols.com. If you

complete the upload by the time you receive your manuscript proofs, we can insert links in your article that lead directly to the protocol details. Your protocol will be made freely available upon publication of your paper. By participating in natureprotocols.com, you are enabling researchers to more readily reproduce or adapt the methodology you use. Natureprotocols.com is fully searchable, providing your protocols and paper with increased utility and visibility. Please submit your protocol to <https://protocolexchange.researchsquare.com/>. After entering your nature.com username and password you will need to enter your manuscript number (NG-A59618R1). Further information can be found at <https://www.nature.com/nature-portfolio/editorial-policies/reporting-standards#protocols>

Sincerely,

Safia Danovi, PhD
Senior Editor, Nature Genetics
ORCID: 0009-0007-7822-5479